# The effects of gravel cushion particle size and thickness on the coefficient of restitution in rockfall impacts

*Zhu Chun[1,2,3], Wang Dongsheng[2,3], Xia Xing[2,3], Tao ZhiGang[2,3], He ManChao[1,2,3], Cao Chen\*[1]*

Corresponding Email: ccao@jlu.edu.cn (First Corresponding author)

zhuchuncumtb@163.com;

*(1.     College of Construction Engineering, Jilin University, Changchun 130026, China)*

*(2.     State Key Laboratory for Geomechanics & Deep Underground Engineering, Beijing 100083, China)*

*(3.     School of Mechanics and Civil Engineering, China University of Mining & Technology, Beijing 100083,China)*

**Abstracts:** Gravel cushions are widely used to absorb the impact energy of falling rocks in open-pit mines. A particularly important application is to enhance the energy-absorbing capacity of rockfall sheds. In this paper, we study how varying the thickness and particle size of a gravel cushion influences its energy-consumption and buffering effects. We performed a series of laboratory drop tests by dropping blocks from a fixed height onto cushions of different thicknesses and particle sizes. The results indicate that, for a given impact energy, the cushion thickness has a strong influence on the measured coefficient of restitution (*COR*) and therefore impact pressure. Additional tests were performed to study how the radius of the block and the height it is dropped from affect the measured *COR*. This showed that as the movement height of the block is increased the *COR* also increases, and blocks with larger radii exhibit a larger variability in measured *COR*. Finally, we investigated the influence of rockfall block radius, *r*, movement height, *H*, cushion thickness, *h*, and particle size, *d*, on the *COR* and the damage depth, *L*, of the cushion. The test results reveal that the cushion thickness is the primary design parameter, controlling not only *COR* but also the stability of the cushion material. The results provide a theoretical and practical basis for the design of gravel cushions for rockfall protection.

**Keywords:** Rockfall; cushion thickness; laboratory test; particle size; coefficient of restitution (*COR*).

## 1 Introduction

Rockfall constitutes a serious hazard in the working areas and facilities of the world's open-pit mines. Where slope surfaces are seriously weathered and the disturbing forces from mining are strong, landslides and rock-body collapse are prone to occur during rainfall. In rockfall, rocks roll down slope due to instability caused by gravity or exogenic action and come to rest at an obstacle or in the gentler part of the slope (Huang et al., 2007). Rockfall is widely distributed and occurs suddenly, posing a serious threat to life and property (Pantelidis, 2010). In response to frequent rockfall disasters in recent years, numerous scholars in China and abroad have conducted in-depth studies into the characteristics of rockfall movement through theoretical analysis, field investigation, and numerical simulation. For example, Mignelli et al. (2014), applied a rockfall risk management approach to the road infrastructure network of the Regione Autonoma Valle D'Aosta in order to calculate the level of risk and the potential for its reduction by rockfall protection devices. A comparative analysis of road accidents in the Aosta Valley was then undertaken to verify the methodology. Asteriou et al. (2016) examined the effects of rock shape by performing tests with spherical and cubic blocks, finding that spherical blocks show higher and more consistent coefficient of restitution (*COR*) values than cubic blocks. Howald et al. (2017) evaluated the protective capacity of existing and newly proposed protection measures and considered the possible reclassification of hazard as a function of the mitigation role played by the

measure. Furthermore, numerical simulation software has been adopted to analyze the characteristics of rockfall movement. The software ROCFALL 3.0 has been adopted in dam construction, road construction and the protection of historical places to calculate the velocity and locus of rockfall and avoid damage to the project (Topal et al., 2006; Koleini and Van Rooy, 2011; Saroglou et al., 2012; Sadagah, 2015). State-of-the-art simulation techniques incorporating nonsmooth contact dynamics and multibody dynamics have been applied to and adapted for the efficient simulation of rockfall trajectories, and the influence of rock geometry on rockfall dynamics has been studied through numerical simulation (Leine et al., 2014).

The research outlined above indicates that several types of protection measure can be effective in controlling rockfall. Trees have a significant blocking effect on rolling rocks. Interception influence tests of the effect of trees on rockfall have been designed based on analysis of the velocity change, the distance traveled by the rockfall, and the probability of collision between trees and rockfall (Notaro, 2012; Monnet et al., 2017). Semi-rigid rockfall protection barriers have been installed along areas threatened by rockfall events, and Miranda et al. (2015) have carried out a numerical investigation of such protection barriers to obtain essential structural information such as their energy-absorption capacity. Furthermore, Lambert et al. (2014) conducted real-scale impact experiments with impact energies ranging from 200 kJ to 2200 kJ. They studied the response of rockfall protection embankments composed of a 4-m high cellular wall to a rock impact and compared this with previous real-scale experiments on other types of embankment. Finally, Sun et al. (2016) used a tire cushion layer to absorb rockfall impact, utilizing the radial deformation of the tire. They built a reinforced concrete structure model with a tire cushion layer and carried out artificial rockfall tests.

The protection research outlined above is mainly applicable to conventional human settlements, and it is expensive and inconvenient to use these measures to control rockfall in an open-pit mine. A relatively common way of preventing and controlling rockfall hazard in an open-pit mine is to lay an energy-consuming layer on a safety platform (Labiouse et al., 1996). However, research into such cushions seldom considers the effects of the particle size of the cushion on the characteristics of rockfall movement. In particular, the combined effects of the particle size and thickness of a gravel cushion on the coefficient of restitution (*COR*) have not yet been explored. A large amount of mullock is produced during mining, and this can be broken into particles of different sizes in a crusher and used to pave the platform as an energy-consuming layer. A certain thickness of gravel cushion on the platform can act as a buffer, effectively absorbing the impact energy of rockfall and reducing the impact load on the protective structure while also reducing the kinetic energy of the rockfall and causing it to stall. Because the impact between the rockfall and gravel cushion is of short duration, it involves complicated elastic-plastic deformation and energy conversion, and the energy absorption performance of gravel cushions of different thicknesses and particle sizes are quite different under rockfall impacts. Determining the energy-consumption buffering mechanism of a gravel cushion and calculating the subsequent rockfall movement has become the key to cushion design. Therefore, to control rockfalls effectively, it is necessary to further study the effects of the particle size and thickness of the cushion on *COR* under rockfall impact.

## 2 Coefficient of restitution

It is challenging to predict the trajectory of rebound for a rockfall because it is influenced by
several parameters such as the strength, roughness, stiffness, and inclination of the slope and
blocks (Labiouse and Heidenreich, 2009). However, the coefficient of restitution (*COR*) is widely
used for this purpose (Giani, 1992).

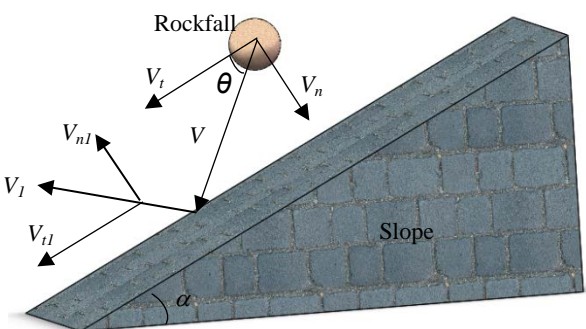


Fig.1 Motion model of rockfall

The definitions of *COR* are various (Chau et al., 2002) but for a block impacting a rocky
slope (Figure 1), it can be defined on the basis of the theory of inelastic collision as:
$$V_{COR} = \left|\frac{V_1}{V}\right|, (1)$$

where $V$ and $V_1$ are the magnitudes of the incident and rebound velocities at the locus, respectively
(m/s).
$V_{COR}$ has normal and tangential components. The normal ($R_n$) and tangential ($R_t$) coefficients
are defined as:
$$R_n = \left|\frac{V_{n1}}{V_n}\right| \; and \; R_t = \left|\frac{V_{t1}}{V_t}\right|, \quad (2)$$

where $R_n$ and $R_t$ are the normal and tangential restitution coefficients, respectively, and $V_n$ and $V_{n1}$
are the normal components and $V_t$ and $V_{t1}$ are the tangential components of the velocity of the
block before and after the impact, respectively (m/s).
The total energy, *E,* of the block consists of the translational ($E_0$) and rotational ($E_W$) energy:
$$E = E_0 + E_w = \frac{1}{2}mv^2 + \frac{1}{2}I\omega^2, \;\; (3)$$

and the total energy coefficient *($ET_{COR}$ )* is proposed to be:
$$ET_{COR} = \frac{\frac{1}{2}mV_1{}^2 + \frac{1}{2}I\omega_1{}^2}{\frac{1}{2}mV^2 + \frac{1}{2}I\omega^2} = \frac{0.6mV_1{}^2}{0.6mV^2} = \frac{V_1{}^2}{V^2} = V_{COR}{}^2, (4)$$

where *m* is the mass of the block, *I* is its moment of inertia, and $\omega$ and $\omega_1$ are the angular velocity
before and after the impact, respectively.
When a dangerous rock-body breaks away from the parent body, it will inevitably generate
collisions with the slope during the rolling process and lose energy. A formula for the approximate
calculation of the total kinetic energy of the rockfall has been derived from engineering surveys
(Yang et al., 2005; Zhu et al. 2018):
$$E = E_0 + E_w = 1.2E_0 = 0.6mV^2 = 0.6m(V_n{}^2 + V_t{}^2), (5)$$

## 3 Experimental studies

### 3.1 Experimental material and apparatus

In order to study the effects of the particle size and thickness of the cushion on *COR* under
rockfall impact conveniently, a high-strength gypsum material was adopted to simulate the
rockfall. A previous study (Chau et al., 2002) recommends a moisture content of 30–50% for the
sample, so in this study, all samples were given a moisture content of 40%.
A large number of tests have shown that spherical falling blocks have higher and more
consistent *COR* values than cubic blocks (Asteriou et al., 2016), and so that the same control
methods will have greater difficulty in containing their effects than those of non-spherical blocks
with the same properties. This indicates that spherical rocks are a common hazard and that if a
cushion is designed to resist these, it can also effectively resist non-spherical rocks. This greater
threat should therefore be the primary concern when designing a protective cushion. For this
reason, spherical blocks with radii of 2 cm, 3 cm, 4 cm and 5 cm (Figure 2a) were used to simulate
rockfall in this study. Additionally, six standard 5-cm diameter, 10-cm high cylindrical samples
were created with which to test the uniaxial compressive strength of the gypsum materials. The
uniaxial compression test is shown in Figure 2b. Due to the inherent error associated with the test,
the ultimate compressive strength of the six samples is different, so the average value is taken as
the compressive strength of the material. The average value at which the specimens are destroyed
is 6.48 Mpa, indicating that a gypsum sample with 40% moisture content is strong enough not to
be shattered during the collision process (Ulusay et al., 2007; Aydin, 2009).

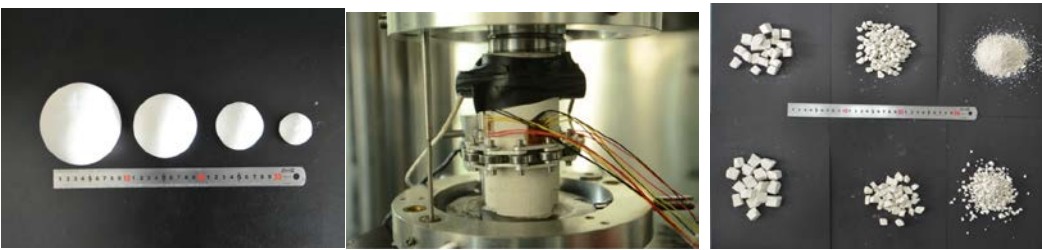

(a) Spherical gypsum samples          (b) Standard specimen under        (c) Sieved granules of different particle sizes
of different sizes                      a uniaxial compression test

Fig.2 Experimental material production and testing process

In order to explore the effect of different cushion thicknesses and particle sizes on the rolling
motion of a rockfall, massive gypsum boards with the same properties as the blocks were broken,
and gypsum particles for simulating the gravel cushion were divided by coarseness using 2 mm, 6
mm, 10 mm, 14 mm, 18 mm and 24 mm sieves (Figure 2c).
A simple rolling stone releasing device is shown in Figure 3. A tube with adjustable
inclination and height is used to vary the translational impact velocity of the blocks (Asteriou et al.,
2012). The blocks slide and roll through the tube to collide with the plate. Two synchronized
digital cameras (1024 × 1024 pixels and a 200 fps capture rate) were used to acquire the velocities
of the blocks in stereoscopic space (Bouguet, 2008; Asteriou et al., 2013).
The two cameras, which obtained the motion, velocity, and kinetic energy automatically,
were placed symmetrically at a distance of approximately 0.9 m from the impact surface (Figure
3). The distance between the two cameras was approximately 1.2 m, making the cameras look
slightly down at the targeted platform.
The synchronized recordings from the two cameras captured a sequence of image stereopairs
at time intervals of 1/200 s. By applying stereo-photogrammetric processing, the position of any
point in both images can be computed in 3D space. The image plane has a 2D coordinate system
where position measurements can be made using pixel coordinates. The camera has a 3D reference
coordinate system that is based on the image plane, pointing in the viewing direction of the camera.
The speed of the rocks can be obtained by measuring the distance they have moved between
adjacent frames.

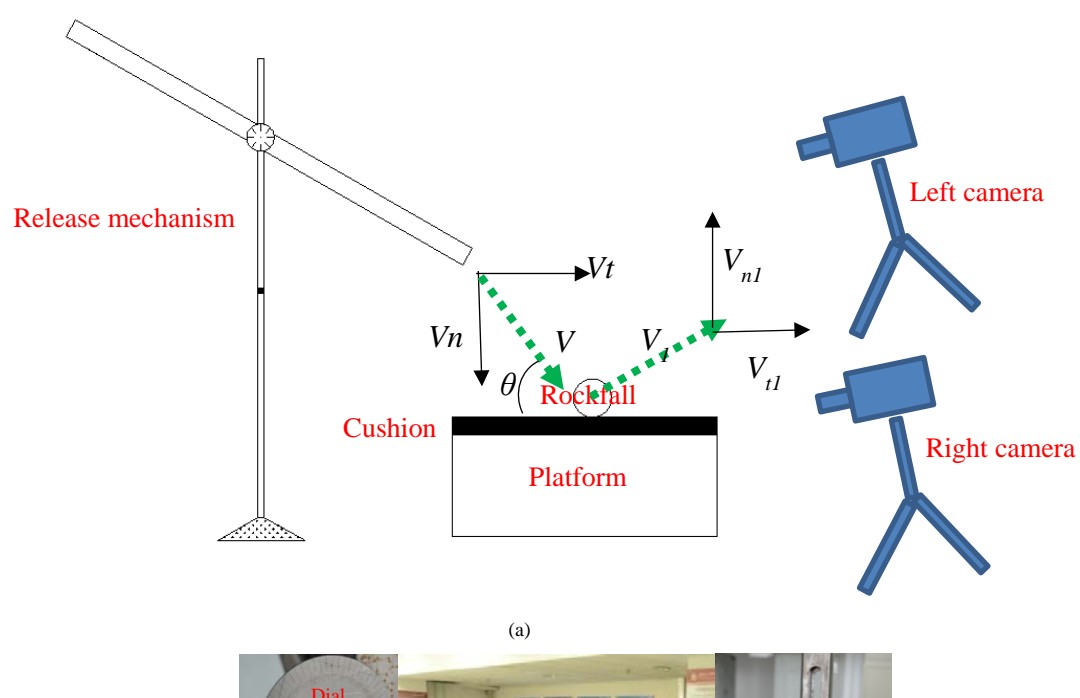

(a)

(b)

Fig.3 The experimental apparatus. (a) Model, (b) Laboratory

To simulate gravel cushions of different thicknesses, a large number of 40 cm length × 40
cm width × 2 cm height hollow gypsum boards were constructed. A 30 cm length × 30 cm width ×
2 cm height section was cut out of the center of each board. The hollow gypsum boards were
stacked on top of each other to simulate gravel cushions of different thickness, and then the hollow
parts of the boards were filled with gypsum particles. The hollow boards were fixed to a massive
40 cm length × 40 cm width × 6 cm height gypsum base to ensure the preservation of momentum
from the impact. In order to accurately measure the speed of the blocks with the cameras and to
avoid interference from the motion of cushion particles affected by the collision, the cushion was
blackened (Figure 4).

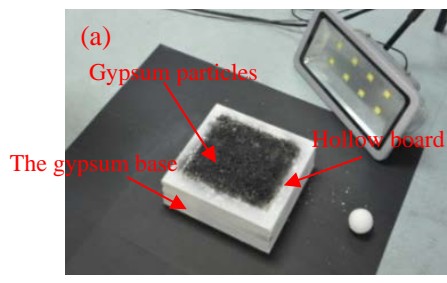 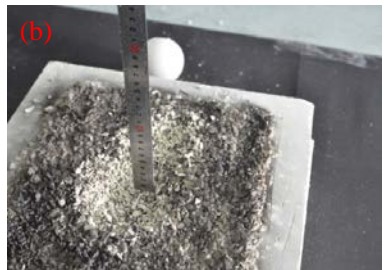


Fig. 4 Photographs of a cushion (a) before and (b) after a rock impact experiment
**3.2 Experimental procedure**
The main uncertainties in the test results arise in tests with large cushion particles, where the
wider scatter of the values is attributed to the contact configuration between the large cushion
particles and the blocks: large cushion particles have numerous different configurations. This also
affected the deviation in the trajectory caused by the impact, which had a drastically higher
uncertainty than for small cushion particles. In order to counteract the effects of chance, a "three
tests for the mean" method was adopted, and the average value was set as the final result given for
each data point in the figures and tables presented here. For cushion particle sizes of 18 mm and
24 mm, each test was repeated five times and the middle three values were used to obtain the
average value, while for cushion particle sizes of less than 18 mm, each test was conducted three
times. The obviously outlying results were the two rare conditions that $V_{COR}=0$ or $V_{COR} > 1$, if
these results were obtained, the tests were repeated to reduce the error.
The 2 cm, 3 cm, 4 cm, and 5 cm radius spherical blocks (Figure 2) were released from a
height of 1.2 m, and the effects of cushion thickness and particle size and of block volume on the
*COR* were studied. $V_{COR}$ for the *CORs* measured in the experiment was calculated using the
magnitudes of the incident and rebound velocities as in Equation (1). The block was inserted into
one side of the tube and, after sliding and rolling through the tube, collided with the collision
surface. The initial impact surface was the massive gypsum base to simulate the platform before
paving with a cushion in an open-pit mine. Paved tests were then performed using thicknesses of 2
cm, 4 cm, 6 cm, 8 cm, 10 cm, 12 cm, and 14 cm and cushion particle sizes of 2 mm, 6 mm, 10 mm,
14 mm, 18 mm, and 24 mm. Five iterations of 628 testing cases were carried out.
In order to investigate the effect of rockfall released from different movement heights on the
*COR* of the collision between rockfall and cushion, experiments were conducted in which blocks
of 2 cm, 3 cm, 4 cm, and 5 cm radius fell from 0.4 m, 0.8 m, 1.2 m, and 1.6 m to collide with an
8-cm thick cushion of different particle sizes. Four iterations of 352 testing cases were carried out.
Photographs of the cushion before and after a rock impact experiment are shown in Figure 4. The
cushion was always repaired completely after each impact experiment to ensure that the next
experiment was free from interference. If any particles had been knocked off the platform, new
particles were added to supplement the cushion, and the surface was blackened again before the
next impact experiment in order for the cameras to obtain accurate measurements of block speed.
**3.3 Experimental results and discussion**
3.3.1 Experimental results
The *COR* for blocks released from a height of 1.2 m to collide with an uncushioned plate is
shown in Table 1 and Figure 5.
Table 1 The *COR* of block collisions with the plate

| $H$=1.2m,$h$=0cm, $d$=0mm | $r$=2cm(Mean/Std dev) | $r$=3cm (Mean/Std dev) | $r$=4cm(Mean/Std dev) | $r$=5cm(Mean/Std dev) |
|---|---|---|---|---|
| | 0.384/0.032 | 0.421/0.020 | 0.437/0.048 | 0.444/0.036 |


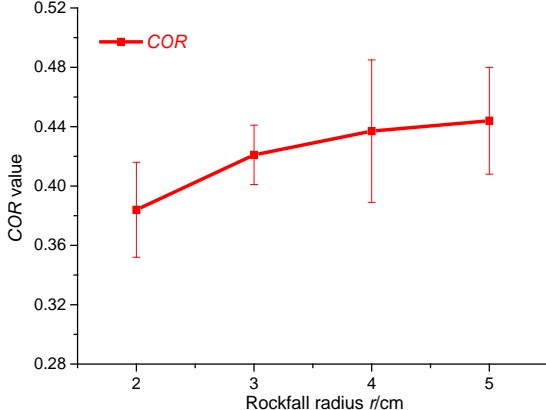


Fig. 5 The *COR* (Mean $\pm$ SD) of block collisions with the plate. (Error bars: one standard deviation)
*CORs* derived from experiments where rocks of different radii were released from a 1.2 m
movement height to collide with a plate paved with cushions of different thicknesses and particle
sizes are plotted in Table 2 and Figure 6. In Figure 6, mean values with error bars are shown for
each test.
Table 2 Experimental results for the first group of tests (movement height $H$=1.2 m)

| | $h$(cm) \ $d$(mm) | 2mm(Mean/Std dev) | 6mm(Mean/Std dev) | 10mm(Mean/Std dev) | 14mm(Mean/Std dev) | 18mm(Mean/Std dev) | 24mm(Mean/Std dev) |
|---|---|---|---|---|---|---|---|
| $r$=2cm | 2cm | 0.326/0.015 | 0.332/0.029 | 0.346/0.029 | 0.343/0.029 | 0.348/0.063 | 0.354/0.059 |
| | 4cm | 0.294/0.019 | 0.325/0.029 | 0.302/0.037 | 0.323/0.038 | 0.317/0.062 | 0.312/0.047 |
| | 6cm | 0.259/0.017 | 0.274/0.034 | 0.282/0.036 | 0.283/0.042 | 0.301/0.043 | 0.296/0.038 |
| | 8cm | 0.243/0.028 | 0.254/0.040 | 0.263/0.048 | 0.271/0.043 | 0.277/0.048 | 0.284/0.074 |
| | 10cm | 0.241/0.038 | 0.247/0.048 | 0.255/0.031 | 0.258/0.051 | 0.264/0.068 | 0.277/0.057 |
| | 12cm | 0.228/0.027 | 0.233/0.042 | 0.247/0.048 | 0.252/0.057 | 0.251/0.062 | 0.266/0.054 |
| | 14cm | 0.22/0.032 | 0.232/0.045 | 0.24/0.032 | 0.236/0.060 | 0.249/0.048 | 0.258/0.054 |
| | $h$(cm) \ $d$(mm) | 2mm(Mean/Std dev) | 6mm(Mean/Std dev) | 10mm(Mean/Std dev) | 14mm(Mean/Std dev) | 18mm(Mean/Std dev) | 24mm(Mean/Std dev) |
| $r$=3cm | 2cm | 0.334/0.019 | 0.341/0.013 | 0.347/0.036 | 0.354/0.050 | 0.352/0.030 | 0.368/0.046 |
| | 4cm | 0.302/0.036 | 0.315/0.042 | 0.316/0.044 | 0.327/0.049 | 0.326/0.036 | 0.334/0.065 |
| | 6cm | 0.277/0.025 | 0.284/0.024 | 0.288/0.033 | 0.318/0.039 | 0.309/0.053 | 0.325/0.072 |
| | 8cm | 0.247/0.026 | 0.262/0.046 | 0.267/0.040 | 0.273/0.055 | 0.281/0.054 | 0.292/0.031 |
| | 10cm | 0.237/0.027 | 0.246/0.027 | 0.254/0.031 | 0.262/0.045 | 0.257/0.049 | 0.268/0.051 |
| | 12cm | 0.226/0.035 | 0.239/0.045 | 0.242/0.019 | 0.248/0.041 | 0.255/0.035 | 0.259/0.042 |
| | 14cm | 0.218/0.053 | 0.224/0.027 | 0.229/0.044 | 0.231/0.054 | 0.246/0.055 | 0.262/0.044 |
| | $h$(cm) \ $d$(mm) | 2mm(Mean/Std dev) | 6mm(Mean/Std dev) | 10mm(Mean/Std dev) | 14mm(Mean/Std dev) | 18mm(Mean/Std dev) | 24mm(Mean/Std dev) |
| $r$=4cm | 2cm | 0.336/0.019 | 0.348/0.022 | 0.356/0.026 | 0.365/0.048 | 0.367/0.036 | 0.372/0.040 |
| | 4cm | 0.309/0.026 | 0.321/0.024 | 0.315/0.030 | 0.325/0.023 | 0.334/0.037 | 0.343/0.045 |
| | 6cm | 0.28/0.014 | 0.309/0.018 | 0.292/0.023 | 0.292/0.012 | 0.312/0.035 | 0.325/0.033 |
| | 8cm | 0.256/0.011 | 0.271/0.023 | 0.276/0.029 | 0.274/0.024 | 0.293/0.031 | 0.302/0.037 |
| | 10cm | 0.252/0.015 | 0.258/0.022 | 0.269/0.025 | 0.265/0.024 | 0.281/0.041 | 0.278/0.043 |
| | 12cm | 0.236/0.010 | 0.245/0.025 | 0.237/0.027 | 0.243/0.038 | 0.252/0.045 | 0.258/0.035 |
| | 14cm | 0.224/0.011 | 0.235/0.022 | 0.232/0.038 | 0.237/0.027 | 0.248/0.038 | 0.253/0.037 |
| | $h$(cm) \ $d$(mm) | 2mm(Mean/Std dev) | 6mm(Mean/Std dev) | 10mm(Mean/Std dev) | 14mm(Mean/Std dev) | 18mm(Mean/Std dev) | 24mm(Mean/Std dev) |
| $r$=5cm | 2cm | 0.34/0.014 | 0.342/0.022 | 0.356/0.035 | 0.368/0.028 | 0.371/0.032 | 0.38/0.036 |
| | 4cm | 0.324/0.013 | 0.311/0.017 | 0.323/0.030 | 0.344/0.028 | 0.343/0.037 | 0.352/0.023 |
| | 6cm | 0.291/0.009 | 0.292/0.021 | 0.318/0.015 | 0.309/0.025 | 0.326/0.047 | 0.33/0.046 |
| | 8cm | 0.265/0.013 | 0.28/0.012 | 0.288/0.025 | 0.293/0.027 | 0.302/0.050 | 0.313/0.043 |
| | 10cm | 0.263/0.017 | 0.265/0.029 | 0.269/0.028 | 0.272/0.024 | 0.271/0.040 | 0.288/0.043 |
| | 12cm | 0.24/0.012 | 0.243/0.027 | 0.252/0.036 | 0.257/0.028 | 0.259/0.046 | 0.266/0.060 |
| | 14cm | 0.22/0.015 | 0.23/0.027 | 0.237/0.012 | 0.242/0.028 | 0.234/0.045 | 0.254/0.034 |

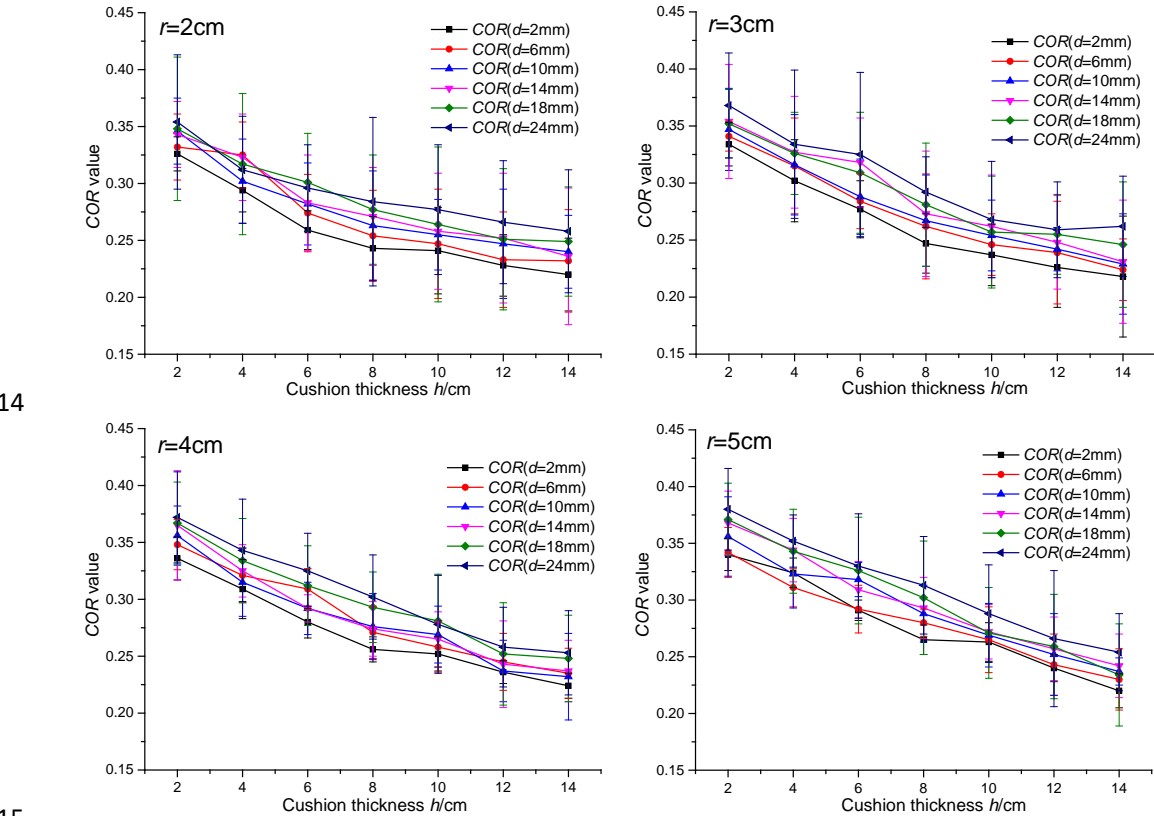


Fig.6 Comparison of the *COR* of blocks of different radii released from a height of 1.2m

*CORs* derived for rocks of different radii released from different movement heights to collide

with an 8-cm thick cushion of various particle sizes are plotted in Table 3 and Figure 7. As with
Figure 6, Figure 7 shows mean values with error bars for each test.

Table 3 Experimental results for the second group of tests (cushion thickness *h*=8 cm)

| | *H*(m)   *d*(mm) | 2mm(Mean/Std dev) | 6mm(Mean/Std dev) | 10mm(Mean/Std dev) | 14mm(Mean/Std dev) | 18mm(Mean/Std dev) | 24mm(Mean/Std dev) |
|---|---|---|---|---|---|---|---|
| *r*=2cm | 0.4m | 0.216/0.020 | 0.228/0.011 | 0.236/0.025 | 0.254/0.030 | 0.256/0.053 | 0.260/0.037 |
| | 0.8m | 0.229/0.009 | 0.234/0.030 | 0.245/0.027 | 0.243/0.029 | 0.262/0.037 | 0.267/0.053 |
| | 1.2m | 0.243/0.019 | 0.254/0.033 | 0.263/0.033 | 0.271/0.044 | 0.277/0.047 | 0.284/0.032 |
| | 1.6m | 0.243/0.013 | 0.252/0.018 | 0.271/0.042 | 0.290/0.047 | 0.283/0.036 | 0.282/0.051 |
| *r*=3cm | *H*(m)   *d*(mm) | 2mm(Mean/Std dev) | 6mm(Mean/Std dev) | 10mm(Mean/Std dev) | 14mm(Mean/Std dev) | 18mm(Mean/Std dev) | 24mm(Mean/Std dev) |
| | 0.4m | 0.224/0.015 | 0.231/0.022 | 0.243/0.023 | 0.252/0.037 | 0.265/0.042 | 0.268/0.055 |
| | 0.8m | 0.236/0.015 | 0.243/0.023 | 0.264/0.037 | 0.262/0.037 | 0.267/0.033 | 0.276/0.045 |
| | 1.2m | 0.247/0.020 | 0.262/0.020 | 0.267/0.032 | 0.273/0.046 | 0.281/0.041 | 0.292/0.044 |
| | 1.6m | 0.254/0.014 | 0.265/0.032 | 0.286/0.026 | 0.289/0.035 | 0.293/0.018 | 0.301/0.032 |
| *r*=4cm | *H*(m)   *d*(mm) | 2mm(Mean/Std dev) | 6mm(Mean/Std dev) | 10mm(Mean/Std dev) | 14mm(Mean/Std dev) | 18mm(Mean/Std dev) | 24mm(Mean/Std dev) |
| | 0.4m | 0.231/0.013 | 0.242/0.015 | 0.239/0.026 | 0.264/0.031 | 0.262/0.029 | 0.276/0.039 |
| | 0.8m | 0.245/0.021 | 0.257/0.012 | 0.262/0.029 | 0.287/0.028 | 0.286/0.039 | 0.290/0.055 |
| | 1.2m | 0.256/0.012 | 0.271/0.036 | 0.276/0.025 | 0.284/0.020 | 0.293/0.038 | 0.302/0.020 |
| | 1.6m | 0.261/0.020 | 0.285/0.018 | 0.286/0.034 | 0.299/0.054 | 0.311/0.041 | 0.310/0.050 |
| *r*=5cm | *H*(m)   *d*(mm) | 2mm(Mean/Std dev) | 6mm(Mean/Std dev) | 10mm(Mean/Std dev) | 14mm(Mean/Std dev) | 18mm(Mean/Std dev) | 24mm(Mean/Std dev) |
| | 0.4m | 0.236/0.010 | 0.253/0.014 | 0.25/0.036 | 0.263/0.033 | 0.276/0.045 | 0.284/0.036 |
| | 0.8m | 0.252/0.017 | 0.267/0.015 | 0.283/0.022 | 0.272/0.037 | 0.294/0.043 | 0.298/0.045 |
| | 1.2m | 0.265/0.011 | 0.28/0.037 | 0.288/0.030 | 0.293/0.049 | 0.302/0.038 | 0.313/0.045 |
| | 1.6m | 0.273/0.027 | 0.287/0.021 | 0.299/0.042 | 0.31/0.039 | 0.308/0.051 | 0.322/0.038 |

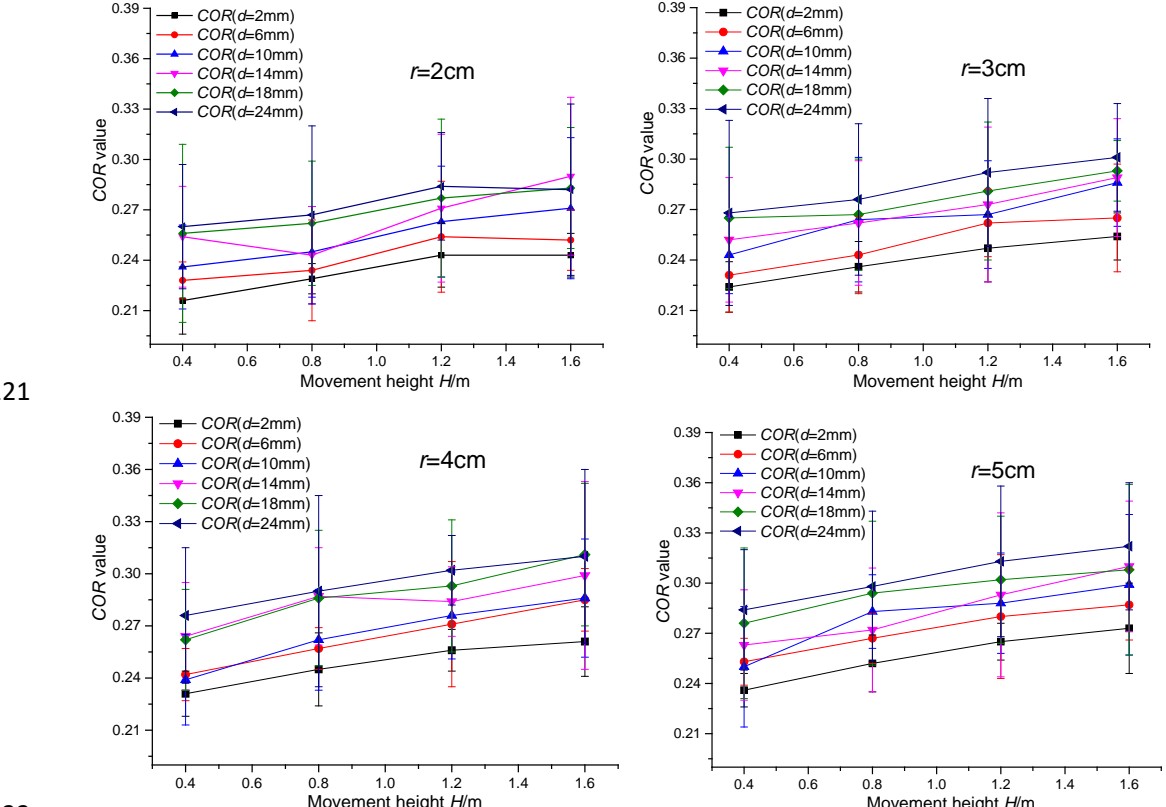


Fig.7 Comparison of the *COR* for blocks of different radii colliding with an 8-cm thick cushion
3.3.2 Discussion

The figures above indicate that cushion thickness and particle size have a strong influence on
the *COR* of collisions between a rockfall and a cushion, whereas the influence of rockfall block
radius is relatively weak. When the particle size of the cushion is small and its thickness is large,
the *COR* of the collision is small, and its effectiveness for energy-consumption is obvious. With an
increase in rockfall block radius and movement height, the impact energy increases dramatically
for rockfalls colliding with a cushion (Kawahara et al., 1998). Under low impact energy, changes
in cushion thickness have a relatively small effect on the *COR* of the collision, and even thin
cushions have a certain energy-absorbing effect, as verified by Pei (2016) and Kawahara (2006).
However, under high impact energy, the difference in energy-absorption of different thicknesses of
gravel cushion is marked. Because a thin cushion can be more easily compressed in a very short
time, the rockfall is more likely to be affected by the underlying platform at low cushion
thicknesses. This makes reducing the cushion thickness equivalent to increasing the effective
stiffness of the cushion, significantly limiting its buffering and energy-absorbing effect. When the
cushion thickness is relatively small, the *COR* increases significantly with a decrease in cushion
thickness. However, when the cushion's thickness is relatively large, this trend is no longer
obvious.

When a constant rockfall release height of 1.2 m is used, the *COR* is large where there is no
cushion and decreases significantly with an increase in cushion thickness. This agrees with the
observations of Kawahara (2005). However, when the cushion reaches a certain thickness, namely,
the ratio of the falling block radius, *r*, to the cushion thickness, *h*, is 1/4–1/3, the rate of reduction
in the *COR* with an increase in cushion thickness gradually decreases. *COR* is more sensitive to
the thickness of cushions with a small particle size than those with a relatively large particle size:
the range in *CORs* caused by thickness variation is wider for small cushion particle sizes, while, as

the thickness of cushions with a large particle size is increased, the *COR* of the collision between the rockfall and cushion changes relatively slightly.

If the cushion thickness is kept constant at 8 cm, as the movement height of the block increases the *COR* also increases, but when blocks of different radii collide with a cushion of the same thickness, the range in the *COR* of blocks with a large radius is larger than for blocks with a relatively small radius. When the blocks move from a relatively low height, the *COR* of the collision is more likely to be affected by the particle size compared to when blocks are released from a greater height. When the cushion particle size is large, the difference in collision configuration between the rockfall and cushion is more pronounced, resulting in a wide range in the *COR* of the collision.

# 4 Orthogonal test design

## 4.1 Orthogonal test procedure

To explore the degree of influence of cushion particle size and thickness on *COR* when a rockfall moves through the cushion, orthogonal test theory was adopted to design a test program (Tao et al., 2017). Orthogonal testing is a design method that allows the testing of multiple factors at multiple levels. It is based on orthogonality and selects representative points from a comprehensive experiment for testing so that fewer trials can fully reflect the impact of the variation of each factor on the index. When these factors cannot be considered in full, the leading factor is considered to achieve the expected effects to a great extent.

Four independent parameters, the rockfall block radius, *r*, movement height, *H*, cushion thickness, *h*, and particle size, *d*, were selected as the basic factors to test. The purpose of doing an orthogonal test was to explore the degree of influence of the four different factors on the *COR* and damage depth, *L*, and find the combination that will give the optimal protective effect when a rockfall collides with a cushion. The damage depth (*L*) is the depth to which the cushion is influenced after a rockfall has collided with it and can be used to represent the degree of damage to the cushion. As shown in Table 4, every factor has four levels:

Table 4 Factors and levels for the orthogonal test

| Factor level | Rockfall radius $r$/cm | Movement height $H$/m | Cushion thickness $h$/cm | Particle size $d$/mm |
|---|---|---|---|---|
| Level 1 | 2 | 0.4 | 2 | 2 |
| Level 2 | 3 | 0.8 | 4 | 6 |
| Level 3 | 4 | 1.2 | 6 | 10 |
| Level 4 | 5 | 1.6 | 8 | 14 |

In order to improve the accuracy of the test, and considering that all of the factors have four levels, the $L_{32}$ ($4^9$) arrangement factor was selected for the testing program. The damage depth, *L*, of the cushion and the *COR* of the rockfall-cushion collision are taken as test indices to explore the degree of influence of the four factors (Pichler et al., 2005).

As there is a high degree of randomness inherent in the rockfall motion, each case was tested three times and the mean value was taken as the final result, so as to improve the accuracy of the experiments. The test results are shown in Table 5.

Table 5 Orthogonal test results

| Test number | Rockfall radius $r$/cm | Movement height $H$/m | Cushion thickness $h$/cm | Particle size $d$/mm | Damage depth of cushion $L$/cm (Mean/Std dev) | $COR$ of collision between rockfall and cushion (Mean/Std dev) |
|---|---|---|---|---|---|---|
| 1 | 2 | 0.4 | 2 | 2 | 0.65/0.082 | 0.278/0.012 |
| 2 | 2 | 0.8 | 4 | 6 | 0.74/0.056 | 0.273/0.023 |
| 3 | 2 | 1.2 | 6 | 10 | 0.93/0.082 | 0.282/0.029 |
| 4 | 2 | 1.6 | 8 | 14 | 1.05/0.046 | 0.295/0.028 |
| 5 | 3 | 0.4 | 2 | 6 | 0.58/0.053 | 0.294/0.012 |
| 6 | 3 | 0.8 | 4 | 2 | 1.45/0.165 | 0.265/0.015 |
| 7 | 3 | 1.2 | 6 | 14 | 1.03/0.171 | 0.317/0.041 |
| 8 | 3 | 1.6 | 8 | 10 | 1.60/0.193 | 0.280/0.020 |
| 9 | 4 | 0.4 | 4 | 10 | 0.62/0.036 | 0.296/0.028 |
| 10 | 4 | 0.8 | 2 | 14 | 0.56/0.104 | 0.338/0.029 |
| 11 | 4 | 1.2 | 8 | 2 | 2.60/0.303 | 0.256/0.022 |
| 12 | 4 | 1.6 | 6 | 6 | 2.20/0.375 | 0.284/0.036 |
| 13 | 5 | 0.4 | 4 | 14 | 0.61/0.076 | 0.309/0.031 |
| 14 | 5 | 0.8 | 2 | 10 | 0.58/0.026 | 0.328/0.037 |
| 15 | 5 | 1.2 | 8 | 6 | 2.12/0.217 | 0.280/0.025 |
| 16 | 5 | 1.6 | 6 | 2 | 2.85/0.321 | 0.273/0.022 |
| 17 | 2 | 0.4 | 8 | 2 | 1.36/0.026 | 0.216/0.016 |
| 18 | 2 | 0.8 | 6 | 6 | 1.24/0.106 | 0.265/0.025 |
| 19 | 2 | 1.2 | 4 | 10 | 1.13/0.149 | 0.302/0.031 |
| 20 | 2 | 1.6 | 2 | 14 | 0.68/0.082 | 0.358/0.038 |
| 21 | 3 | 0.4 | 8 | 6 | 0.92/0.121 | 0.231/0.017 |
| 22 | 3 | 0.8 | 6 | 2 | 1.49/0.187 | 0.256/0.012 |
| 23 | 3 | 1.2 | 4 | 14 | 1.08/0.046 | 0.327/0.031 |
| 24 | 3 | 1.6 | 2 | 10 | 0.84/0.076 | 0.351/0.029 |
| 25 | 4 | 0.4 | 6 | 10 | 0.77/0.135 | 0.287/0.035 |
| 26 | 4 | 0.8 | 8 | 14 | 0.81/0.137 | 0.281/0.027 |
| 27 | 4 | 1.2 | 2 | 2 | 1.03/0.159 | 0.336/0.021 |
| 28 | 4 | 1.6 | 4 | 6 | 1.96/0.115 | 0.318/0.030 |
| 29 | 5 | 0.4 | 6 | 14 | 0.67/0.044 | 0.292/0.019 |
| 30 | 5 | 0.8 | 8 | 10 | 1.05/0.092 | 0.275/0.078 |
| 31 | 5 | 1.2 | 2 | 6 | 1.14/0.098 | 0.347/0.025 |
| 32 | 5 | 1.6 | 4 | 2 | 2.54/0.184 | 0.294/0.027 |

**4.2 Optimization analysis and discussion of test results**

4.2.1 Optimization analysis method (flow)

The method of analysis used to optimize the calculation results and the optimization process is shown in Figure 8.

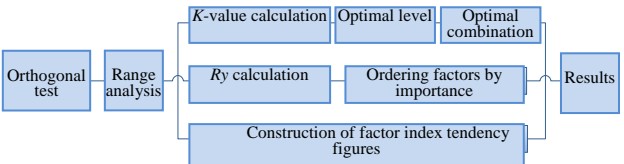


Fig.8 Flow chart for the optimization analysis of the test. $R_y$ is the range in factor $y$. The $K$ value is the sum of the
statistical test results.
The four parameters, rockfall block radius, $r$, movement height, $H$, cushion thickness, $h$, and
particle size, $d$, belong to the factor set $x \in$ (A, B, C, D), and the number of levels for all factors is
four. The statistical test parameter of factor set $x$ at level $y$ can be calculated by determining $K_{xy}$
($x$=A, B, C, D; $y$=1, 2, 3, 4), i.e., the sum of all the test result indices $P_{xy}$ containing level $y$ of
factor $x$, and dividing it by the total number of levels to obtain the average value $k_{xy}$ in which $P_{xy}$ is
the random variable of the normal distribution:
$$,k_{xy} = \frac{K_{xy}}{N_y} = \frac{\sum P_{xy}}{N_y}, \quad (6)$$
where $K_{xy}$ is the statistical parameter of factor $x$ at level $y$, $k_{xy}$ is the average value of $K_{xy}$, and $N_y$ is
the number of levels.
$k_{xy}$ can be used to judge the optimal level and combination of each factor. If a more optimal
result is obtained at a higher index value, then the level that increases the index value should be
selected, i.e., the level with maximum values for all factors $k_{xy}$; conversely, if the smaller the index
value is, the more optimal it is, the level with minimum values for all factors $k_{xy}$ should be selected.
The combination of parameters corresponding to an optimal level of all factors is the optimal
parameter combination. $R_y$ reflects the amount of variation of the test index with fluctuation in
factor level $y$. The larger $R_y$ is, the more sensitive the factor is to the influence of the test index.
The order of importance of the factors can be judged using $R_y$, and the optimal level and
combination of factor $x$ can be judged from $k_{xy}$.
4.2.2 Results of analysis and discussion
Range analysis was used to analyze the orthogonal test results in Table 5. This uses the
damage depth, $L$, of the cushion and the *COR* of the rockfall-cushion collision (Table 6) as
influencing factors to determine the optimum combination of rockfall block radius, $r$, movement
height, $H$, cushion thickness, $h$, and particle size, $d$, for the reduction of *COR*.
Table 6 Range analysis of two influencing factors for all evaluation indices

| Evaluation index | Levels | Rockfall radius $r$/cm | Movement height $H$/m | Cushions thickness $h$/cm | Particle size $d$/mm |
|---|---|---|---|---|---|
| *COR* of collision between rockfall and cushion | $k_{x1}$ | 0.285 | 0.271 | 0.325 | 0.270 |
| | $k_{x2}$ | 0.288 | 0.287 | 0.296 | 0.285 |
| | $k_{x3}$ | 0.298 | 0.305 | 0.281 | 0.301 |
| | $k_{x4}$ | 0.299 | 0.306 | 0.267 | 0.313 |
| | $R_y$ | 0.014 | 0.035 | 0.058 | 0.043 |
| Damage depth of cushion $L$ | $k_{x1}$ | 0.97 | 0.78 | 0.76 | 1.75 |
| | $k_{x2}$ | 1.12 | 0.99 | 1.26 | 1.35 |
| | $k_{x3}$ | 1.33 | 1.38 | 1.40 | 0.94 |
| | $k_{x4}$ | 1.44 | 1.72 | 1.44 | 0.81 |
| | $R_y$ | 0.47 | 0.94 | 0.68 | 0.94 |

The following conclusions can be drawn from Table 6:
(1) The degree of influence of the fours factors on the *COR* of the rockfall-cushion collision
is: cushion thickness (*h*) > particle size (*d*) > movement height *(H)* > block radius (*r*);
(2) The degree of influence of the four factors on the damage depth, *L*, of the cushion is:
movement height (*H*) = particle size (*d*) > cushion thickness (*h*) > block radius (*r*).
*E-I* tendency figures (Tao et al., 2017) are used to further explore the effects of each factor on
the test indices. The level of all factors is the *X*-coordinate (*E*), and the average value of the test
index is the *Y*-coordinate (*I*). The *E-I* tendency plots, Figure 9 and Figure 10, intuitively reflect the
tendency of the test index with a change in factor level and can point the way to further testing.
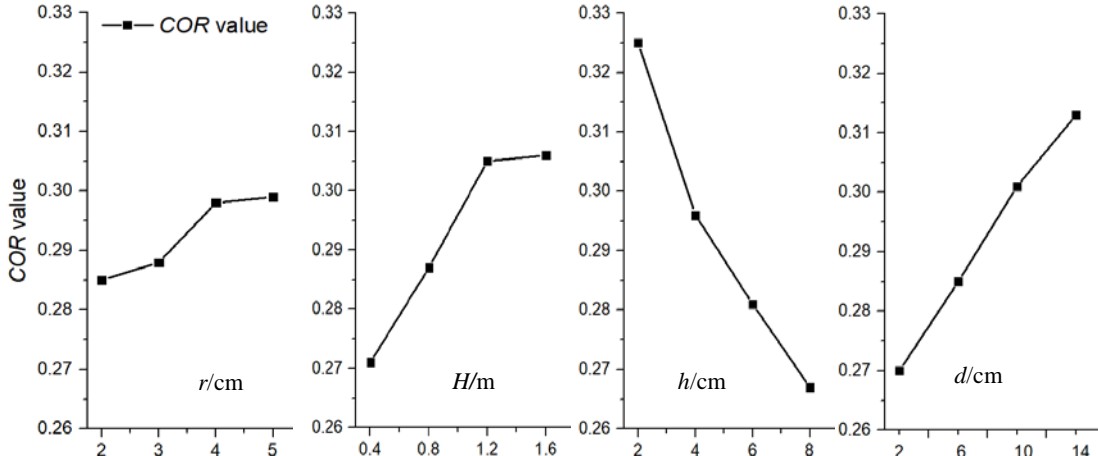

Fig.9 Tendency of each factor as regards the *COR* of the rockfall-cushion collision

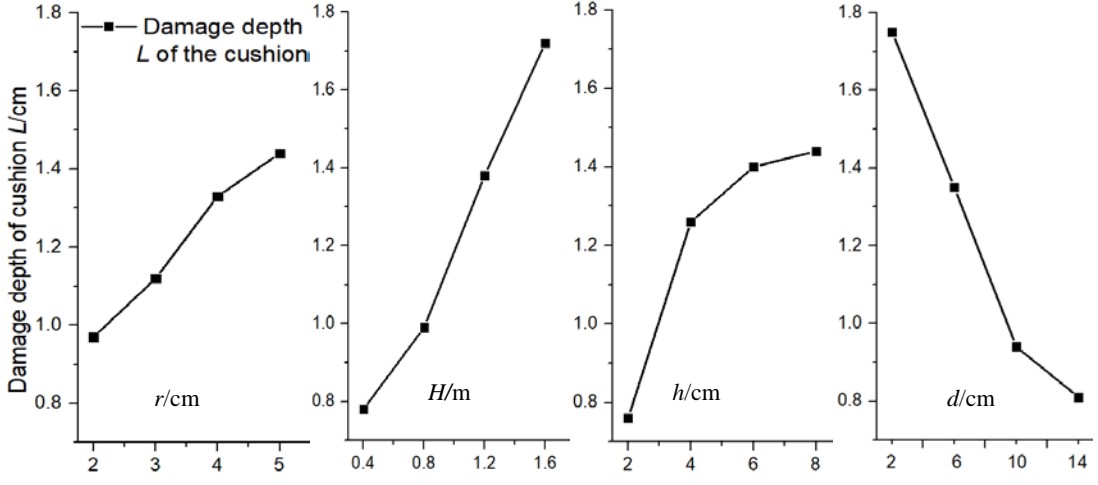

Fig.10 Tendency of each factor as regards damage depth *L* of the cushion

The following conclusions can be derived from Figures 9 and 10:
(1) The smallest optimal combination of parameters of the *COR* of the rockfall-cushion
collision is A1B1C4D1; that is, when *r*=2 cm, *H*=0.4 m, *h*=8 cm, and *d*=2 mm, the *COR* of the
collision is smallest (Figure 9).
(2) The shallowest optimal combination of parameters of the damage depth, *L,* of the cushion
is A1B1C1D4; that is, when *r*=2 cm, *H*=0.4 m, *h*=2 cm, and *d*=14 mm, the damage depth, *L*, of the
cushion is the shallowest (Figure 10).
To sum up, the cushion thickness, *h*, has the most significant influence on the *COR* of the
rockfall-cushion collision, while it has a relatively minor effect on the damage depth, *L*, of the
cushion. The second most important factor is particle size, *d*, it also can effectively affect the *COR*,
but the cushion can easily be destroyed when a rockfall with high kinetic energy collides with a

cushion of small particle size. The degree of influence of the rockfall block radius, *r*, on the two indices is far less than that of the other factors. When a gravel cushion is used to control rockfall down a slope, both the effectiveness with which it controls the rockfall and its durability are taken into account (Pichler et al., 2005) so the cushion thickness, *h*, should be the primary consideration in cushion design. The optimal thickness is 3–4 times the radius of the majority of the rockfall blocks. The smaller the particle size is, the smaller the *COR* is, but the cushion is also more likely to be destroyed. Therefore the appropriate particle size must be determined by combining the expected block size and drop height of the rockfall so that the cushion not only achieves the effect of reducing *COR* but also maintains its stability.

# 5 Conclusions

The buffering and energy-dissipation mechanism of gravel cushions with different properties under different impact energies were studied in laboratory collision tests, leading to the following conclusions:

1. Unlike conventional protection measures, a gravel cushion makes full use of waste mullock produced in the process of mine extension, which can be conveniently broken up into particles of the appropriate size. This can not only reduce the costs of reducing rockfall hazard and of mullock transportation and relieve overloading of the mine's dump but can also achieve better control of rockfalls, realizing the goal of "stone conquers stone."

2. In a series of laboratory tests, blocks of different radii were dropped from different heights onto different cushion materials. The results indicate that, for a given impact energy, the cushion thickness, *h*, has a strong influence on the measured coefficient of restitution (*COR*) and therefore impact pressure. From the point where the ratio of the falling block radius, *r*, to the cushion thickness, *h*, is 1/4–1/3, the rate of reduction in the *COR* with an increase in cushion thickness gradually decreases. When the blocks move from a relatively low height, the *COR* of the rockfall-cushion collision is more likely to be affected by the particle size than when blocks are released from a greater height. Therefore, in the process of cushion design, the estimated physical properties and drop height of the potentially dangerous rock should be investigated to estimate the impact energy of the rockfall.

3. Through an orthogonal test, it is found that the cushion thickness, *h*, has the most significant influence on the *COR* of the rockfall-cushion collision. The second most important factor is particle size, *d*, with a smaller particle size leading to a smaller *COR*. However, the cushion can easily be destroyed when a rockfall with high kinetic energy collides with a small particle size cushion. Therefore, cushion design should take structural reliability as well as effectiveness and any economic constraints into account. The appropriate particle size must be determined on the basis of the block size and drop height of the expected rockfall so that the cushion can not only achieve the effect of reducing *COR* but also maintain its stability.

4. Until now, it has not been possible to dictate a universal rule that the majority of engineering personnel can follow in the design of gravel cushions for a platform. This is a troubling blind spot. However, this work shows that, as well as increasing the cushion thickness, changing its particle size can improve the rockfall-controlling effect, and that the optimal particle size can be determined on the basis of the expected block size and drop height of the rockfall. This provides a widely applicable theoretical and practical basis for cushion design for open-pit mine

rockfall protection.

## Acknowledgements

This work is supported by the National Natural Science Foundation Item of China (No. 41502323), Beijing Natural Science Foundation of China (8142032) and China Postdoctoral Science Foundation Funded Project (No. 2017M621212).

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
