# Peer review of "The effects of gravel cushion particle size and thickness on the coefficient of restitution in rockfall impacts"

_Natural Hazards and Earth System Sciences, 2018_

## Referee Comment (RC1) · Anonymous Referee #1 · 20 Feb 2018

The research topic of this paper has a strong background in the mining industry as gravel cushion is widely used in open-pit mine for the mitigation of rockfall hazards. When a rock collides with gravel cushion, i.e. a pile of small granules, the kinetic energy of the rock will be absorbed, leading to a smaller run-out zone for the falling rock. To this aspect, the authors performed in-house experiments and investigated the effect of cushion's composition (particle size and thickness) on coefficient of restitution under rockfall impacts. In particular, the energy consumption and the buffer mechanisms of gravel cushion of various compositions were studied using differing impact conditions (i.e. the size of rock and its initial release height were altered). It was found that the thickness of cushion has a significant influence on the reduction of rock energy.

Therefore, this parameter should be carefully considered as a dominating factor for the design of cushion in the mining applications. The paper addressed a seemingly simple but yet important problem in the field of coal mining/rockfall. However, the research results would be more valuable if the following issues can be further clarified (major revision): (1) The English of this paper is not optimal, which prevents a clear understand of the experimental procedures and the physical significance behind observations. The format of the paper needs to be better arranged (e.g. sometimes the figure and the caption are not on the same page). There are also many typo-errors. It is STRONGLY suggested that the authors try to improve the language of the paper through the corrections either from a native English speaker or scientific proofreading & editing services. (2) Section of 'Introduction': It is ideal to include more newly published researches to highlight the importance and uniqueness of the current study. (3) Section of 'Coefficient of restitution': It is better to use the absolute values of the velocities in Equations (1) and (2) since one notices that velocity has a direction. Also it is not clear in e.g. Figures 7-9 that the CORs are calculated based on the velocities of normal direction or tangential direction, or they are calculated based on the kinetic energy? It is necessary to give a clear definition of the normal and tangential directions for a rock-cushion impact. Also the expression 'the COR of cushion' does not seem to be correct since COR is not a physical property of cushion. (4) Section of 'Experimental studies': It should be 'radius' in 'Spherical blocks with diameters of . . .'. It is better to give more details about the experimental procedures. For example, how the rock velocities are calculated from the frames (any calibration or correction of the view distortion)? What is the relative position of the two cameras? If only the vertical velocity of rock is measured, does that mean the view axis of the camera is parallel to the cushion platform? The paragraph below Figure 5 describes how to prepare the cushion platform. However, it is not easy to understand how the platform is established due to the poor English expression. It is suggested that the authors list clearly the experimental parameters in a table so that one sees clearly how the two groups of tests were performed. From Figure 5a it seems that rock has a tangential velocity when it impacts with the cushion platform. However,
the authors used the release height (by changing the inclination of the release path?) as a reference parameter, which influences both the normal and the tangential velocity of rock at impact. Thus it is not clear how the COR is affected by the distribution of kinematic energy between the normal and tangential directions (or the COR is calculated only based on the normal direction?). One wonders whether there still exists boundary effect in experiments, although the authors have tested spherical rocks and cushion platforms of different sizes (or thicknesses). It requires to clearly show that the current results are boundary-effect free. In Figures 8 and 9 the radius of rock and the diameters of particles do not have units. How many experiments are performed for each data points in these figures? One needs to show the uncertainties. How the packing structure of the cushion particles will influence the COR result? This is a topic which is worth discussing because one notices in Figure 4 that the geometry of particles is very regular. Do they form special interlocking structure in the platform? If so how does it affect the rock's impact-rebound behavior? How is the cushion prepared once again after one impact experiment so that the influence of particle packing structure on the next experiment is minimized? It would be great if the author could show the photos of cushion before and after the rock impact experiments. In the discussion part the authors mentioned 'Because the small thickness cushion can be compressed in a very short time . . .'. It needs to be clearly shown or demonstrated. The last paragraph of this section is very difficult to understand. It may be helpful if the important discussions are listed into bullet points so that one gets the ideas more quickly. (5) Section of 'Orthogonal test design': The definition of 'damage depth' should be clearly given. The principle of 'orthogonal test' should be given for the reader who is not familiar with the concept. What is the purpose of doing this test? In Tables 2 and 3 not only the mean value but also the uncertainties should be given. (6) Section of 'Conclusions': It is interesting to see some comments from the authors on non-spherical rocks. In nature the shape of rock is always non-spherical or polyhedral. In addition, treating rocks as non-spherical bodies is nowadays already 'standard' for rockfall simulations such as in RAMMS::ROCKFALL (Leine et al., Simulation of rockfall trajectories with consideration

of rock shape, Multibody System Dynamics, 2014). From the authors' point of view, how will rock geometry influence the conclusions obtained in this work?

---

## Referee Comment (RC2) · Anonymous Referee #2 · 1 Mar 2018

Main goal of the paper is to evaluate directions for cushion particle size and thickness when used in open-pit mining as energy dissipating and thus safety mechanism. The aim of the work tries to draw conclusions from laboratory tests to a universally applicable rule for cushioning design. As such, the investigated topic is clearly of interest for the general NHESS readership. The test procedure is such that a leading parameter in cushioning design should be evaluated via a so-called orthogonal test design, which is using differing impact conditions, altering rock size, release height, cushion particle size). Main conclusion is that the cushion thickness is the leading parameter.

However, the presented work needs major revisions in several aspects: Primarily, the

experimental findings need to be printed alongside with their error bars. Without the given uncertainties, it is not obvious whether the drawn conclusions can be labelled as significant or something be called as leading parameter. It is of key importance for the authors as for the reader likewise to be able to judge the results against their experimental uncertainties. From the description of the experimental work, it seems that the only measure to mitigate statistical outliers is, that each series result is the mean of an experimental triplet. It has to be shown that this procedure is sufficient to generate statistically significant data. If error bars should even out all recorded differences, the experimental method has to be improved.

Secondly, the text requires substantial refurbishments with respect to language. The authors should invest in clearer structure when describing the experimental setup as well as the testing procedure. Furthermore, figures should be labelled correctly and descriptive in order to facilitate the reading. Additionally, the measurement units should be consistent within the legends and the text (mm and cm, etc.). The use of proofing tools and the revision by a native speaker is highly advised to make the text more readable.

A few technical comments: The Introduction should be shortened, since the paper clearly focuses on open-pit mining questions and an overview and listing of references for general rockfall mitigation measures is not needed. References for the approximate formula for the total energy of rockfall should be backed up with a better accessible source. The experimental setup needs to be clarified, especially the positioning and use of the cameras (field of view, image processing, etc.). The concept of an orthogonal test theory should be explained and/or backed with a better accessible source. Is it just the altering of the four parameters of interest? The drawn conclusions are not written in a concise manner. Focusing on the main experimental result in a clear way would be favorable.

---

## Author Comment (AC1) · 17 Mar 2018

Answer to referee 1 comments Reviewer 1 R 1: The English of this paper is not optimal, which prevents a clear understand of the experimental procedures and the physical significance behind observations. The format of the paper needs to be better arranged (e.g. sometimes the figure and the caption are not on the same page). There are also many typo-errors. It is STRONGLY suggested that the authors try to improve the language of the paper through the corrections either from a native English speaker or scientific proof reading  editing services.

AC: Thanks for the reviewer's suggestion.  My manuscript has been edited for En-

glish by using an English editing service, the embellishment proof is supplied as the attachment. The format of this paper has been adjusted to make readers easier to understand.

R 2: Section of 'Introduction': It is ideal to include more newly published researches to highlight the importance and uniqueness of the current study.

AC: Thanks for the reviewer's suggestion. I have introduced some newly important references in the 'Introduction' section to highlight the importance and uniqueness of the current study, the newly references are listed as follows.

[1] Howald et al. (2017) evaluated the protective capacity of existing and newly proposed protection measures, and considered the possible reclassification of hazard as a function of the mitigation role played by the measure.

EP Howald, JM Abbruzzese, C Grisanti. An approach for evaluating the role of protection measures in rockfall hazard zoning based on the Swiss experience. Natural Hazards Earth System Sciences, 2017,17(7):1127-1144.

[2] Mignelli (2014), meanwhile, applied a rockfall risk management approach to the road infrastructure network of the Regione Autonoma Valle D'Aosta in order to calculate the level of risk and the potential for its reduction by rockfall protection devices. A comparative analysis of road accidents in the Aosta Valley was then undertaken to verify the methodology.

C Mignelli, D Peila, SL Russo, et al. Analysis of rockfall risk on mountainside roads: evaluation of the effect of protection devices. Natural Hazards, 2014,73(1):23-35.

[3] The effect of shape has been examined by performing tests with spherical and cubic blocks, finding that spherical blocks show higher and more consistent COR values than cubic blocks (Asteriou et al., 2016).

Pavlos Asteriou, George Tsiambaos. Empirical Model for Predicting Rockfall Trajectory Direction. Rock Mech Rock Eng, 2016,49:927-941.

[4] State-of-the-art simulation techniques incorporating nonsmooth contact dynamics and multibody dynamics have been applied to and adapted for the efficient simulation of rockfall trajectories, and the influence of rock geometry on rockfall dynamics has been studied through numerical simulation (Leineet al., 2014).

RI Leine, A Schweizer, M Christen, et al. Simulation of rockfall trajectories with consideration of rock shape. Multibody System Dynamics, 2014,32(2):241-271.

[5] Semi-rigid rockfall protection barriers have been installed along areas threatened by rockfall events, and numerical investigation of semi-rigid rockfall protection barriers has been carried out to obtain essential structural information such as the energy-absorption capacity of such barriers (Miranda et al., 2015)

SD Miranda, C Gentilini, G Gottardi, et al. Virtual testing of existing semi-rigid rockfall protection barriers. Engineering Structures, 2015, 85:83-94.

[6] Lambert et al. (2014) conducted real-scale impact experiments with impact energies ranging from 200 to 2200 kJ. They studied the response of rockfall protection embankments composed of a 4-m high cellular wall when exposed to a rock impact and compared this with previous real-scale experiments on other types of embankment. S Lambert, A Heymann, P Gotteland, et al. Real-scale investigation of the kinematic response of a rockfall protection embankment. Natural Hazards Earth System Sciences, 2014,14(5):1269-1281.

[7] Sun et al. (2016) used a tire cushion layer to absorb rockfall impact, utilizing the radial deformation of the tire. They built a reinforced concrete structure model with a tire cushion layer and carried out artificial rockfall tests.

J Sun, Z Chu, Y Liu, et al. Performance of Used Tire Cushion Layer under Rockfall Impact. Shock and Vibration, 2016, 2016 (10):1-10.

R 3: Section of 'Coefficient of restitution': It is better to use the absolute values of the velocities in Equations (1) and (2) since one notices that velocity has a direction.

Also it is not clear in e.g. Figures 7-9 that the CORs are calculated based on the velocities of normal direction or tangential direction, or they are calculated based on the kinetic energy? It is necessary to give a clear defi̧nition of the normal and tangential directions for a rock-cushion impact. Also the expression 'the COR of cushion' does not seem to be correct since COR is not a physical property of cushion.

AC: Thanks for the reviewer's suggestion. I agree with the suggestion that using the absolute values of velocities in Equations (1) and (2) since velocity has a direction. I have supplemented in 'Experimental procedure' section that VCOR for the CORs measured in Figures 7-9 was calculated using the magnitudes of the incident and rebound velocities, and the calculation method of VCOR is shown in Equation (1). They are not calculated based on the velocities of normal, tangential direction or kinetic energy.

I agree with the point that the expression 'the COR of cushion' is not correct since COR is not a physical property of cushion, so I have adopted the expression of 'the COR of collision between rockfall and cushion' to substitute for 'the COR of cushion' throughout.

R 4: (1) Section of 'Experimental studies': It should be 'radius' in 'Spherical blocks with diameters of ...'. It is better to give more details about the experimental procedures. For example, how the rock velocities are calculated from the frames (any calibration or correction of the view distortion)? What is the relative position of the two cameras? If only the vertical velocity of rock is measured, does that mean the view axis of the camera is parallel to the cushion platform?

AC: Thanks for the reviewer's suggestion. I have detailed about the experimental procedures. Such as: the calculation of rock velocities, the relative position of the two cameras and the main uncertainties in the test results, the descriptions are as follows: The two cameras, which obtained the motion, velocity, and kinetic energy automatically, were placed symmetrically at a distance of approximately 0.9m from the impact surface (Figure 5). The distance between the two cameras was about 1.2m, making

the cameras look down slightly at the targeted platform. The synchronized recordings from the two cameras captured a sequence of image stereopairs at time intervals of 1/200 s. By applying stereo-photogrammetric processing, the position of any point in both images can be computed in 3D space. In general, a digital image is a perspective projection of 3D space to the camera lenses. The image plane has a 2D coordinate system where position measurements can be made using pixel coordinates. The camera has a 3D reference coordinate system that is based on the image plane pointing in the viewing direction of the camera. The speed of the rocks can be obtained by measuring the distance they have moved between adjacent frames. Therefore, if only the vertical velocity of rock is measured, I think it doesn't mean that the view axis of the camera is parallel to the cushion platform.

R4(2) The paragraph below Figure 5 describes how to prepare the cushion platform. However, it is not easy to understand how the platform is established due to the poor English expression. It is suggested that the authors list clearly the experimental parameters in a table so that one sees clearly how the two groups of tests were performed.

Thanks for the reviewer's suggestion. My manuscript has been edited for English by using an English editing service and I have detailed about the experimental procedures, the preparation process of the cushion platform is described as follows: To simulate gravel cushions of different thicknesses, a large number of 40 cm length $\times$ 40 cm width $\times$ 2 cm height hollow gypsum boards were made. A 30cm length $\times$ 30cm width $\times$ 2cm height section was cut out of the center of each board. The hollow gypsum boards were stacked on top of each other to simulate gravel cushions of different thickness, and then the hollow parts of the boards were filled with gypsum particles. The hollow boards were fixed to a massive 40cm length $\times$ 40cm width $\times$ 6cm height gypsum base to ensure the preservation of momentum from the impact. As is shown in Tables 1 and 2 (See Supplement), because the space occupied by tables of two groups experimental parameters is so large, thus I think choose the table to express how the two groups of tests were performed is not relatively suitable in manuscript. Therefore, I have detailed the experiment procedure and Figures of experiment result, the experiment content has also been edited for English by using an English editing service to facilitate reader to understand.

R4(3) From Figure 5a it seems that rock has a tangential velocity when it impacts with the cushion platform. However, the authors used the release height (by changing the inclination of the release path?) as a reference parameter, which influences both the normal and the tangential velocity of rock at impact. Thus it is not clear how the COR is affected by the distribution of kinematic energy between the normal and tangential directions (or the COR is calculated only based on the normal direction?).

Thanks for the reviewer's suggestion. I used the release height as a reference parameter, but I change the vertical height between the released position and the ground instead of changing the inclination of the release path. I think VCOR are more representative to reflect the motion situation of rockfall before and after colliding with the cushion compared with other forms of COR, thus the CORs measured in tests are VCOR calculated based on the velocity magnitude of the incident and rebound stage, the calculation method is shown in Equation (1).

R4(4) One wonders whether there still exists boundary effect in experiments, although the authors have tested spherical rocks and cushion platforms of different sizes (or thicknesses). It requires to clearly show that the current results are boundary-effect free.

Thanks for the reviewer's suggestion. I think there is a slight boundary effect in experiments, but which can basically be neglected. Due to the restriction of laboratory test, the coverage area of cushion is not so wide, but the proportion of the size of rockfall and the coverage area of cushion is relatively small in the experiment, and the impact range when rockfall collided with the cushion is far less than the coverage area of the cushion (See Figure 7 in Supplement). Therefore the current results are basically boundary-effect free.

R4(5) In Figures 8 and 9 the radius of rock and the diameters of particles do not have units. How many experiments are performed for each data points in these figures?

Thanks for the reviewer's suggestion. I have added units in Figures 8 and 9. The main uncertainties in the test results arise in tests with large cushion particles, where the wider scatter of the values is attributed to the contact configuration between the large cushion particles and the blocks: large cushion particles have numerous different configurations. This also affected the deviation in the trajectory caused by the impact, which had a drastically higher uncertainty than for small cushion particles. In order to counteract the effects of chance, a "three tests for the mean" method was adopted, and the average value was set as the final result for each data point in the figures and tables presented here. For cushion particle sizes of 1.8cm and 2.4cm, each test was repeated five times, and the middle three values were used to obtain the average value, while for cushion particle sizes of less than 1.8 cm, each test was conducted three times. If an obviously outlying result was obtained, the test was repeated to reduce the error. I have mentioned it in 'Experimental procedure' section.

R4(6) One needs to show the uncertainties. How the packing structure of the cushion particles will influence the COR result? This is a topic which is worth discussing because one notices in Figure 4 that the geometry of particles is very regular. Do they form special interlocking structure in the platform? If so how does it affect the rock's impact-rebound behavior? How is the cushion prepared once again after one impact experiment so that the influence of particle packing structure on the next experiment is minimized? It would be great if the author could show the photos of cushion before and after the rock impact experiments. In the discussion part the authors mentioned 'Because the small thickness cushion can be compressed in a very short time ...'. It needs to be clearly shown or demonstrated. The last paragraph of this section is very difficult to understand. It may be helpful if the important discussions are listed into bullet points so that one gets the ideas more quickly.

Thanks for the reviewer's suggestion. The main uncertainties in the test results arise in

tests with large cushion particles, where the wider scatter of the values is attributed to the contact conifi̧guration between the large cushion particles and the blocks: large cushion particles have numerous different conifi̧gurations. This also affected the deviation in the trajectory caused by the impact, which had a drastically higher uncertainty than for small cushion particles. I think the packing structure of the cushion particles don't have the influence on the COR result. Because the proportion of the size of rockfall and the coverage area of cushion is relatively small in the experiment, and the impact range when rockfall collided with the cushion is less than the coverage area of the cushion (See Figure 7 in Supplement). In field engineering, when the rockfall collide with the cushion, the gravel particles cushion of other areas will also generate interlocking effect on the particles in the collision area, which is similar to the packing structure in the tests. Photographs of the cushion before and after a rock impact experiment are shown in Figure 7 (See supplement). The cushion was always repaired completely after each impact experiment to ensure that the next experiment was free from interference. If any particles had collided out from the platform, new particles were added to supplement the cushion, and the surface was blackened again before the next impact experiment in order for the cameras to obtain accurate measurements of block speed. Fig. 7 Photographs of a cushion (a) before and (b) after a rock impact experiment I have revised and reorganized the 'Discussion' part by using an English editing service. Such as, I have adopt 'Because a thin cushion can be more easily compressed in a very short time' to substitute for 'Because the small thickness cushion can be compressed in a very short time.

R 5: Section of 'Orthogonal test design': The deifi̧nition of 'damage depth' should be clearly given. The principle of 'orthogonal test' should be given for the reader who is not familiar with the concept. What is the purpose of doing this test? In Tables 2 and 3 not only the mean value but also the uncertainties should be given.

AC: Thanks for the reviewer's suggestion. I have introduced the clear definition of 'damage depth' and the principle of 'orthogonal test' in 'Orthogonal test design' section

to facilitate readers to understand. The definition of 'damage depth (L)' is the depth to which the cushion is influenced after a rockfall has collided with it and can be used to represent the degree of damage to the cushion. The principle of 'orthogonal test' is described as follows: Orthogonal testing is a design method that allows testing of multiple factors and multiple levels. It is based on orthogonality and selects representative points from a comprehensive experiment for testing. The orthogonal test method has the advantages of being uniformly dispersed, neat and comparable, making each test highly representative so that fewer trials can fully reflect the impact of the variation of each factor on the index. The purpose of doing an orthogonal test is to explore the degree of influence of the four different factors on the COR and damage depth, L, and find the best combination to reach the optimal protective effect when a rockfall collides with a cushion. When these factors cannot be considered in full, the leading factor is considered to achieve the expected effects to a great extent. As there is a high degree of randomness inherent in the rockfall motion, each case was tested three times and the mean value was taken as the final result, so as to improve the accuracy of the experiments. The test results including the uncertainties are shown in Table 3 (See Supplement).

R 6: Section of 'Conclusions': It is interesting to see some comments from the authors on non-spherical rocks. In nature the shape of rock is always non-spherical or polyhedral. In addition, treating rocks as non-spherical bodies is nowadays already 'standard' for rockfall simulations such as in RAMMS::ROCKFALL (Leine et al., Simulation of rockfall trajectories with consideration of rock shape, Multibody System Dynamics, 2014). From the authors' point of view, how will rock geometry influence the conclusions obtained in this work?

AC: Thanks for the reviewer's suggestion. Compared with the non-spherical bodies, spherical bodies with same quality are relatively difficult to be resisted by the same control methods through a large number of tests. such as in (Asteriou et al, Empirical Model for Predicting Rockfall Trajectory Direction, Rock Mech Rock Eng, 2016). The effect of shape was examined by performing tests with spherical and cubical blocks. Spherical blocks presented higher and more consistent COR values compared to cubical blocks. The difference in the scatter of the values is attributed to the contact configuration of the blocks; spheres impact in a repeatable manner while cubes have numerous different configurations. A phenomenon is also reported in (Leine et al., Simulation of rockfall trajectories with consideration of rock shape, Multibody System Dynamics, 2014; Giani, G. Rock Slope Stability Analysis. Balkema, Rotterdam,1992) and it is suggested that tabular shaped rocks gradually become rounded and wheel-like due to sharp corners breaking off during the descent. Because the kinetic energy of rocks with non-spherical or polyhedral shape can be reduced more sharply during the process of rolling. If the designed cushion can resist the spherical rocks, and it also can resist effectively the non-spherical rocks. When designing the protective cushion, we should consider the serious conditions of spherical rocks to ensure fully the safety of worker, thus I think it is significant to perform cushion test using spherical rock, and rock geometry will have a slight influence on this conclusions.

Please also note the supplement to this comment:
https://www.nat-hazards-earth-syst-sci-discuss.net/nhess-2018-16/nhess-2018-16-AC1-supplement.pdf
* * *
Interactive
comment

[Figure]

Fig. 7 Photographs of a cushion (a) before and (b) after a rock impact experiment

**Fig. 1.**

**Supplement:**

[Figure]

[Figure]

Fig. 7 Photographs of a cushion (a) before and (b) after a rock impact experiment

Table. 1 the experimental parameters of the first group of tests

| | | d(mm) h(cm) | 2mm | 6mm | 10mm | 14mm | 18mm | 24mm |
|---|---|---|---|---|---|---|---|---|
| The first group of tests (movement height H=1.2m) | R=2cm | 2cm | | | | | | |
| | | 4cm | | | | | | |
| | | 6cm | | | | | | |
| | | 8cm | | | | | | |
| | | 10cm | | | | | | |
| | | 12cm | | | | | | |
| | | 14cm | | | | | | |
| | R=3cm | d(mm) h(cm) 2mm | 6mm | 10mm | 14mm | 18mm | 24mm | |
| | | 2cm | | | | | | |
| | | 4cm | | | | | | |
| | | 6cm | | | | | | |
| | | 8cm | | | | | | |
| | | 10cm | | | | | | |
| | | 12cm | | | | | | |
| | | 14cm | | | | | | |
| | R=4cm | d(mm) h(cm) 2mm | 6mm | 10mm | 14mm | 18mm | 24mm | |
| | | 2cm | | | | | | |
| | | 4cm | | | | | | |
| | | 6cm | | | | | | |
| | | 8cm | | | | | | |
| | | 10cm | | | | | | |
| | | 12cm | | | | | | |
| | | 14cm | | | | | | |
| | R=5cm | d(mm) h(cm) 2mm | 6mm | 10mm | 14mm | 18mm | 24mm | |
| | | 2cm | | | | | | |
| | | 4cm | | | | | | |
| | | 6cm | | | | | | |
| | | 8cm | | | | | | |
| | | 10cm | | | | | | |
| | | 12cm | | | | | | |
| | | 14cm | | | | | | |

Table. 2 the experimental parameters of the second group of tests

| | | d(mm) H(m) | 2mm | 6mm | 10mm | 14mm | 18mm | 24mm |
|---|---|---|---|---|---|---|---|---|
| The second group of tests (chushion thickness h=8cm) | R=2cm | 0.4m | | | | | | |
| | | 0.8m | | | | | | |
| | | 1.2m | | | | | | |
| | | 1.6m | | | | | | |
| | R=3cm | d(mm) H(m) 2mm | 6mm | 10mm | 14mm | 18mm | 24mm | |
| | | 0.4m | | | | | | |
| | | 0.8m | | | | | | |
| | | 1.2m | | | | | | |
| | | 1.6m | | | | | | |
| | R=4cm | d(mm) H(m) 2mm | 6mm | 10mm | 14mm | 18mm | 24mm | |
| | | 0.4m | | | | | | |

| | | 0.8m | | | | | | |
|---|---|---|---|---|---|---|---|---|
| | | 1.2m | | | | | | |
| | | 1.6m | | | | | | |
| | | H(m) \ d(mm) | 2mm | 6mm | 10mm | 14mm | 18mm | 24mm |
| | R=5cm | 0.4m | | | | | | |
| | | 0.8m | | | | | | |
| | | 1.2m | | | | | | |
| | | 1.6m | | | | | | |

Table 3. Orthogonal test results with the uncertainties

| r=2cm,H=0.4m,h=2cm,d=2mm | | | Average Value |
|---|---|---|---|
| 0.269 | 0.274 | 0.291 | 0.278 |
| r=2cm,H=0.8m,h=4cm,d=6mm | | | Average Value |
| 0.253 | 0.268 | 0.298 | 0.273 |
| r=2cm,H=1.2m,h=6cm,d=10mm | | | Average Value |
| 0.255 | 0.278 | 0.313 | 0.282 |
| r=2cm,H=1.6m,h=8cm,d=14mm | | | Average Value |
| 0.266 | 0.298 | 0.321 | 0.295 |

| r=3cm,H=0.4m,h=2cm,d=6mm | | | Average Value |
|---|---|---|---|
| 0.287 | 0.287 | 0.308 | 0.294 |
| r=3cm,H=0.8m,h=4cm,d=2mm | | | Average Value |
| 0.252 | 0.261 | 0.282 | 0.265 |
| r=3cm,H=1.2m,h=6cm,d=14mm | | | Average Value |
| 0.275 | 0.319 | 0.357 | 0.317 |
| r=3cm,H=1.6m,h=8cm,d=10mm | | | Average Value |
| 0.264 | 0.273 | 0.303 | 0.280 |

| r=4cm,H=0.4m,h=4cm,d=10mm | | | Average Value |
|---|---|---|---|
| 0.265 | 0.304 | 0.319 | 0.296 |
| r=4cm,H=0.8m,h=2cm,d=14mm | | | Average Value |
| 0.304 | 0.354 | 0.356 | 0.338 |
| r=4cm,H=1.2m,h=8cm,d=2mm | | | Average Value |
| 0.232 | 0.261 | 0.275 | 0.256 |
| r=4cm,H=1.6m,h=6cm,d=6mm | | | Average Value |
| 0.247 | 0.286 | 0.319 | 0.284 |

| r=5cm,H=0.4m,h=4cm,d=14mm | | | Average Value |
|---|---|---|---|
| 0.283 | 0.300 | 0.344 | 0.309 |
| r=5cm,H=0.8m,h=2cm,d=10mm | | | Average Value |
| 0.288 | 0.336 | 0.360 | 0.328 |
| r=5cm,H=1.2m,h=8cm,d=6mm | | | Average Value |
| 0.251 | 0.291 | 0.298 | 0.280 |
| r=5cm,H=1.6m,h=6cm,d=2mm | | | Average Value |

| 0.249 | 0.277 | 0.293 | 0.273 |
|---|---|---|---|

| r=2cm,H=0.4m,h=8cm,d=2mm | | | Average Value |
|---|---|---|---|
| 0.199 | 0.218 | 0.231 | 0.216 |
| r=2cm,H=0.8m,h=6cm,d=6mm | | | Average Value |
| 0.239 | 0.267 | 0.289 | 0.265 |
| r=2cm,H=1.2m,h=4cm,d=10mm | | | Average Value |
| 0.273 | 0.298 | 0.335 | 0.302 |
| r=2cm,H=1.6m,h=2cm,d=14mm | | | Average Value |
| 0.319 | 0.361 | 0.394 | 0.358 |

| r=3cm,H=0.4m,h=8cm,d=6mm | | | Average Value |
|---|---|---|---|
| 0.211 | 0.239 | 0.243 | 0.231 |
| r=3cm,H=0.8m,h=6cm,d=2mm | | | Average Value |
| 0.243 | 0.258 | 0.267 | 0.256 |
| r=3cm,H=1.2m,h=4cm,d=14mm | | | Average Value |
| 0.291 | 0.344 | 0.346 | 0.327 |
| r=3cm,H=1.6m,h=2cm,d=10mm | | | Average Value |
| 0.324 | 0.347 | 0.382 | 0.351 |

| r=4cm,H=0.4m,h=6cm,d=10mm | | | Average Value |
|---|---|---|---|
| 0.254 | 0.284 | 0.323 | 0.287 |
| r=4cm,H=0.8m,h=8cm,d=14mm | | | Average Value |
| 0.259 | 0.273 | 0.311 | 0.281 |
| r=4cm,H=1.2m,h=2cm,d=2mm | | | Average Value |
| 0.315 | 0.337 | 0.356 | 0.336 |
| r=4cm,H=1.6m,h=4cm,d=6mm | | | Average Value |
| 0.291 | 0.312 | 0.351 | 0.318 |

| r=5cm,H=0.4m,h=6cm,d=14mm | | | Average Value |
|---|---|---|---|
| 0.272 | 0.295 | 0.309 | 0.292 |
| r=5cm,H=0.8m,h=8cm,d=10mm | | | Average Value |
| 0.193 | 0.284 | 0.348 | 0.275 |
| r=5cm,H=1.2m,h=2cm,d=6mm | | | Average Value |
| 0.323 | 0.346 | 0.372 | 0.347 |
| r=5cm,H=1.6m,h=4cm,d=2mm | | | Average Value |
| 0.270 | 0.289 | 0.323 | 0.294 |

[Figure]

**CERTIFICATE OF ENGLISH EDITING**

This is to certify that the manuscript entitled
**The effects of gravel cushion particle size and thickness on coefficient of restitution under the rockfall impacts**
commissioned to us has been carefully edited by a native English-speaking editor of MogoEdit, and the grammar, spelling, and punctuation have been verified and corrected where needed. Based on this review, we believe that the language in this paper meets academic journal requirements. Please contact us with any questions.

[Figure]

*Gang Zhang*

Dr. Gang Zhang
Founder & CEO of MogoEdit

Date of Issue
March 15, 2018

**Disclaimer:** The changes in the document may be accepted or rejected by the authors in their sole discretion after our editing. However, MogoEdit is not responsible for revisions made to the document after our edit on **March 15, 2018**.

MogoEdit is a professional English editing company who provides English language editing, translation, and publication support services to individuals and corporate customers worldwide. As a company invested by the affiliate fund of Chinese Academy of Science, MogoEdit is one of the leading language editing service providers in China, whose clients come from more than 1000 universities and research institutes.

MogoEdit Website:    http://en.mogoedit.com/
500+ native English editors:   http://en.mogoedit.com/editors

[Figure]

Mogo Internet Technology Co., LTD.
No. 25, 1st Gaoxin Road, Xi'an 710075, PR China +86 02988317483 support@mogoedit.com

**The effects of gravel cushion particle size and thickness on coefficient of restitution under the rockfall impacts**

*Zhu Chun[1,2,3], Wang Dongsheng[2,3], Xia Xing[2,3], Tao ZhiGang[2,3], HeManChao[1,2,3], Cao Chen\*[1]*

Corresponding Email:zhuchuncumtb@163.com; ccao@jlu.edu.cn (Corresponding Author)

*(1.    College of Construction Engineering, Jilin University, Changchun 130026, China)*

*(2.    State Key Laboratory for Geomechanics & Deep Underground Engineering, Beijing 100083, China)*

*(3.    School of Mechanics and Civil Engineering, China University of Mining & Technology, Beijing 100083,China)*

**Abstracts:** Gravel cushions are widely used for energy-absorption in open-pit mine rockfall protection. This study investigates the energy consumption and buffering mechanism of different thicknesses and particle sizes of gravel cushion under the effects of impact. A series of laboratory tests were conducted for different cushion parameters, varying both the radius (and hence mass) of the falling block and its drop height. Tests using a constant rockfall release height indicate that changes in cushion thickness have an appreciably different effect on the coefficient of restitution (COR) of the cushion under different impact energies. Tests with identical cushion thickness, but with blocks of different radii colliding with the cushion show that the range in COR for blocks of a large radius is greater than for blocks with a relatively small radius. The degree of influence of the particle size and thickness of the cushion on the COR when rockfall moves through the cushion was also studied .Based on the orthogonal test principle, 32 orthogonal tests are conducted to explore the degree of influence of each factor on the damage depth,$L$, of the cushion and the COR of the collision between rockfall and cushion. The results show that cushion thickness,$h$, should be a primary consideration during cushion design, as an appropriate cushion not only effectively reduces COR but also remains more stable. This study thus provides a widely-applicable theoretical and practical basis for the design of cushions for mitigating rockfall hazard in open pit mines.

**Keywords:** Rockfall; cushion thickness; laboratory test; particle size; coefficient of restitution (COR).

**1 Introduction**

Rockfall constitutes a serious hazard in the working areas and facilities of the world's open-pit mines. Where slope surfaces are seriously weathered and the disturbing forces from mining are strong, landslides and rock-body collapse are prone to occur during rainfall. In rockfall, rocks roll down slope due to instability caused by gravity or exogenic action and come to rest at an obstacle or in the gentler part of the slope (Huang et al., 2007). Rockfall is widely distributed and occurs suddenly, posing a serious threat to life and property (Pantelidis, 2009; Pantelidis, 2010). In response to frequent rockfall disasters in recent years, numerous scholars in China and abroad have conducted in-depth studies into the characteristics of rockfall movement through theoretical analysis, field tests, and numerical simulation. For example, the collision rebound phenomenon of test blocks on a sandy slope has been studied through indoor small-scale, mid-sized and large-scale tests (Heidenreich, 2004; Labiouse, 2009).The effectiveness of protection measures and their influence on rockfall hazard zoning have also been evaluated. For example, Howaldetal.(2017) evaluated the protective capacity of existing and newly proposed protection measures and considered the possible reclassification of hazard as a function of the mitigation role played by the measure. Mignelli et al.(2014), meanwhile, applied a rockfall risk management approach to the road infrastructure network of the Regione Autonoma Valle D'Aosta in order to calculate the level of risk and the potential for its reduction by rockfall protection devices. A comparative analysis of road accidents in the Aosta Valley was then undertaken to verify the methodology. Thornton et al. (1998) used Hertz contact theory, the view that material accords with ideal elastic-plastic characteristics, to study the modes of calculation of the normal and tangential collision coefficients of restitution of spheres. The effect of shape has been examined by performing tests with spherical and cubic blocks, finding that spherical blocks show higher and more consistent COR values than cubic blocks (Asteriou et al., 2016).Numerical simulation software has been adopted to analyze the characteristics of rockfall movement. The software RocFall 3.0 has been adopted indam construction, road construction andthe protection ofhistorical placesto calculate the velocity and locus of rockfalland avoid damage to theproject(Topal et al., 2006; Koleini and Van Rooy, 2011; Saroglou et al., 2012; Sadagah, 2015).State-of-the-art simulation techniques incorporating nonsmooth contact dynamics and multibody dynamics have been applied to and adapted for the efficient simulation of rockfall trajectories, and the influence of rock geometry on rockfall dynamics has been studied through numerical simulation (Leineet al., 2014).

The research outlined above indicates that several types of protection measure can be effective in controlling rockfall. Trees have a significant blocking effect on rolling rocks. Interception influence tests on the effect of trees on rockfall have been designed based on analysis of the velocity change, the distance traveled by the rockfall, and the probability of collision between trees and rockfall (Huang, 2010;Notaro, 2012; Monnetet al., 2017). Semi-rigid rockfall protection barriers have been installed along areas threatened by rockfall events, and numerical investigation of semi-rigid rockfall protection barriers has been carried out to obtain essential structural information such as the energy-absorption capacity of such barriers (Miranda et al., 2015).A large-scale field test of the impact caused by rockfallon reinforced concrete beams has been conductedandthe process of dynamic response studied and compared with the results of numerical simulation (Kishi et al., 2002; Bhatti et al., 2009; Kishi et al., 2010; Bhatti et al. 2010).Concrete barriers are classified as rigid barriers, as they absorb most of the impact and all of the residual kinetic energy of the falling rock instead of dissipating it as do flexible nets. Experience has shown that rigid walls have a tendency to break under high-impact loads and may shatter, sometimes violently (Badger et al., 2009). A method whereby short concrete-filled steel tubes were inserted between the pillars and cover plate of a rock shed was proposed, and the deformation and energy absorption characteristics of the supporting member were studied through tests and theoretical analysis (Delhomme et al., 2005; Mommessin et al., 2004). Kawahara et al.(2006) conducted a large number of experiments for different soils under different combinations of falling mass and drop height and studied the influence of soil characteristics on the impact response to rockfall. Furthermore, Lambert et al. (2014) conducted real-scale impact experiments with impact energies ranging from 200 to 2200 kJ. They studied the response of rockfall protection embankments composed of a 4-m high cellular wall when exposed to a rock impact and compared this with previous real-scale experiments on other types of embankment. Finally, Sun et al. (2016) used a tire cushion layer to absorb rockfall impact, utilizing the radial deformation of the tire. They built a reinforced concrete structure model with a tire cushion layer and carried out artificial rockfall tests.

[revised manuscript text omitted]

In order to explore the effect of different cushion thicknesses and particle sizeson the rolling motion of a rockfall, massive gypsum boards with thesame propertiesas the blocks were broken,and gypsum particles for simulating the gravel cushion were divided by coarseness
using0.2cm, 0.6cm, 1.0cm, 1.4cm, 1.8cmand 2.4cmsieves (Figure 4).

[Figure]

Fig.4 Sieved granules of different particle sizes

A simple rolling stone releasing device is shown in Figure 5, atube with adjustable inclination
and height is used to adjust the translational impact velocity of the blocks (Asteriou et al., 2012).
The blocks slide and roll through the tube to collide with the plate. Two synchronized digital
cameras (1024×1024 pixels and a 200 fps capture rate) were used to acquire the velocities of the
blocks in stereoscopic space (Bouguet, 2008; Asteriou et al., 2013).

The two cameras, which obtained the motion, velocity, and kinetic energy automatically,
were placed symmetrically at a distance of approximately 0.9m from the impact surface (Figure 5).
The distance between the two cameras was about 1.2m, making the cameras look down slightly at
the targeted platform.

[revised manuscript text omitted]

Fig. 7 Photographs of a cushion (a) before and (b) after arock impact experiment

**3.3 Experimental results and discussion**

3.3.1 Experimental results

The *COR* for blocks released from a height of 1.2m to collide with an uncushioned plate is shown in Figure 8.

[Figure]

Fig. 8The COR of block collisionswith the plate

CORs derived from experiments where rockfalls of different radii were released from a 1.2m movement height to collide with a paved plate with various cushion thicknesses and particle sizes are plotted in Figure 9.

[Figure]

Fig.9 Comparison of the COR of different blocks released from a height of 1.2m

CORs derived for rockfalls of different radii released from different movement heights to collide with an8-cm thick cushion of various particle size are plotted inFigure10.

[Figure]

[Figure]

Fig.10 Comparison of the COR for different blocks colliding with an 8-cm thick cushion

3.3.2 Discussion

The figures above indicate that cushion thickness and particle size have a strong influence on the COR of collisions between a rockfall and a cushion, whereas the influence of rockfall block radius is relatively weak. When the particle size of the cushion is small and its thickness is large, the COR of the collision is small, and its effectiveness for energy-consumption is obvious. With an increase in rockfall block radius and movement height, the impact energy increases dramatically for rockfalls colliding with a cushion (Kawahara et al., 1998). Under low impact energy, changes in cushion thickness have a relatively small effect on the COR of the collision between rockfall and cushion, and even thin cushions have a certain energy-absorbing effect, as verified by Pei (2016) and Kawahara (2006). However, under high impact energy, the difference in energy-absorption of different thicknesses of gravel cushion is marked. Because a thin cushion can be more easily compressed in a very short time, the rockfall is more likely to be affected by the underlying platform at low cushion thicknesses. This makes reducing the cushion thickness equivalent to increasing the effective stiffness of the cushion, significantly limiting its buffering and energy-absorbing effect. When the cushion thickness is relatively small, the COR increases significantly with a decrease in cushion thickness. However, when the cushion's thickness is relatively large, this trend is no longer obvious.

When a constant rockfall release height of 1.2m is used, the COR is large where there is no cushion and decreases significantly with an increase in cushion thickness, which agrees with the observations of Kawahara (2005). However, when the cushion reaches a certain thickness, namely, the ratio of the falling block radius, $r$, to the cushion thickness, $h$, is 1/4–1/3, the rate of reduction in the COR with an increasein cushion thickness gradually decreases.COR is more sensitive to the thickness ofcushions with asmall particle size than those witharelatively large particle size:the range inCORscaused bythicknessvariationis wider for small cushion particle sizes, while, as thethickness of cushions with a large particle size is increased, the COR of the collision between rockfall and cushionchangesrelatively slightly.

If the cushion thickness is kept constant at 8cm, as the movement height of the block increases the COR also increases, but when blocks of different radii collide with a cushion of the same thickness, the range in the COR of blocks with a large radius is larger than for blocks with a relatively small radius. When the blocks move from a relatively low height, the COR of the collision between rockfall and cushion is more likely to be affected by the particle size compared to when blocks are released from a greater height. When the cushion particle size is large, the difference in collision configuration between the rockfall and cushion is more pronounced, resulting in a wide range in the COR of the collision between rockfall and cushion.

**4 Orthogonal test design**

**4.1 Orthogonal test procedure**

To explore the degree of influence of cushion particle size and thickness on *COR* when a rockfall moves through the cushion, orthogonal test theory was adopted to design a test program (Tao et al., 2017). Orthogonal testing is a design method that allows testing of multiple factors and multiple levels. It is based on orthogonality and selects representative points from a comprehensive experiment for testing. The orthogonal test method has the advantages of being uniformly dispersed, neat and comparable, making each test highly representative so that fewer trials can fully reflect the impact of the variation of each factor on the index.

[revised manuscript text omitted]

4.2.2 Results of analysis and discussion

Range analysis was used to analyze the orthogonal test results in Table 2. If the influencing factors for the range analysis are the damage depth,$L$, of the cushion and the COR of the collision between rockfall and cushion (Table 3), then the optimum parameter combination for rockfall block radius,$r$, movement height,$H$, cushion thickness, $h$, and particle size, $d$, to reduce COR can be obtained.

Table 3 Influencing factor range analysis of all evaluation indices

| Evaluation index | Levels | Rockfall radius $r/cm$ | Movement height $H/m$ | Cushions thickness $h/cm$ | Particle size $d/cm$ |
|---|---|---|---|---|---|
| COR of collision between rockfall and cushion | $k_1$ | 0.285 | 0.271 | 0.325 | 0.270 |
| | $k_2$ | 0.288 | 0.287 | 0.296 | 0.285 |
| | $k_3$ | 0.298 | 0.305 | 0.281 | 0.301 |
| | $k_4$ | 0.299 | 0.306 | 0.267 | 0.313 |
| | $R_y$ | 0.014 | 0.035 | 0.058 | 0.043 |
| Damage depth of cushion $L$ | $k_1$ | 0.97 | 0.78 | 0.76 | 1.75 |
| | $k_2$ | 1.12 | 0.99 | 1.26 | 1.35 |
| | $k_3$ | 1.33 | 1.38 | 1.40 | 0.94 |
| | $k_4$ | 1.44 | 1.72 | 1.44 | 0.81 |
| | $R_y$ | 0.47 | 0.94 | 0.68 | 0.94 |

The following conclusions can be derived from Table 3:

(1) The degree of influence of the factors considered on the COR of the collision between rockfall and cushion is: cushion thickness ($h$)>particle size ($d$)>movement height $(H)$>block radius ($r$); (2) The degree of influence of the factors considered on the damage depth,$L$, of the cushionis:

particle size ($d$)=movement height ($H$)>cushion thickness ($h$)>block radius ($r$).

$E$-$I$ tendency figures (Tao et al., 2017) are used to further explore the effects of each factor on the test indices. The level of all factors is the X-coordinate ($E$), and the average value of the test index is the Y-coordinate ($I$). The $E$-$I$ tendency drawings shown in Figure 12 and Figure 13

intuitively reflect the tendency of the test index with a change in factor level and can point the way to further testing.

[Figure]

Fig.12 Tendency of each factor as regards the COR of the collision between rockfall and cushion

[Figure]

Fig.13Tendency of each factor as regards damage depth **L** of the cushion

The following conclusions can be derived from Figures11and12:

(1) The smallest optimal parameter combination of the COR of the collision between rockfall and cushion is A1B1C4D1; that is, when $r$=2cm, $H$=0.4m, $h$=8, $d$=1.4, the *COR* of the collision between rockfall and cushion is the smallest (Figure 12).

(2)The shallowest optimal parameter combination of damage depth,*L,*of the cushion isA1B1C1D4; that is, when $r$=2cm, $H$=0.4m, $h$=2, $d$=1.4, the damage depth,*L*, of the cushion is the shallowest (Figure 13).

To sum up, the cushion thickness,*h*, has the most significant influence on the COR of the collision between rockfall and cushion, while it has arelatively minor effect on the damage depth,*L*, of the cushion. The second most importantfactoris particle size,*d*, but the cushion can easily be destroyed when arockfall with a high kinetic energy collides with a cushion of small particle size. The degree of influence of the rockfall block radius,*r*, on the two indicesis far less than that of the other factors. When a gravel cushion is used to control rockfall down a slope, the effectiveness with which it controls the rockfall and its durability is taken into account (Pichler et al., 2005) so the cushion thickness,*h*, should be the primary consideration in cushion design. The optimal thickness is 3–4 times the radius of the majority of the rockfall blocks. The smaller the particle size is, the smaller the COR is, but the cushion is also more likely to be destroyed so the appropriate particle size must be determined by combining the expected and evaluated block size and drop height of the rockfall so that the cushion not only achieves the effect of reducing COR but also maintains its stability.

**5  Conclusions**

The buffering and energy-dissipation mechanism of gravel cushions with different properties under different impact energies were studied through laboratory collision tests, leading to in the following conclusions:

1. Unlike conventional protection measures, a gravel cushion makes full use of waste mullock produced in the process of mine extension, which can be conveniently broken up into particles of the appropriate size. This can not only reduce the costs of reducing rockfall hazard and of mullock transportation and relieve overloading of the mine's dump but can also achieve better control of rockfalls, realizing the goal of "stone conquers stone."

2. Through laboratory tests of cushions with different parameters, varying both the radius (and hence mass) of the falling block and its drop height, it is found that a change in the thickness of the cushion has a more significant effect on the COR of the collision between rockfall and cushion under the impact of a rockfall with high impact energy than under the impact of a rockfall with low impact energy. 
[revised manuscript text omitted]

---

## Author Comment (AC2) · 17 Mar 2018

Answer to referee 2 comments Reviewer 2 R 1: Primarily, the experimental findings need to be printed alongside with their error bars. Without the given uncertainties, it is not obvious whether the drawn conclusions can be labelled as significant or something be called as leading parameter. It is of key importance for the authors as for the reader likewise to be able to judge the results against their experimental uncertainties. From the description of the experimental work, it seems that the only measure to mitigate statistical outliers is, that each series result is the mean of an experimental triplet. It has to be shown that this procedure is sufficient to generate statistically significant data. If error bars should even out all recorded differences, the experimental method has to be improved.

AC: Thanks for the reviewer's suggestion. I think the packing structure of the cushion particles don't have the influence on the COR result. Because the proportion of the size of rockfall and the coverage area of cushion is relatively small in the experiment, and the impact range when rockfall collided with the cushion is less than the coverage area of the cushion (See Figure 7 in Supplement). In field engineering, when the rockfall collide with the cushion, the gravel particles cushion of other areas will also generate interlocking effect on the particles in the collision area, which is similar to the packing structure in the tests. The main uncertainties in the test results arise in tests with large cushion particles, where the wider scatter of the values is attributed to the contact configuration between the large cushion particles and the blocks: large cushion particles have numerous different configurations. This also affected the deviation in the trajectory caused by the impact, which had a drastically higher uncertainty than for small cushion particles. In order to counteract the effects of chance, a "three tests for the mean" method was adopted, and the average value was set as the final result for each data point in the figures and tables presented here. For cushion particle sizes of 1.8cm and 2.4cm, each test was repeated five times, and the middle three values were used to obtain the average value, while for cushion particle sizes of less than 1.8 cm, each test was conducted three times. If an obviously outlying result was obtained, the test was repeated to reduce the error, thus this procedure is sufficient to generate statistically significant data. I have mentioned it in 'Experimental procedure' section. The test results including the error bars are shown in Table 1-4 (See Supplement).

R 2: Secondly, the text requires substantial refurbishments with respect to language. The authors should invest in clearer structure when describing the experimental setup as well as the testing procedure. Furthermore, figures should be labelled correctly and descriptive in order to facilitate the reading. Additionally, the measurement units should be consistent within the legends and the text (mm and cm, etc.). The use of proofing tools and the revision by a native speaker is highly advised to make the text more readable.

AC: Thanks for the reviewer's suggestion. My manuscript has been edited for English by using an English editing service, the embellishment proof is supplied as the supplement. I have added units in Figures 8 and 9. The structure format of this paper and description of Figure have been revised to make readers easier to understand.

R 3: A few technical comments: The Introduction should be shortened, since the paper clearly focuses on open-pit mining questions and an overview and listing of references for general rockfall mitigation measures is not needed. References for the approximate formula for the total energy of rockfall should be backed up with a better accessible source. The experimental setup needs to be clarified, especially the positioning and use of the cameras (field of view, image processing, etc.). The concept of an orthogonal test theory should be explained and/or backed with a better accessible source. Is it just the altering of the four parameters of interest? The drawn conclusions are not written in a concise manner. Focusing on the main experimental result in a clear way would be favorable.

AC: Thanks for the reviewer's suggestion. I have deleted some outdated references, and introduced some newly important references in the 'Introduction' section to highlight the importance and uniqueness of the current study. The list of references for some general rockfall mitigation measures is to illustrate the current research on protective methods for rockfall is mainly for urban or mountain area, not suitable for open pit mine area. I will shorten some unsuitable references in the 'Introduction' section. I have added some significant references for the approximate formula for the total energy of rockfall, and believe they are credible accessible sources. I have detailed about the experimental procedures. Such as: the calculation of rock velocities, the relative position of the two cameras and preparation process of the cushion platform, the descriptions are as follows: The two cameras, which obtained the motion, velocity, and kinetic energy automatically, were placed symmetrically at a distance of approximately

0.9m from the impact surface (Figure 5). The distance between the two cameras was about 1.2m, making the cameras look down slightly at the targeted platform. The synchronized recordings from the two cameras captured a sequence of image stereopairs at time intervals of 1/200 s. By applying stereo-photogrammetric processing, the position of any point in both images can be computed in 3D space. In general, a digital image is a perspective projection of 3D space to the camera lenses. The image plane has a 2D coordinate system where position measurements can be made using pixel coordinates. The camera has a 3D reference coordinate system that is based on the image plane pointing in the viewing direction of the camera. The speed of the rocks can be obtained by measuring the distance they have moved between adjacent frames. To simulate gravel cushions of different thicknesses, a large number of 40 cm length × 40 cm width × 2 cm height hollow gypsum boards were made. A 30cm length × 30cm width × 2cm height section was cut out of the center of each board. The hollow gypsum boards were stacked on top of each other to simulate gravel cushions of different thickness, and then the hollow parts of the boards were filled with gypsum particles. The hollow boards were fixed to a massive 40cm length × 40cm width × 6cm height gypsum base to ensure the preservation of momentum from the impact. In order to accurately measure the speed of the blocks with the cameras and to avoid interference from the motion of cushion particles affected by the collision, the cushion was blackened (Figure 6). I have introduced the clear principle of 'orthogonal test' in 'Orthogonal test design' section to facilitate readers to understand. The principle of 'orthogonal test' is described as follows: Orthogonal testing is a design method that allows testing of multiple factors and multiple levels. It is based on orthogonality and selects representative points from a comprehensive experiment for testing. The orthogonal test method has the advantages of being uniformly dispersed, neat and comparable, making each test highly representative so that fewer trials can fully reflect the impact of the variation of each factor on the index. The purpose of doing an orthogonal test is to explore the degree of influence of the four different factors on the COR and damage depth, L, and find the best combination to reach the optimal protective effect when a rockfall collides

with a cushion. When these factors cannot be considered in full, the leading factor is considered to achieve the expected effects to a great extent.

The drawn conclusions have been reorganized, focusing on the main experimental result in a clear way.

Please also note the supplement to this comment:
https://www.nat-hazards-earth-syst-sci-discuss.net/nhess-2018-16/nhess-2018-16-AC2-supplement.pdf

[Figure]

Fig. 7 Photographs of a cushion (a) before and (b) after a rock impact experiment

**Fig. 1.**

**Supplement:**

[Figure]

[Figure]

Fig. 7 Photographs of a cushion (a) before and (b) after a rock impact experiment

Table 1 The COR of block collisions with the plate

| H=1.2m ,h=0cm,d=0mm | | | | | Average Value |
|---|---|---|---|---|---|
| r=2cm | | 0.363 | 0.368 | 0.421 | | 0.384 |
| r=3cm | | 0.398 | 0.428 | 0.437 | | 0.421 |
| r=4cm | | 0.386 | 0.482 | 0.443 | | 0.437 |
| r=5cm | | 0.403 | 0.458 | 0.471 | | 0.444 |

Table 2 Comparison of the COR of different blocks released from a height of 1.2m

| H=1.2m,r=2cm,h=2cm | | | | | | Average Value |
|---|---|---|---|---|---|---|
| d=2mm | | 0.310 | 0.329 | 0.339 | | 0.326 |
| d=6mm | | 0.298 | 0.348 | 0.35 | | 0.332 |
| d=10mm | | 0.319 | 0.343 | 0.376 | | 0.346 |
| d=14mm | | 0.314 | 0.344 | 0.371 | | 0.343 |
| d=18mm | 0.262 | 0.291 | 0.337 | 0.416 | 0.443 | 0.348 |
| d=24mm | 0.234 | 0.288 | 0.371 | 0.403 | 0.421 | 0.354 |

| H=1.2m,r=2cm,h=4cm | | | | | | Average Value |
|---|---|---|---|---|---|---|
| d=2mm | | 0.281 | 0.285 | 0.316 | | 0.294 |
| d=6mm | | 0.292 | 0.338 | 0.345 | | 0.325 |
| d=10mm | | 0.261 | 0.314 | 0.331 | | 0.302 |
| d=14mm | | 0.285 | 0.324 | 0.360 | | 0.323 |
| d=18mm | 0.215 | 0.252 | 0.323 | 0.376 | 0.386 | 0.317 |
| d=24mm | 0.267 | 0.270 | 0.304 | 0.362 | 0.403 | 0.312 |

| H=1.2m,r=2cm,h=6cm | | | | | | Average Value |
|---|---|---|---|---|---|---|
| d=2mm | | 0.240 | 0.267 | 0.270 | | 0.259 |
| d=6mm | | 0.239 | 0.277 | 0.306 | | 0.274 |
| d=10mm | | 0.246 | 0.283 | 0.317 | | 0.282 |
| d=14mm | | 0.251 | 0.267 | 0.331 | | 0.283 |
| d=18mm | 0.231 | 0.259 | 0.299 | 0.345 | 0.382 | 0.301 |
| d=24mm | 0.226 | 0.261 | 0.291 | 0.336 | 0.370 | 0.296 |

| H=1.2m,r=2cm,h=8cm | | | | | | Average Value |
|---|---|---|---|---|---|---|
| d=2mm | | 0.215 | 0.243 | 0.271 | | 0.243 |

| d=6mm | | 0.211 | 0.261 | 0.290 | | 0.254 |
|---|---|---|---|---|---|---|
| d=10mm | | 0.216 | 0.261 | 0.312 | | 0.263 |
| d=14mm | | 0.223 | 0.286 | 0.304 | | 0.271 |
| d=18mm | 0.205 | 0.225 | 0.285 | 0.321 | 0.356 | 0.277 |
| d=24mm | 0.193 | 0.206 | 0.292 | 0.354 | 0.371 | 0.284 |

| H=1.2m,r=2cm,h=10cm | | | | | Average Value |
|---|---|---|---|---|---|
| d=2mm | | 0.201 | 0.245 | 0.277 | | 0.241 |
| d=6mm | | 0.198 | 0.250 | 0.293 | | 0.247 |
| d=10mm | | 0.226 | 0.251 | 0.288 | | 0.255 |
| d=14mm | | 0.210 | 0.253 | 0.311 | | 0.258 |
| d=18mm | 0.190 | 0.192 | 0.273 | 0.327 | 0.363 | 0.264 |
| d=24mm | 0.204 | 0.214 | 0.292 | 0.325 | 0.350 | 0.277 |

| H=1.2m,r=2cm,h=12cm | | | | | Average Value |
|---|---|---|---|---|---|
| d=2mm | | 0.207 | 0.218 | 0.259 | | 0.228 |
| d=6mm | | 0.190 | 0.236 | 0.273 | | 0.233 |
| d=10mm | | 0.214 | 0.225 | 0.302 | | 0.247 |
| d=14mm | | 0.200 | 0.243 | 0.313 | | 0.252 |
| d=18mm | 0.189 | 0.198 | 0.236 | 0.319 | 0.328 | 0.251 |
| d=24mm | 0.198 | 0.221 | 0.251 | 0.326 | 0.343 | 0.266 |

| H=1.2m,r=2cm,h=14cm | | | | | Average Value |
|---|---|---|---|---|---|
| d=2mm | | 0.184 | 0.230 | 0.246 | | 0.22 |
| d=6mm | | 0.199 | 0.214 | 0.283 | | 0.232 |
| d=10mm | | 0.204 | 0.251 | 0.265 | | 0.24 |
| d=14mm | | 0.183 | 0.224 | 0.301 | | 0.236 |
| d=18mm | 0.162 | 0.196 | 0.260 | 0.291 | 0.314 | 0.249 |
| d=24mm | 0.194 | 0.208 | 0.250 | 0.316 | 0.353 | 0.258 |

| H=1.2m,r=3cm,h=2cm | | | | | Average Value |
|---|---|---|---|---|---|
| d=2mm | | 0.313 | 0.338 | 0.351 | | 0.334 |
| d=6mm | | 0.326 | 0.348 | 0.349 | | 0.341 |
| d=10mm | | 0.308 | 0.354 | 0.379 | | 0.347 |
| d=14mm | | 0.311 | 0.342 | 0.409 | | 0.354 |
| d=18mm | 0.261 | 0.333 | 0.336 | 0.387 | 0.396 | 0.352 |
| d=24mm | 0.256 | 0.322 | 0.369 | 0.413 | 0.420 | 0.368 |

| H=1.2m,r=3cm,h=4cm | | | | | Average Value |
|---|---|---|---|---|---|
| d=2mm | | 0.261 | 0.316 | 0.329 | | 0.302 |
| d=6mm | | 0.276 | 0.309 | 0.360 | | 0.315 |

| d=10mm | | 0.267 | 0.329 | 0.352 | | 0.316 |
|---|---|---|---|---|---|---|
| d=14mm | | 0.282 | 0.320 | 0.379 | | 0.327 |
| d=18mm | 0.245 | 0.298 | 0.314 | 0.366 | 0.381 | 0.326 |
| d=24mm | 0.253 | 0.276 | 0.321 | 0.405 | 0.415 | 0.334 |

| H=1.2m,r=3cm,h=6cm | | | | | Average Value |
|---|---|---|---|---|---|
| d=2mm | | 0.254 | 0.273 | 0.304 | | 0.277 |
| d=6mm | | 0.262 | 0.281 | 0.309 | | 0.284 |
| d=10mm | | 0.255 | 0.288 | 0.321 | | 0.288 |
| d=14mm | | 0.283 | 0.311 | 0.360 | | 0.318 |
| d=18mm | 0.185 | 0.261 | 0.300 | 0.366 | 0.384 | 0.309 |
| d=24mm | 0.248 | 0.248 | 0.337 | 0.390 | 0.393 | 0.325 |

| H=1.2m,r=3cm,h=8cm | | | | | Average Value |
|---|---|---|---|---|---|
| d=2mm | | 0.229 | 0.235 | 0.277 | | 0.247 |
| d=6mm | | 0.213 | 0.270 | 0.303 | | 0.262 |
| d=10mm | | 0.241 | 0.247 | 0.313 | | 0.267 |
| d=14mm | | 0.223 | 0.264 | 0.332 | | 0.273 |
| d=18mm | 0.214 | 0.220 | 0.303 | 0.320 | 0.341 | 0.281 |
| d=24mm | 0.235 | 0.261 | 0.292 | 0.323 | 0.354 | 0.292 |

| H=1.2m,r=3cm,h=10cm | | | | | Average Value |
|---|---|---|---|---|---|
| d=2mm | | 0.212 | 0.233 | 0.266 | | 0.237 |
| d=6mm | | 0.223 | 0.240 | 0.275 | | 0.246 |
| d=10mm | | 0.228 | 0.246 | 0.288 | | 0.254 |
| d=14mm | | 0.213 | 0.271 | 0.302 | | 0.262 |
| d=18mm | 0.192 | 0.211 | 0.251 | 0.309 | 0.324 | 0.257 |
| d=24mm | 0.211 | 0.232 | 0.246 | 0.326 | 0.341 | 0.268 |

| H=1.2m,r=3cm,h=12cm | | | | | Average Value |
|---|---|---|---|---|---|
| d=2mm | | 0.194 | 0.220 | 0.264 | | 0.226 |
| d=6mm | | 0.199 | 0.231 | 0.287 | | 0.239 |
| d=10mm | | 0.228 | 0.234 | 0.264 | | 0.242 |
| d=14mm | | 0.204 | 0.256 | 0.284 | | 0.248 |
| d=18mm | 0.213 | 0.228 | 0.242 | 0.295 | 0.320 | 0.255 |
| d=24mm | 0.186 | 0.217 | 0.259 | 0.301 | 0.336 | 0.259 |

| H=1.2m,r=3cm,h=14cm | | | | | Average Value |
|---|---|---|---|---|---|
| d=2mm | | 0.172 | 0.206 | 0.276 | | 0.218 |
| d=6mm | | 0.203 | 0.215 | 0.254 | | 0.224 |
| d=10mm | | 0.188 | 0.224 | 0.275 | | 0.229 |
| d=14mm | | 0.178 | 0.229 | 0.286 | | 0.231 |
| d=18mm | 0.186 | 0.192 | 0.244 | 0.302 | 0.335 | 0.246 |

| | | | | | |
|---|---|---|---|---|---|
| d=24mm | 0.191 | 0.228 | 0.246 | 0.312 | 0.343 | 0.262 |

| H=1.2m,r=4cm,h=2cm | | | | | | Average Value |
|---|---|---|---|---|---|---|
| d=2mm | | 0.316 | 0.338 | 0.354 | | 0.336 |
| d=6mm | | 0.329 | 0.343 | 0.372 | | 0.348 |
| d=10mm | | 0.333 | 0.351 | 0.384 | | 0.356 |
| d=14mm | | 0.321 | 0.358 | 0.416 | | 0.365 |
| d=18mm | 0.285 | 0.329 | 0.371 | 0.401 | 0.411 | 0.367 |
| d=24mm | 0.313 | 0.334 | 0.369 | 0.413 | 0.432 | 0.372 |

| H=1.2m,r=4cm,h=4cm | | | | | | Average Value |
|---|---|---|---|---|---|---|
| d=2mm | | 0.284 | 0.307 | 0.336 | | 0.309 |
| d=6mm | | 0.298 | 0.319 | 0.346 | | 0.321 |
| d=10mm | | 0.288 | 0.310 | 0.347 | | 0.315 |
| d=14mm | | 0.308 | 0.316 | 0.351 | | 0.325 |
| d=18mm | 0.263 | 0.295 | 0.339 | 0.368 | 0.374 | 0.334 |
| d=24mm | 0.275 | 0.309 | 0.326 | 0.394 | 0.410 | 0.343 |

| H=1.2m,r=4cm,h=6cm | | | | | | Average Value |
|---|---|---|---|---|---|---|
| d=2mm | | 0.267 | 0.279 | 0.294 | | 0.28 |
| d=6mm | | 0.294 | 0.304 | 0.329 | | 0.309 |
| d=10mm | | 0.273 | 0.286 | 0.317 | | 0.292 |
| d=14mm | | 0.280 | 0.292 | 0.304 | | 0.292 |
| d=18mm | 0.257 | 0.282 | 0.303 | 0.351 | 0.384 | 0.312 |
| d=24mm | 0.240 | 0.294 | 0.322 | 0.359 | 0.367 | 0.325 |

| H=1.2m,r=4cm,h=8cm | | | | | | Average Value |
|---|---|---|---|---|---|---|
| d=2mm | | 0.249 | 0.250 | 0.269 | | 0.256 |
| d=6mm | | 0.250 | 0.267 | 0.296 | | 0.271 |
| d=10mm | | 0.248 | 0.274 | 0.306 | | 0.276 |
| d=14mm | | 0.252 | 0.271 | 0.299 | | 0.274 |
| d=18mm | 0.235 | 0.263 | 0.291 | 0.325 | 0.344 | 0.293 |
| d=24mm | 0.259 | 0.267 | 0.298 | 0.341 | 0.367 | 0.302 |

| H=1.2m,r=4cm,h=10cm | | | | | | Average Value |
|---|---|---|---|---|---|---|
| d=2mm | | 0.239 | 0.249 | 0.268 | | 0.252 |
| d=6mm | | 0.235 | 0.261 | 0.278 | | 0.258 |
| d=10mm | | 0.247 | 0.264 | 0.296 | | 0.269 |
| d=14mm | | 0.254 | 0.249 | 0.292 | | 0.265 |
| d=18mm | 0.226 | 0.243 | 0.276 | 0.324 | 0.331 | 0.281 |
| d=24mm | 0.231 | 0.236 | 0.277 | 0.321 | 0.354 | 0.278 |

| H=1.2m,r=4cm,h=12cm | | | | | Average Value |
|---|---|---|---|---|---|
| d=2mm | | 0.229 | 0.231 | 0.248 | | 0.236 |
| d=6mm | | 0.221 | 0.244 | 0.270 | | 0.245 |
| d=10mm | | 0.216 | 0.227 | 0.268 | | 0.237 |
| d=14mm | | 0.207 | 0.239 | 0.283 | | 0.243 |
| d=18mm | 0.186 | 0.208 | 0.251 | 0.297 | 0.319 | 0.252 |
| d=24mm | 0.217 | 0.218 | 0.272 | 0.284 | 0.326 | 0.258 |

| H=1.2m,r=4cm,h=14cm | | | | | Average Value |
|---|---|---|---|---|---|
| d=2mm | | 0.217 | 0.218 | 0.237 | | 0.224 |
| d=6mm | | 0.215 | 0.231 | 0.259 | | 0.235 |
| d=10mm | | 0.196 | 0.229 | 0.271 | | 0.232 |
| d=14mm | | 0.219 | 0.224 | 0.268 | | 0.237 |
| d=18mm | 0.181 | 0.207 | 0.256 | 0.281 | 0.314 | 0.248 |
| d=24mm | 0.196 | 0.219 | 0.247 | 0.293 | 0.335 | 0.253 |

| H=1.2m,r=5cm,h=2cm | | | | | Average Value |
|---|---|---|---|---|---|
| d=2mm | | 0.328 | 0.336 | 0.356 | | 0.34 |
| d=6mm | | 0.320 | 0.342 | 0.364 | | 0.342 |
| d=10mm | | 0.324 | 0.351 | 0.393 | | 0.356 |
| d=14mm | | 0.342 | 0.364 | 0.398 | | 0.368 |
| d=18mm | 0.317 | 0.338 | 0.374 | 0.401 | 0.426 | 0.371 |
| d=24mm | 0.302 | 0.354 | 0.365 | 0.421 | 0.437 | 0.38 |

| H=1.2m,r=5cm,h=4cm | | | | | Average Value |
|---|---|---|---|---|---|
| d=2mm | | 0.316 | 0.317 | 0.339 | | 0.324 |
| d=6mm | | 0.297 | 0.306 | 0.330 | | 0.311 |
| d=10mm | | 0.294 | 0.321 | 0.354 | | 0.323 |
| d=14mm | | 0.318 | 0.340 | 0.374 | | 0.344 |
| d=18mm | 0.318 | 0.321 | 0.322 | 0.386 | 0.397 | 0.343 |
| d=24mm | 0.284 | 0.328 | 0.355 | 0.373 | 0.403 | 0.352 |

| H=1.2m,r=5cm,h=6cm | | | | | Average Value |
|---|---|---|---|---|---|
| d=2mm | | 0.286 | 0.286 | 0.301 | | 0.291 |
| d=6mm | | 0.274 | 0.287 | 0.315 | | 0.292 |
| d=10mm | | 0.301 | 0.324 | 0.329 | | 0.318 |
| d=14mm | | 0.291 | 0.298 | 0.338 | | 0.309 |
| d=18mm | 0.244 | 0.286 | 0.314 | 0.378 | 0.398 | 0.326 |
| d=24mm | 0.270 | 0.293 | 0.316 | 0.381 | 0.401 | 0.33 |

| H=1.2m,r=5cm,h=8cm | | | | | Average Value |
|---|---|---|---|---|---|
| d=2mm | | 0.254 | 0.262 | 0.279 | | 0.265 |
| d=6mm | | 0.270 | 0.276 | 0.294 | | 0.28 |
| d=10mm | | 0.261 | 0.294 | 0.309 | | 0.288 |
| d=14mm | | 0.272 | 0.283 | 0.324 | | 0.293 |
| d=18mm | 0.226 | 0.256 | 0.294 | 0.356 | 0.362 | 0.302 |
| d=24mm | 0.234 | 0.286 | 0.291 | 0.362 | 0.374 | 0.313 |

| H=1.2m,r=5cm,h=10cm | | | | | Average Value |
|---|---|---|---|---|---|
| d=2mm | | 0.247 | 0.261 | 0.281 | | 0.263 |
| d=6mm | | 0.233 | 0.273 | 0.289 | | 0.265 |
| d=10mm | | 0.249 | 0.257 | 0.301 | | 0.269 |
| d=14mm | | 0.250 | 0.268 | 0.298 | | 0.272 |
| d=18mm | 0.208 | 0.237 | 0.261 | 0.315 | 0.320 | 0.271 |
| d=24mm | 0.221 | 0.255 | 0.273 | 0.336 | 0.373 | 0.288 |

| H=1.2m,r=5cm,h=12cm | | | | | Average Value |
|---|---|---|---|---|---|
| d=2mm | | 0.230 | 0.236 | 0.254 | | 0.24 |
| d=6mm | | 0.213 | 0.251 | 0.265 | | 0.243 |
| d=10mm | | 0.220 | 0.245 | 0.291 | | 0.252 |
| d=14mm | | 0.242 | 0.240 | 0.289 | | 0.257 |
| d=18mm | 0.197 | 0.224 | 0.242 | 0.311 | 0.342 | 0.259 |
| d=24mm | 0.208 | 0.209 | 0.261 | 0.328 | 0.337 | 0.266 |

| H=1.2m,r=5cm,h=14cm | | | | | Average Value |
|---|---|---|---|---|---|
| d=2mm | | 0.207 | 0.217 | 0.236 | | 0.22 |
| d=6mm | | 0.204 | 0.229 | 0.257 | | 0.23 |
| d=10mm | | 0.227 | 0.234 | 0.250 | | 0.237 |
| d=14mm | | 0.215 | 0.241 | 0.270 | | 0.242 |
| d=18mm | 0.184 | 0.189 | 0.235 | 0.278 | 0.304 | 0.234 |
| d=24mm | 0.201 | 0.218 | 0.258 | 0.286 | 0.316 | 0.254 |

Table 3 Comparison of the COR for different blocks colliding with an 8-cm thick
                          cushion

| h=8cm,r=2cm,H=0.4m | | | | | Average Value |
|---|---|---|---|---|---|
| d=2mm | | 0.197 | 0.214 | 0.237 | | 0.216 |
| d=6mm | | 0.221 | 0.222 | 0.241 | | 0.228 |
| d=10mm | | 0.216 | 0.228 | 0.264 | | 0.236 |
| d=14mm | | 0.223 | 0.256 | 0.283 | | 0.254 |
| d=18mm | 0.204 | 0.201 | 0.261 | 0.306 | 0.311 | 0.256 |
| d=24mm | 0.203 | 0.221 | 0.264 | 0.295 | 0.316 | 0.260 |

| h=8cm,r=2cm,H=0.8m | | | | | Average Value |
|---|---|---|---|---|---|
| d=2mm | | 0.219 | 0.231 | 0.237 | | 0.229 |
| d=6mm | | 0.200 | 0.248 | 0.254 | | 0.234 |
| d=10mm | | 0.220 | 0.242 | 0.273 | | 0.245 |
| d=14mm | | 0.213 | 0.246 | 0.270 | | 0.243 |
| d=18mm | 0.214 | 0.221 | 0.274 | 0.291 | 0.327 | 0.262 |
| d=24mm | 0.191 | 0.212 | 0.271 | 0.318 | 0.331 | 0.267 |

| h=8cm,r=2cm,H=1.2m | | | | | Average Value |
|---|---|---|---|---|---|
| d=2mm | | 0.228 | 0.236 | 0.265 | | 0.243 |
| d=6mm | | 0.217 | 0.267 | 0.278 | | 0.254 |
| d=10mm | | 0.231 | 0.262 | 0.296 | | 0.263 |
| d=14mm | | 0.222 | 0.283 | 0.308 | | 0.271 |
| d=18mm | 0.207 | 0.226 | 0.287 | 0.318 | 0.336 | 0.277 |
| d=24mm | 0.226 | 0.247 | 0.299 | 0.306 | 0.337 | 0.284 |

| h=8cm,r=2cm,H=1.6m | | | | | Average Value |
|---|---|---|---|---|---|
| d=2mm | | 0.232 | 0.239 | 0.258 | | 0.243 |
| d=6mm | | 0.240 | 0.243 | 0.273 | | 0.252 |
| d=10mm | | 0.223 | 0.294 | 0.296 | | 0.271 |
| d=14mm | | 0.245 | 0.287 | 0.338 | | 0.290 |
| d=18mm | 0.221 | 0.249 | 0.279 | 0.321 | 0.332 | 0.283 |
| d=24mm | 0.218 | 0.234 | 0.276 | 0.336 | 0.359 | 0.282 |

| h=8cm,r=3cm,H=0.4m | | | | | Average Value |
|---|---|---|---|---|---|
| d=2mm | | 0.208 | 0.227 | 0.237 | | 0.224 |
| d=6mm | | 0.212 | 0.226 | 0.255 | | 0.231 |
| d=10mm | | 0.228 | 0.231 | 0.270 | | 0.243 |
| d=14mm | | 0.216 | 0.251 | 0.289 | | 0.252 |
| d=18mm | 0.186 | 0.222 | 0.267 | 0.306 | 0.316 | 0.265 |
| d=24mm | 0.179 | 0.211 | 0.272 | 0.321 | 0.331 | 0.268 |

| h=8cm,r=3cm,H=0.8m | | | | | Average Value |
|---|---|---|---|---|---|
| d=2mm | | 0.222 | 0.235 | 0.251 | | 0.236 |
| d=6mm | | 0.218 | 0.247 | 0.264 | | 0.243 |
| d=10mm | | 0.235 | 0.251 | 0.306 | | 0.264 |
| d=14mm | | 0.220 | 0.274 | 0.292 | | 0.262 |
| d=18mm | 0.231 | 0.242 | 0.254 | 0.305 | 0.329 | 0.267 |

| d=24mm | 0.224 | 0.234 | 0.270 | 0.324 | 0.354 | 0.276 |

| h=8cm,r=3cm,H=1.2m | | | | | Average Value |
|---|---|---|---|---|---|
| d=2mm | | 0.229 | 0.243 | 0.269 | | 0.247 |
| d=6mm | | 0.241 | 0.265 | 0.28 | | 0.262 |
| d=10mm | | 0.234 | 0.27 | 0.297 | | 0.267 |
| d=14mm | | 0.221 | 0.288 | 0.310 | | 0.273 |
| d=18mm | 0.198 | 0.239 | 0.283 | 0.321 | 0.334 | 0.281 |
| d=24mm | 0.227 | 0.251 | 0.287 | 0.338 | 0.356 | 0.292 |

| h=8cm,r=3cm,H=1.6m | | | | | Average Value |
|---|---|---|---|---|---|
| d=2mm | | 0.242 | 0.251 | 0.269 | | 0.254 |
| d=6mm | | 0.228 | 0.279 | 0.288 | | 0.265 |
| d=10mm | | 0.261 | 0.284 | 0.313 | | 0.286 |
| d=14mm | | 0.256 | 0.286 | 0.325 | | 0.289 |
| d=18mm | 0.262 | 0.276 | 0.291 | 0.312 | 0.323 | 0.293 |
| d=24mm | 0.214 | 0.272 | 0.295 | 0.336 | 0.351 | 0.301 |

| h=8cm,r=4cm,H=0.4m | | | | | Average Value |
|---|---|---|---|---|---|
| d=2mm | | 0.220 | 0.227 | 0.246 | | 0.231 |
| d=6mm | | 0.233 | 0.234 | 0.259 | | 0.242 |
| d=10mm | | 0.212 | 0.241 | 0.264 | | 0.239 |
| d=14mm | | 0.241 | 0.252 | 0.299 | | 0.264 |
| d=18mm | 0.209 | 0.239 | 0.253 | 0.294 | 0.321 | 0.262 |
| d=24mm | 0.213 | 0.236 | 0.278 | 0.314 | 0.316 | 0.276 |

| h=8cm,r=4cm,H=0.8m | | | | | Average Value |
|---|---|---|---|---|---|
| d=2mm | | 0.223 | 0.247 | 0.265 | | 0.245 |
| d=6mm | | 0.248 | 0.252 | 0.271 | | 0.257 |
| d=10mm | | 0.236 | 0.257 | 0.293 | | 0.262 |
| d=14mm | | 0.260 | 0.285 | 0.316 | | 0.287 |
| d=18mm | 0.216 | 0.247 | 0.287 | 0.324 | 0.338 | 0.286 |
| d=24mm | 0.224 | 0.230 | 0.302 | 0.338 | 0.351 | 0.290 |

| h=8cm,r=4cm,H=1.2m | | | | | Average Value |
|---|---|---|---|---|---|
| d=2mm | | 0.245 | 0.254 | 0.269 | | 0.256 |
| d=6mm | | 0.230 | 0.287 | 0.296 | | 0.271 |
| d=10mm | | 0.249 | 0.281 | 0.298 | | 0.276 |
| d=14mm | | 0.268 | 0.278 | 0.306 | | 0.284 |
| d=18mm | 0.225 | 0.256 | 0.291 | 0.332 | 0.348 | 0.293 |
| d=24mm | 0.248 | 0.282 | 0.303 | 0.321 | 0.367 | 0.302 |

| h=8cm,r=4cm,H=1.6m | | | | | Average Value |
|---|---|---|---|---|---|
| d=2mm | | 0.243 | 0.257 | 0.283 | | 0.261 |
| d=6mm | | 0.269 | 0.282 | 0.304 | | 0.285 |
| d=10mm | | 0.254 | 0.283 | 0.321 | | 0.286 |
| d=14mm | | 0.237 | 0.327 | 0.333 | | 0.299 |
| d=18mm | 0.234 | 0.273 | 0.306 | 0.354 | 0.375 | 0.311 |
| d=24mm | 0.211 | 0.262 | 0.307 | 0.361 | 0.394 | 0.310 |

| h=8cm,r=5cm,H=0.4m | | | | | Average Value |
|---|---|---|---|---|---|
| d=2mm | | 0.229 | 0.231 | 0.248 | | 0.236 |
| d=6mm | | 0.243 | 0.247 | 0.269 | | 0.253 |
| d=10mm | | 0.211 | 0.256 | 0.283 | | 0.25 |
| d=14mm | | 0.232 | 0.259 | 0.298 | | 0.263 |
| d=18mm | 0.213 | 0.228 | 0.284 | 0.316 | 0.354 | 0.276 |
| d=24mm | 0.207 | 0.250 | 0.281 | 0.321 | 0.348 | 0.284 |

| h=8cm,r=5cm,H=0.8m | | | | | Average Value |
|---|---|---|---|---|---|
| d=2mm | | 0.239 | 0.246 | 0.271 | | 0.252 |
| d=6mm | | 0.252 | 0.268 | 0.281 | | 0.267 |
| d=10mm | | 0.261 | 0.284 | 0.304 | | 0.283 |
| d=14mm | | 0.230 | 0.288 | 0.298 | | 0.272 |
| d=18mm | 0.218 | 0.253 | 0.291 | 0.338 | 0.351 | 0.294 |
| d=24mm | 0.206 | 0.247 | 0.316 | 0.331 | 0.383 | 0.298 |

| h=8cm,r=5cm,H=1.2m | | | | | Average Value |
|---|---|---|---|---|---|
| d=2mm | | 0.258 | 0.259 | 0.278 | | 0.265 |
| d=6mm | | 0.237 | 0.300 | 0.303 | | 0.28 |
| d=10mm | | 0.259 | 0.286 | 0.319 | | 0.288 |
| d=14mm | | 0.247 | 0.287 | 0.345 | | 0.293 |
| d=18mm | 0.251 | 0.266 | 0.298 | 0.342 | 0.354 | 0.302 |
| d=24mm | 0.236 | 0.272 | 0.306 | 0.361 | 0.381 | 0.313 |

| h=8cm,r=5cm,H=1.6m | | | | | Average Value |
|---|---|---|---|---|---|
| d=2mm | | 0.244 | 0.279 | 0.296 | | 0.273 |
| d=6mm | | 0.268 | 0.284 | 0.309 | | 0.287 |
| d=10mm | | 0.254 | 0.307 | 0.336 | | 0.299 |
| d=14mm | | 0.271 | 0.311 | 0.348 | | 0.31 |
| d=18mm | 0.246 | 0.258 | 0.307 | 0.359 | 0.387 | 0.308 |

| d=24mm | 0.225 | 0.279 | 0.338 | 0.349 | 0.368 | 0.322 |

Table 4. Orthogonal test results

| r=2cm,H=0.4m,h=2cm,d=2mm | | | Average Value |
|---|---|---|---|
| 0.269 | 0.274 | 0.291 | 0.278 |
| r=2cm,H=0.8m,h=4cm,d=6mm | | | Average Value |
| 0.253 | 0.268 | 0.298 | 0.273 |
| r=2cm,H=1.2m,h=6cm,d=10mm | | | Average Value |
| 0.255 | 0.278 | 0.313 | 0.282 |
| r=2cm,H=1.6m,h=8cm,d=14mm | | | Average Value |
| 0.266 | 0.298 | 0.321 | 0.295 |

| r=3cm,H=0.4m,h=2cm,d=6mm | | | Average Value |
|---|---|---|---|
| 0.287 | 0.287 | 0.308 | 0.294 |
| r=3cm,H=0.8m,h=4cm,d=2mm | | | Average Value |
| 0.252 | 0.261 | 0.282 | 0.265 |
| r=3cm,H=1.2m,h=6cm,d=14mm | | | Average Value |
| 0.275 | 0.319 | 0.357 | 0.317 |
| r=3cm,H=1.6m,h=8cm,d=10mm | | | Average Value |
| 0.264 | 0.273 | 0.303 | 0.280 |

| r=4cm,H=0.4m,h=4cm,d=10mm | | | Average Value |
|---|---|---|---|
| 0.265 | 0.304 | 0.319 | 0.296 |
| r=4cm,H=0.8m,h=2cm,d=14mm | | | Average Value |
| 0.304 | 0.354 | 0.356 | 0.338 |
| r=4cm,H=1.2m,h=8cm,d=2mm | | | Average Value |
| 0.232 | 0.261 | 0.275 | 0.256 |
| r=4cm,H=1.6m,h=6cm,d=6mm | | | Average Value |
| 0.247 | 0.286 | 0.319 | 0.284 |

| r=5cm,H=0.4m,h=4cm,d=14mm | | | Average Value |
|---|---|---|---|
| 0.283 | 0.300 | 0.344 | 0.309 |
| r=5cm,H=0.8m,h=2cm,d=10mm | | | Average Value |
| 0.288 | 0.336 | 0.360 | 0.328 |
| r=5cm,H=1.2m,h=8cm,d=6mm | | | Average Value |
| 0.251 | 0.291 | 0.298 | 0.280 |
| r=5cm,H=1.6m,h=6cm,d=2mm | | | Average Value |
| 0.249 | 0.277 | 0.293 | 0.273 |

| r=2cm,H=0.4m,h=8cm,d=2mm | | | Average Value |
|---|---|---|---|
| 0.199 | 0.218 | 0.231 | 0.216 |
| r=2cm,H=0.8m,h=6cm,d=6mm | | | Average Value |

| | | | |
|---|---|---|---|
| 0.239 | 0.267 | 0.289 | 0.265 |
| r=2cm,H=1.2m,h=4cm,d=10mm | | | Average Value |
| 0.273 | 0.298 | 0.335 | 0.302 |
| r=2cm,H=1.6m,h=2cm,d=14mm | | | Average Value |
| 0.319 | 0.361 | 0.394 | 0.358 |

| | | | |
|---|---|---|---|
| r=3cm,H=0.4m,h=8cm,d=6mm | | | Average Value |
| 0.211 | 0.239 | 0.243 | 0.231 |
| r=3cm,H=0.8m,h=6cm,d=2mm | | | Average Value |
| 0.243 | 0.258 | 0.267 | 0.256 |
| r=3cm,H=1.2m,h=4cm,d=14mm | | | Average Value |
| 0.291 | 0.344 | 0.346 | 0.327 |
| r=3cm,H=1.6m,h=2cm,d=10mm | | | Average Value |
| 0.324 | 0.347 | 0.382 | 0.351 |

| | | | |
|---|---|---|---|
| r=4cm,H=0.4m,h=6cm,d=10mm | | | Average Value |
| 0.254 | 0.284 | 0.323 | 0.287 |
| r=4cm,H=0.8m,h=8cm,d=14mm | | | Average Value |
| 0.259 | 0.273 | 0.311 | 0.281 |
| r=4cm,H=1.2m,h=2cm,d=2mm | | | Average Value |
| 0.315 | 0.337 | 0.356 | 0.336 |
| r=4cm,H=1.6m,h=4cm,d=6mm | | | Average Value |
| 0.291 | 0.312 | 0.351 | 0.318 |

| | | | |
|---|---|---|---|
| r=5cm,H=0.4m,h=6cm,d=14mm | | | Average Value |
| 0.272 | 0.295 | 0.309 | 0.292 |
| r=5cm,H=0.8m,h=8cm,d=10mm | | | Average Value |
| 0.193 | 0.284 | 0.348 | 0.275 |
| r=5cm,H=1.2m,h=2cm,d=6mm | | | Average Value |
| 0.323 | 0.346 | 0.372 | 0.347 |
| r=5cm,H=1.6m,h=4cm,d=2mm | | | Average Value |
| 0.270 | 0.289 | 0.323 | 0.294 |

[Figure]

**CERTIFICATE OF ENGLISH EDITING**

This is to certify that the manuscript entitled

*The effects of gravel cushion particle size and thickness on coefficient of restitution under the rockfall impacts*

commissioned to us has been carefully edited by a native English-speaking editor of MogoEdit, and the grammar, spelling, and punctuation have been verified and corrected where needed. Based on this review, we believe that the language in this paper meets academic journal requirements. Please contact us with any questions.

[Figure]

*Gang Zhang*

Dr. Gang Zhang
Founder & CEO of MogoEdit

Date of Issue
March 15, 2018

**Disclaimer:** The changes in the document may be accepted or rejected by the authors in their sole discretion after our editing. However, MogoEdit is not responsible for revisions made to the document after our edit on **March 15, 2018**.

MogoEdit is a professional English editing company who provides English language editing, translation, and publication support services to individuals and corporate customers worldwide. As a company invested by the affiliate fund of Chinese Academy of Science, MogoEdit is one of the leading language editing service providers in China, whose clients come from more than 1000 universities and research institutes.

[Figure]

MogoEdit Website:   http://en.mogoedit.com/
500+ native English editors:   http://en.mogoedit.com/editors

Mogo Internet Technology Co., LTD.
No. 25, 1st Gaoxin Road, Xi'an 710075, PR China +86 02988317483 support@mogoedit.com

**The effects of gravel cushion particle size and thickness on**

**coefficient of restitution under the rockfall impacts**

*Zhu Chun[1,2,3], Wang Dongsheng[2,3], Xia Xing[2,3], Tao ZhiGang[2,3], HeManChao[1,2,3], Cao Chen\*[1]*

Corresponding Email:zhuchuncumtb@163.com; ccao@jlu.edu.cn (Corresponding Author)

*(1.     College of Construction Engineering, Jilin University, Changchun 130026, China)*

*(2.     State Key Laboratory for Geomechanics & Deep Underground Engineering, Beijing 100083, China)*

*(3.     School of Mechanics and Civil Engineering, China University of Mining & Technology, Beijing 100083,China)*

**Abstracts:** Gravel cushions are widely used for energy-absorption in open-pit mine rockfall protection. This study investigates the energy consumption and buffering mechanism of different thicknesses and particle sizes of gravel cushion under the effects of impact. A series of laboratory tests were conducted for different cushion parameters, varying both the radius (and hence mass) of the falling block and its drop height. Tests using a constant rockfall release height indicate that changes in cushion thickness have an appreciably different effect on the coefficient of restitution (COR) of the cushion under different impact energies. Tests with identical cushion thickness, but with blocks of different radii colliding with the cushion show that the range in COR for blocks of a large radius is greater than for blocks with a relatively small radius. The degree of influence of the particle size and thickness of the cushion on the COR when rockfall moves through the cushion was also studied .Based on the orthogonal test principle, 32 orthogonal tests are conducted to explore the degree of influence of each factor on the damage depth,*L,* of the cushion and the COR of the collision between rockfall and cushion. The results show that cushion thickness,*h*, should be a primary consideration during cushion design, as an appropriate cushion not only effectively reduces COR but also remains more stable. This study thus provides a widely-applicable theoretical and practical basis for the design of cushions for mitigating rockfall hazard in open pit mines.

**Keywords:** Rockfall; cushion thickness; laboratory test; particle size; coefficient of restitution (COR).

**1 Introduction**

Rockfall constitutes a serious hazard in the working areas and facilities of the world's open-pit mines. Where slope surfaces are seriously weathered and the disturbing forces from mining are strong, landslides and rock-body collapse are prone to occur during rainfall. In rockfall, rocks roll down slope due to instability caused by gravity or exogenic action and come to rest at an obstacle or in the gentler part of the slope (Huang et al., 2007). Rockfall is widely distributed and occurs suddenly, posing a serious threat to life and property (Pantelidis, 2009; Pantelidis, 2010). In response to frequent rockfall disasters in recent years, numerous scholars in China and abroad have conducted in-depth studies into the characteristics of rockfall movement through theoretical analysis, field tests, and numerical simulation. For example, the collision rebound phenomenon of test blocks on a sandy slope has been studied through indoor small-scale, mid-sized and large-scale tests (Heidenreich, 2004; Labiouse, 2009).The effectiveness of protection measures and their influence on rockfall hazard zoning have also been evaluated. For example, Howaldetal.(2017) evaluated the protective capacity of existing and newly proposed protection measures and considered the possible reclassification of hazard as a function of the mitigation role played by the measure. Mignelli et al.(2014), meanwhile, applied a rockfall risk management approach to the road infrastructure network of the Regione Autonoma Valle D'Aosta in order to calculate the level of risk and the potential for its reduction by rockfall protection devices. A comparative analysis of road accidents in the Aosta Valley was then undertaken to verify the methodology. Thornton et al. (1998) used Hertz contact theory, the view that material accords with ideal elastic-plastic characteristics, to study the modes of calculation of the normal and tangential collision coefficients of restitution of spheres. The effect of shape has been examined by performing tests with spherical and cubic blocks, finding that spherical blocks show higher and more consistent COR values than cubic blocks (Asteriou et al., 2016).Numerical simulation software has been adopted to analyze the characteristics of rockfall movement. The software RocFall 3.0 has been adopted indam construction, road construction andthe protection ofhistorical placesto calculate the velocity and locus of rockfalland avoid damage to theproject(Topal et al., 2006; Koleini and Van Rooy, 2011; Saroglou et al., 2012; Sadagah, 2015).State-of-the-art simulation techniques incorporating nonsmooth contact dynamics and multibody dynamics have been applied to and adapted for the efficient simulation of rockfall trajectories, and the influence of rock geometry on rockfall dynamics has been studied through numerical simulation (Leineet al., 2014).

The research outlined above indicates that several types of protection measure can be effective in controlling rockfall. Trees have a significant blocking effect on rolling rocks. Interception influence tests on the effect of trees on rockfall have been designed based on analysis of the velocity change, the distance traveled by the rockfall, and the probability of collision between trees and rockfall (Huang, 2010;Notaro, 2012; Monnetet al., 2017). Semi-rigid rockfall protection barriers have been installed along areas threatened by rockfall events, and numerical investigation of semi-rigid rockfall protection barriers has been carried out to obtain essential structural information such as the energy-absorption capacity of such barriers (Miranda et al., 2015).A large-scale field test of the impact caused by rockfallon reinforced concrete beams has been conductedandthe process of dynamic response studied and compared with the results of numerical simulation (Kishi et al., 2002; Bhatti et al., 2009; Kishi et al., 2010; Bhatti et al. 2010).Concrete barriers are classified as rigid barriers, as they absorb most of the impact and all of the residual kinetic energy of the falling rock instead of dissipating it as do flexible nets. Experience has shown that rigid walls have a tendency to break under high-impact loads and may shatter, sometimes violently (Badger et al., 2009). A method whereby short concrete-filled steel tubes were inserted between the pillars and cover plate of a rock shed was proposed, and the deformation and energy absorption characteristics of the supporting member were studied through tests and theoretical analysis (Delhomme et al., 2005; Mommessin et al., 2004). Kawahara et al.(2006) conducted a large number of experiments for different soils under different combinations of falling mass and drop height and studied the influence of soil characteristics on the impact response to rockfall. Furthermore, Lambert et al. (2014) conducted real-scale impact experiments with impact energies ranging from 200 to 2200 kJ. They studied the response of rockfall protection embankments composed of a 4-m high cellular wall when exposed to a rock impact and compared this with previous real-scale experiments on other types of embankment. Finally, Sun et al. (2016) used a tire cushion layer to absorb rockfall impact, utilizing the radial deformation of the tire. They built a reinforced concrete structure model with a tire cushion layer and carried out artificial rockfall tests.

[revised manuscript text omitted]

In order to explore the effect of different cushion thicknesses and particle sizeson the rolling
motion of a rockfall, massive gypsum boards with thesame propertiesas the blocks were broken,and gypsum particles for simulating the gravel cushion were divided by coarseness
using0.2cm, 0.6cm, 1.0cm, 1.4cm, 1.8cmand 2.4cmsieves (Figure 4).

[Figure]

Fig.4 Sieved granules of different particle sizes

A simple rolling stone releasing device is shown in Figure 5, atube with adjustable inclination
and height is used to adjust the translational impact velocity of the blocks (Asteriou et al., 2012).
The blocks slide and roll through the tube to collide with the plate. Two synchronized digital
cameras (1024×1024 pixels and a 200 fps capture rate) were used to acquire the velocities of the
blocks in stereoscopic space (Bouguet, 2008; Asteriou et al., 2013).

The two cameras, which obtained the motion, velocity, and kinetic energy automatically,
were placed symmetrically at a distance of approximately 0.9m from the impact surface (Figure 5).
The distance between the two cameras was about 1.2m, making the cameras look down slightly at
the targeted platform.

[revised manuscript text omitted]

Fig. 7 Photographs of a cushion (a) before and (b) after arock impact experiment

**3.3 Experimental results and discussion**

3.3.1 Experimental results

The *COR* for blocks released from a height of 1.2m to collide with an uncushioned plate is shown in Figure 8.

[Figure]

Fig. 8 The COR of block collisions with the plate

CORs derived from experiments where rockfalls of different radii were released from a 1.2m movement height to collide with a paved plate with various cushion thicknesses and particle sizes are plotted in Figure 9.

[Figure]

Fig.9 Comparison of the COR of different blocks released from a height of 1.2m

CORs derived for rockfalls of different radii released from different movement heights to collide with an8-cm thick cushion of various particle size are plotted inFigure10.

[Figure]

[Figure]

Fig.10 Comparison of the COR for different blocks colliding with an 8-cm thick cushion

3.3.2 Discussion

The figures above indicate that cushion thickness and particle size have a strong influence on the COR of collisions between a rockfall and a cushion, whereas the influence of rockfall block radius is relatively weak. When the particle size of the cushion is small and its thickness is large, the COR of the collision is small, and its effectiveness for energy-consumption is obvious. With an increase in rockfall block radius and movement height, the impact energy increases dramatically for rockfalls colliding with a cushion (Kawahara et al., 1998). Under low impact energy, changes in cushion thickness have a relatively small effect on the COR of the collision between rockfall and cushion, and even thin cushions have a certain energy-absorbing effect, as verified by Pei (2016) and Kawahara (2006). However, under high impact energy, the difference in energy-absorption of different thicknesses of gravel cushion is marked. Because a thin cushion can be more easily compressed in a very short time, the rockfall is more likely to be affected by the underlying platform at low cushion thicknesses. This makes reducing the cushion thickness equivalent to increasing the effective stiffness of the cushion, significantly limiting its buffering and energy-absorbing effect. When the cushion thickness is relatively small, the COR increases significantly with a decrease in cushion thickness. However, when the cushion's thickness is relatively large, this trend is no longer obvious.

When a constant rockfall release height of 1.2m is used, the COR is large where there is no cushion and decreases significantly with an increase in cushion thickness, which agrees with the observations of Kawahara (2005). However, when the cushion reaches a certain thickness, namely, the ratio of the falling block radius,$r$, to the cushion thickness,$h$, is 1/4–1/3, the rate of reduction in the COR with an increasein cushion thickness gradually decreases.COR is more sensitive to the thickness ofcushions with asmall particle size than those witharelatively large particle size:the range inCORscaused bythicknessvariationis wider for small cushion particle sizes, while, as thethickness of cushions with a large particle size is increased, the COR of the collision between rockfall and cushionchangesrelatively slightly.

If the cushion thickness is kept constant at 8cm, as the movement height of the block increases the COR also increases, but when blocks of different radii collide with a cushion of the same thickness, the range in the COR of blocks with a large radius is larger than for blocks with a relatively small radius. When the blocks move from a relatively low height, the COR of the collision between rockfall and cushion is more likely to be affected by the particle size compared to when blocks are released from a greater height. When the cushion particle size is large, the difference in collision configuration between the rockfall and cushion is more pronounced, resulting in a wide range in the COR of the collision between rockfall and cushion.

**4 Orthogonal test design**

**4.1 Orthogonal test procedure**

To explore the degree of influence of cushion particle size and thickness on *COR* when a rockfall moves through the cushion, orthogonal test theory was adopted to design a test program (Tao et al., 2017). Orthogonal testing is a design method that allows testing of multiple factors and multiple levels. It is based on orthogonality and selects representative points from a comprehensive experiment for testing. The orthogonal test method has the advantages of being uniformly dispersed, neat and comparable, making each test highly representative so that fewer trials can fully reflect the impact of the variation of each factor on the index.

[revised manuscript text omitted]

4.2.2 Results of analysis and discussion

Range analysis was used to analyze the orthogonal test results in Table 2. If the influencing factors for the range analysis are the damage depth,$L,$ of the cushion and the COR of the collision between rockfall and cushion (Table 3), then the optimum parameter combination for rockfall block radius,$r$, movement height,$H$, cushion thickness, $h$, and particle size, $d$, to reduce COR can be obtained.

Table 3 Influencing factor range analysis of all evaluation indices

| Evaluation index | Levels | Rockfall radius $r/cm$ | Movement height $H/m$ | Cushions thickness $h/cm$ | Particle size $d/cm$ |
|---|---|---|---|---|---|
| COR of collision between rockfall and cushion | $k_1$ | 0.285 | 0.271 | 0.325 | 0.270 |
| | $k_2$ | 0.288 | 0.287 | 0.296 | 0.285 |
| | $k_3$ | 0.298 | 0.305 | 0.281 | 0.301 |
| | $k_4$ | 0.299 | 0.306 | 0.267 | 0.313 |
| | $R_y$ | 0.014 | 0.035 | 0.058 | 0.043 |
| Damage depth of cushion $L$ | $k_1$ | 0.97 | 0.78 | 0.76 | 1.75 |
| | $k_2$ | 1.12 | 0.99 | 1.26 | 1.35 |
| | $k_3$ | 1.33 | 1.38 | 1.40 | 0.94 |
| | $k_4$ | 1.44 | 1.72 | 1.44 | 0.81 |
| | $R_y$ | 0.47 | 0.94 | 0.68 | 0.94 |

The following conclusions can be derived from Table 3:

(1) The degree of influence of the factors considered on the COR of the collision between rockfall and cushion is: cushion thickness ($h$)>particle size ($d$)>movement height $(H)$>block radius ($r$); (2) The degree of influence of the factors considered on the damage depth,$L,$ of the cushionis:

particle size ($d$)=movement height ($H$)>cushion thickness ($h$)>block radius ($r$).

$E$-$I$ tendency figures (Tao et al., 2017) are used to further explore the effects of each factor on the test indices. The level of all factors is the X-coordinate ($E$), and the average value of the test index is the Y-coordinate ($I$). The $E$-$I$ tendency drawings shown in Figure 12 and Figure 13

intuitively reflect the tendency of the test index with a change in factor level and can point the way to further testing.

[Figure]

Fig.12 Tendency of each factor as regards the COR of the collision between rockfall and cushion

[Figure]

Fig.13Tendency of each factor as regards damage depth **L** of the cushion

The following conclusions can be derived from Figures11and12:

(1) The smallest optimal parameter combination of the COR of the collision between rockfall and cushion is A1B1C4D1; that is, when $r=2cm$, $H=0.4m$, $h=8$, $d=1.4$, the *COR* of the collision between rockfall and cushion is the smallest (Figure 12).

(2)The shallowest optimal parameter combination of damage depth,*L,*of the cushion isA1B1C1D4; that is, when $r=2cm$, $H=0.4m$, $h=2$, $d=1.4$, the damage depth,*L*, of the cushion is the shallowest (Figure 13).

To sum up, the cushion thickness,*h*, has the most significant influence on the COR of the collision between rockfall and cushion, while it has arelatively minor effect on the damage depth,*L*, of the cushion. The second most importantfactoris particle size,*d*, but the cushion can easily be destroyed when arockfall with a high kinetic energy collides with a cushion of small particle size. The degree of influence of the rockfall block radius,*r*, on the two indicesis far less than that of the other factors. When a gravel cushion is used to control rockfall down a slope, the effectiveness with which it controls the rockfall and its durability is taken into account (Pichler et al., 2005) so the cushion thickness,*h*, should be the primary consideration in cushion design. The optimal thickness is 3–4 times the radius of the majority of the rockfall blocks. The smaller the particle size is, the smaller the COR is, but the cushion is also more likely to be destroyed so the appropriate particle size must be determined by combining the expected and evaluated block size and drop height of the rockfall so that the cushion not only achieves the effect of reducing COR but also maintains its stability.

**5 Conclusions**

The buffering and energy-dissipation mechanism of gravel cushions with different properties under different impact energies were studied through laboratory collision tests, leading to in the following conclusions:

1. Unlike conventional protection measures, a gravel cushion makes full use of waste mullock produced in the process of mine extension, which can be conveniently broken up into particles of the appropriate size. This can not only reduce the costs of reducing rockfall hazard and of mullock transportation and relieve overloading of the mine's dump but can also achieve better control of rockfalls, realizing the goal of "stone conquers stone."

2. Through laboratory tests of cushions with different parameters, varying both the radius (and hence mass) of the falling block and its drop height, it is found that a change in the thickness of the cushion has a more significant effect on the COR of the collision between rockfall and cushion under the impact of a rockfall with high impact energy than under the impact of a rockfall with low impact energy. 
[revised manuscript text omitted]

---

## Referee Comment (RC3) · Anonymous Referee #1 · 20 Mar 2018

The paper has been significantly improved considering the suggestions from the reviewer and the questions have been properly answered. Therefore, it is suggested to accept the paper for publication falls the following minor points are corrected (minor revision): (1) There are still many typo errors which might due to that the authors directly copied the improved manuscript form Word to LaTeX. Most of the time the neighboring words are connected together, or a blank space between words is missing, or the font size is not right. The authors should check very carefully the following (but may not limited to) lines and make corrections: 38-41, 56-60, 68-72, 75, 81, 85-89, 95, 107, 120, 123, 138-141, 158-159, 168-173, 179, 197, 222, 226-232, 238, 252, 276-281, 290, 300-301, 309, 321-322, 334, 342-344, 351, 359, 362, 366-376, 390, 396, 402-405. (2)

[Figure]

In line 158, it should be 'spherical blocks with diameters of 4 cm, 6 cm, . . .'. (3) In Tables 1-3, the units for the parameters should not be italic or bold. (4) It is suggested the authors to check the Eqn. (6) and the related text contents whether the parameters and subscripts are correctly written. (5) At proper places one can shortly address why the authors use spherical instead of non-spherical blocks for tests. A short comment extracted from the text already given by the authors in the answer to reviewer would be good. (6) The style of the references is not kept the same. Please very carefully check the references one by one. Attention the typos which are similar to the comment (1). (7) It is suggested to provide the three tables ('The experimental parameters of the first group of tests', 'The experimental parameters of the second group of tests', and 'Orthogonal test results with the uncertainties') as supplemental material for the paper. Both the average value and the standard deviation should be given in these tables, if it is not ideal to plot the uncertainties in the Figures 8-10. (8) Please check the style of the variable names used in the whole text, including figures and tables. Sometimes they are italic, sometimes not. It is better to keep the style consistent.

---

## Referee Comment (RC4) · Anonymous Referee #2 · 21 Mar 2018

The authors' effort to improve the manuscript are acknowledged and do lead to a general improvement of the contribution. However, the main criticism is not addressed. The urge of presenting experimental data alongside with their uncertainties is compulsory for every experimental work. It cannot be stressed enough: The measurement of a physical quantity without specifying its uncertainty is meaningless.

Although the effort of attaining the comprising set of 628 tests is valued, the specified test procedure and its error handling is not evaluated profoundly enough to strengthen the authors claims. The authors themselves state that "if an obviously outlying result was obtained, the test was repeated to reduce the error." In experimental series, a uni-

fied test procedure yields the given results and subsequent data analysis then shows the range of extremal values. The judgment of an "obviously outlying result" does not correspond to a scientifically detached experimental setup and mind setting. Outliers – or at least unexpected measurement results - might hint to unexpected effects and processes, to experimental limitations, to faulty experimental procedures, etc. and are not to be discarded a priori. The authors are urged to rethink their approach to experimental findings and focus on a nonbiased data collection.

The full table of measurement results presented in the appendix do not show the uncertainties, but has to be evaluated by the reader itself. The requirement on a scientifically valuable publication and its figures is to concisely transport the gathered knowledge to the readership. It should serve to showcase the received results together with their experimental limitations – in case they are of significant size. A summary of barely treated raw results is favorable with respect to the data origin point of view, but does not serve the purpose of transporting knowledge to the reader.

As an example for a minimal data treatment, a revised version of Figure 8 is attached with the uncertainties drawn from the presented measurement data. If taken the standard deviation from the attached measurements in Table 1 of the Appendix, then the graph looks the following. It is clear to the reader that the change in COR from 4 cm to 5 cm block size is not apparent within the error bars. Only after such considerations, any conclusion on data quality and/or experimental sufficiency in terms of number of drops per series are possible.

As example for a thorough and concise presentation of similar results, consult for example the article "Geotechnical and kinematic parameters affecting the coefficients of restitution for rock fall analysis" P.Asteriou et. al. (https://doi.org/10.1016/j.ijrmms.2012.05.029). Such data presentation is expected and needed for the COR data to merit possible publication.

Optimization analysis and discussion of test result: Due to the lack of data quality

verification in the first step, the optimization analysis is based on bad ground. However, if amendments are done and the presented search for the leading parameter for COR and damage depth should remain unchanged, the following improvements need to be done:

The presented formula (6) lacks $R_y$, so the derivation of $R_y$ is not clear. Additionally, it is suggested that instead of 4 it is suggested to state "Number of levels" such that a reader who jumps to the equations is not baffled by this specific number. As the formula is not complete and in Table 3 levels are labelled $k_1$ to $k_4$ but in the upper discussion it is only a $k_{xy}$, it is not clear how to obtain the factors presented in Table 3 for the individual levels. A clarification in notation and procedure is needed.

Furthermore, Figures 12 and 13 show many trend lines. Again, no uncertainties are given with respect to the trend lines. This is mandatory for the reader to judge the significance of the trend. An uncertainty boundary for the trend lines, estimating the error propagation from experiment to statistical evaluation should be included.

A few comments to further authors responses: Orthogonal test theory: The procedure is introduced, but merely as a disclaimer. Orthogonality is a basic concept in Linear Algebra. The labelling, though, has been misused in software testing and in test procedures as describe in this work, where it only should label the treated input factors as "independent". Although an "orthogonal test design" sounds elaborate, it is not to be advertised as "uniformly dispersed, neat and comparable, making it highly representative". A deletion of this text section is requested. Only data quality can judge whether a test procedure lives up to those high expectations. It is strongly suggested not to oversell used techniques. Although a deletion of "orthogonal" is favored, a clear statement of "orthogonal test design meaning changing four input parameters independently" or similar is favored.

Although some improvements in language and readability have been carried out, the text still is full of typos, especially tangled words are ubiquitous. A more careful proof

reading of any final submission is mandatory. Special attention should be given to consistent variable labelling, figure layouts, figure captions, page breaks, typos.

Overall the presented work still needs major refurbishments in order to be eligible for publication.
* * *
[Figure]

**Fig. 1.** Revised Figure 8 of the manuscript.

---

## Author Comment (AC3) · 29 Mar 2018

The paper has been significantly improved considering the suggestions from the reviewer and the questions have been properly answered. Therefore, it is suggested to accept the paper for publication falls the following minor points are corrected (minor revision): (1) There are still many typo errors which might due to that the authors directly copied the improved manuscript form Word to LaTeX. Most of the time the neighboring words are connected together, or a blank space between words is missing, or the font size is not right. The authors should check very carefully the following (but may not limited to) lines and make corrections: 38-41, 56-60, 68-72, 75, 81, 85-89, 95, 107, 120,

123, 138-141, 158-159, 168-173, 179, 197, 222, 226-232, 238, 252, 276-281, 290, 300-301, 309, 321-322, 334, 342-344, 351, 359, 362, 366-376, 390, 396, 402-405.

AC: Thanks for the reviewer's suggestion. I have readjusted the manuscript form carefully. I am so sorry about this mistake due to the version difference.

(2) In line 158, it should be 'spherical blocks with diameters of 4 cm, 6 cm, . . .'.

AC: Thanks for the reviewer's suggestion. I have revised the sentence.

The spherical blocks with radii of 2 cm, 3 cm, 4 cm and 5 cm (Figure 2) are made to simulate rockfall.

(3) In Tables 1-3, the units for the parameters should not be italic or bold.

AC: Thanks for the reviewer's suggestion. I have revised the units for parameters in Tables 1-3.

(4) It is suggested the authors to check the Eqn. (6) and the related text contents whether the parameters and subscripts are correctly written.

AC: Thanks for the reviewer's suggestion. I have revised the Eqn. (6) and the related text contents. The location of $R_y$ is at Figure 11, I have moved the representation to the proper position. I have replaced the '4' with 'Number of levels' to facilitate the readers' understanding.

(In the manuscript) The analysis method used to optimize the calculation results and the optimization process is shown in Figure 11 , and $R_y$ is the range of factory.

Fig.11 Flow chart for the optimization analysis of the test (See attachment)

The four parameters, rockfall block radius, r, movement height, H, cushion thickness, h, and particle size, d, belong to the factor set x∈(A, B, C, D), and the number of levels for all factors is four. The statistical test parameter under level y of factor set x can be calculated by determining $K_{xy}$ (x=A, B, C, D; y=1, 2, 3, 4), i.e., the sum of all the test

result indices Pxy containing level y of factor x, and dividing it by the total number of levels to obtain the average value kxy in which Pxy is the random variable of the normal distribution:

Formula (6) (See attachment)

where Kxy is the statistical parameter of factor x at level y, kxy is the average value of Kxy, and Ny is the number of levels.

(5) At proper places one can shortly address why the authors use spherical instead of non-spherical blocks for tests. A short comment extracted from the text already given by the authors in the answer to reviewer would be good.

AC: Thanks for the reviewer's suggestion. I have supplemented the reason why the authors use spherical instead of non-spherical blocks for tests in second paragraph of 'Experimental Studies' section.

Compared with the non-spherical blocks, spherical blocks with same quality are relatively difficult to be resisted by the same control methods through a large number of tests, spherical blocks presented higher and more consistent COR values compared to cubical blocks. (Asteriou et al, 2016). A phenomenon was also reported that tabular shaped rocks gradually become rounded and wheel-like due to sharp corners breaking off during the descent (Leine et al., 2014). If the designed cushion can resist the spherical rocks, and it also can effectively resist the non-spherical rocks. When designing the protective cushion, the serious conditions of spherical rocks should be considered to ensure fully the safety of worker.

(6) The style of the references is not kept the same. Please very carefully check the references one by one. Attention the typos which are similar to the comment (1).

AC: Thanks for the reviewer's suggestion. I have revised fully the style of the references according to the requirements of Journal.

(7) It is suggested to provide the three tables ('The experimental parameters of the

first group of tests', 'The experimental parameters of the second group of tests', and 'Orthogonal test results with the uncertainties') as supplemental material for the paper. Both the average value and the standard deviation should be given in these tables, if it is not ideal to plot the uncertainties in the Figures 8-10.

AC: Thanks for the reviewer's suggestion. I have calculated the standard deviation of test data in the Figures 8-10 and Table 2, please check the attachment, I have redrawn the Figure 8 with the error bar (Mean $\pm$ SD) as an example (See attachment). However, the Figure 9 and Figure 10 include too many curves, if I redraw each curves with the error bar, the Figure 9 and Figure 10 will be confusing and Intricate, thus I will added the standard deviation for three test results of the same experiment as the supplemental material for the paper.

(8) Please check the style of the variable names used in the whole text, including figures and tables. Sometimes they are italic, sometimes not. It is better to keep the style consistent.

AC: Thanks for the reviewer's suggestion. I have revised the style of the variable names to keep the consistency.

Please also note the supplement to this comment:
https://www.nat-hazards-earth-syst-sci-discuss.net/nhess-2018-16/nhess-2018-16-AC3-supplement.pdf

**Supplement:**

$$k_{xy} = \frac{K_{xy}}{N_y} = \sum P_{xy} \Big/ N_y \; , \; (6)$$

where $K_{xy}$ is the statistical parameter of factor $x$ at level $y$, $k_{xy}$ is the average value of $K_{xy}$, and $N_y$ is the number of levels.

[Figure]

Fig. 8 The *COR* (Mean ± SD) of block collisions with the plate. (Error bars: one standard deviation)

[Figure]

Fig.11 Flow chart for the optimization analysis of the test

Table 1 the experimental parameters of the first group of tests

| | | h(cm) \ d(mm) | 2mm | 6mm | 10mm | 14mm | 18mm | 24mm |
|---|---|---|---|---|---|---|---|---|
| The first group of tests (movement height H=1.2m) | R=2cm | 2cm | | | | | | |
| | | 4cm | | | | | | |
| | | 6cm | | | | | | |
| | | 8cm | | | | | | |
| | | 10cm | | | | | | |
| | | 12cm | | | | | | |
| | | 14cm | | | | | | |
| | R=3cm | h(cm) \ d(mm) | 2mm | 6mm | 10mm | 14mm | 18mm | 24mm |
| | | 2cm | | | | | | |
| | | 4cm | | | | | | |
| | | 6cm | | | | | | |
| | | 8cm | | | | | | |
| | | 10cm | | | | | | |
| | | 12cm | | | | | | |
| | | 14cm | | | | | | |
| | R=4cm | h(cm) \ d(mm) | 2mm | 6mm | 10mm | 14mm | 18mm | 24mm |
| | | 2cm | | | | | | |
| | | 4cm | | | | | | |
| | | 6cm | | | | | | |
| | | 8cm | | | | | | |
| | | 10cm | | | | | | |
| | | 12cm | | | | | | |
| | | 14cm | | | | | | |
| | R=5cm | h(cm) \ d(mm) | 2mm | 6mm | 10mm | 14mm | 18mm | 24mm |

| | | | 2cm | | | | | |
|---|---|---|---|---|---|---|---|---|
| | | | 4cm | | | | | |
| | | | 6cm | | | | | |
| | | | 8cm | | | | | |
| | | | 10cm | | | | | |
| | | | 12cm | | | | | |
| | | | 14cm | | | | | |

Table 2 the experimental parameters of the second group of tests

| | | H(m) \ d(mm) | 2mm | 6mm | 10mm | 14mm | 18mm | 24mm |
|---|---|---|---|---|---|---|---|---|
| The second group of tests (chushion thickness h=8cm) | R=2cm | 0.4m | | | | | | |
| | | 0.8m | | | | | | |
| | | 1.2m | | | | | | |
| | | 1.6m | | | | | | |
| | R=3cm | H(m) \ d(mm) | 2mm | 6mm | 10mm | 14mm | 18mm | 24mm |
| | | 0.4m | | | | | | |
| | | 0.8m | | | | | | |
| | | 1.2m | | | | | | |
| | | 1.6m | | | | | | |
| | R=4cm | H(m) \ d(mm) | 2mm | 6mm | 10mm | 14mm | 18mm | 24mm |
| | | 0.4m | | | | | | |
| | | 0.8m | | | | | | |
| | | 1.2m | | | | | | |
| | | 1.6m | | | | | | |
| | R=5cm | H(m) \ d(mm) | 2mm | 6mm | 10mm | 14mm | 18mm | 24mm |
| | | 0.4m | | | | | | |
| | | 0.8m | | | | | | |
| | | 1.2m | | | | | | |
| | | 1.6m | | | | | | |

Table 3 The COR of block collisions with the plate

| H=1.2m ,h=0cm,d=0mm | | | | | Average Value | standard deviation |
|---|---|---|---|---|---|---|
| r=2cm | | 0.363 | 0.368 | 0.421 | 0.384 | 0.032 |
| r=3cm | | 0.398 | 0.428 | 0.437 | 0.421 | 0.020 |
| r=4cm | | 0.386 | 0.482 | 0.443 | 0.437 | 0.048 |
| r=5cm | | 0.403 | 0.458 | 0.471 | 0.444 | 0.036 |

Table 4 Comparison of the COR of different blocks released from a height of 1.2m

(Standard deviation and average value of COR is calculated by the middle three values for cushion particle sizes of 1.8 cm and 2.4 cm)

| H=1.2m,r=2cm,h=2cm | | | | | | Average Value | standard deviation |
|---|---|---|---|---|---|---|---|
| d=2mm | | 0.310 | 0.329 | 0.339 | | 0.326 | 0.326 |
| d=6mm | | 0.298 | 0.348 | 0.35 | | 0.332 | 0.332 |
| d=10mm | | 0.319 | 0.343 | 0.376 | | 0.346 | 0.346 |
| d=14mm | | 0.314 | 0.344 | 0.371 | | 0.343 | 0.343 |
| d=18mm | 0.262 | 0.291 | 0.337 | 0.416 | 0.443 | 0.348 | 0.348 |

| d=24mm | 0.234 | 0.288 | 0.371 | 0.403 | 0.421 | 0.354 | 0.354 |
|---|---|---|---|---|---|---|---|

| H=1.2m,r=2cm,h=4cm | | | | | Average Value | standard deviation |
|---|---|---|---|---|---|---|
| d=2mm | | 0.281 | 0.285 | 0.316 | | 0.294 | 0.019 |
| d=6mm | | 0.292 | 0.338 | 0.345 | | 0.325 | 0.029 |
| d=10mm | | 0.261 | 0.314 | 0.331 | | 0.302 | 0.037 |
| d=14mm | | 0.285 | 0.324 | 0.360 | | 0.323 | 0.038 |
| d=18mm | 0.215 | 0.252 | 0.323 | 0.376 | 0.386 | 0.317 | 0.062 |
| d=24mm | 0.267 | 0.270 | 0.304 | 0.362 | 0.403 | 0.312 | 0.047 |

| H=1.2m,r=2cm,h=6cm | | | | | Average Value | standard deviation |
|---|---|---|---|---|---|---|
| d=2mm | | 0.240 | 0.267 | 0.270 | | 0.259 | 0.017 |
| d=6mm | | 0.239 | 0.277 | 0.306 | | 0.274 | 0.034 |
| d=10mm | | 0.246 | 0.283 | 0.317 | | 0.282 | 0.036 |
| d=14mm | | 0.251 | 0.267 | 0.331 | | 0.283 | 0.042 |
| d=18mm | 0.231 | 0.259 | 0.299 | 0.345 | 0.382 | 0.301 | 0.043 |
| d=24mm | 0.226 | 0.261 | 0.291 | 0.336 | 0.370 | 0.296 | 0.038 |

| H=1.2m,r=2cm,h=8cm | | | | | Average Value | standard deviation |
|---|---|---|---|---|---|---|
| d=2mm | | 0.215 | 0.243 | 0.271 | | 0.243 | 0.028 |
| d=6mm | | 0.211 | 0.261 | 0.290 | | 0.254 | 0.040 |
| d=10mm | | 0.216 | 0.261 | 0.312 | | 0.263 | 0.048 |
| d=14mm | | 0.223 | 0.286 | 0.304 | | 0.271 | 0.043 |
| d=18mm | 0.205 | 0.225 | 0.285 | 0.321 | 0.356 | 0.277 | 0.048 |
| d=24mm | 0.193 | 0.206 | 0.292 | 0.354 | 0.371 | 0.284 | 0.074 |

| H=1.2m,r=2cm,h=10cm | | | | | Average Value | standard deviation |
|---|---|---|---|---|---|---|
| d=2mm | | 0.201 | 0.245 | 0.277 | | 0.241 | 0.038 |
| d=6mm | | 0.198 | 0.250 | 0.293 | | 0.247 | 0.048 |
| d=10mm | | 0.226 | 0.251 | 0.288 | | 0.255 | 0.031 |
| d=14mm | | 0.210 | 0.253 | 0.311 | | 0.258 | 0.051 |
| d=18mm | 0.190 | 0.192 | 0.273 | 0.327 | 0.363 | 0.264 | 0.068 |
| d=24mm | 0.204 | 0.214 | 0.292 | 0.325 | 0.350 | 0.277 | 0.057 |

| H=1.2m,r=2cm,h=12cm | | | | | Average Value | standard deviation |
|---|---|---|---|---|---|---|
| d=2mm | | 0.207 | 0.218 | 0.259 | | 0.228 | 0.027 |
| d=6mm | | 0.190 | 0.236 | 0.273 | | 0.233 | 0.042 |
| d=10mm | | 0.214 | 0.225 | 0.302 | | 0.247 | 0.048 |

| d=14mm | | 0.200 | 0.243 | 0.313 | | 0.252 | 0.057 |
|---|---|---|---|---|---|---|---|
| d=18mm | 0.189 | 0.198 | 0.236 | 0.319 | 0.328 | 0.251 | 0.062 |
| d=24mm | 0.198 | 0.221 | 0.251 | 0.326 | 0.343 | 0.266 | 0.054 |

| H=1.2m,r=2cm,h=14cm | | | | | Average Value | standard deviation |
|---|---|---|---|---|---|---|
| d=2mm | | 0.184 | 0.230 | 0.246 | | 0.22 | 0.032 |
| d=6mm | | 0.199 | 0.214 | 0.283 | | 0.232 | 0.045 |
| d=10mm | | 0.204 | 0.251 | 0.265 | | 0.24 | 0.032 |
| d=14mm | | 0.183 | 0.224 | 0.301 | | 0.236 | 0.060 |
| d=18mm | 0.162 | 0.196 | 0.260 | 0.291 | 0.314 | 0.249 | 0.048 |
| d=24mm | 0.194 | 0.208 | 0.250 | 0.316 | 0.353 | 0.258 | 0.054 |

| H=1.2m,r=3cm,h=2cm | | | | | Average Value | standard deviation |
|---|---|---|---|---|---|---|
| d=2mm | | 0.313 | 0.338 | 0.351 | | 0.334 | 0.019 |
| d=6mm | | 0.326 | 0.348 | 0.349 | | 0.341 | 0.013 |
| d=10mm | | 0.308 | 0.354 | 0.379 | | 0.347 | 0.036 |
| d=14mm | | 0.311 | 0.342 | 0.409 | | 0.354 | 0.050 |
| d=18mm | 0.261 | 0.333 | 0.336 | 0.387 | 0.396 | 0.352 | 0.030 |
| d=24mm | 0.256 | 0.322 | 0.369 | 0.413 | 0.420 | 0.368 | 0.046 |

| H=1.2m,r=3cm,h=4cm | | | | | Average Value | standard deviation |
|---|---|---|---|---|---|---|
| d=2mm | | 0.261 | 0.316 | 0.329 | | 0.302 | 0.036 |
| d=6mm | | 0.276 | 0.309 | 0.360 | | 0.315 | 0.042 |
| d=10mm | | 0.267 | 0.329 | 0.352 | | 0.316 | 0.044 |
| d=14mm | | 0.282 | 0.320 | 0.379 | | 0.327 | 0.049 |
| d=18mm | 0.245 | 0.298 | 0.314 | 0.366 | 0.381 | 0.326 | 0.036 |
| d=24mm | 0.253 | 0.276 | 0.321 | 0.405 | 0.415 | 0.334 | 0.065 |

| H=1.2m,r=3cm,h=6cm | | | | | Average Value | standard deviation |
|---|---|---|---|---|---|---|
| d=2mm | | 0.254 | 0.273 | 0.304 | | 0.277 | 0.025 |
| d=6mm | | 0.262 | 0.281 | 0.309 | | 0.284 | 0.024 |
| d=10mm | | 0.255 | 0.288 | 0.321 | | 0.288 | 0.033 |
| d=14mm | | 0.283 | 0.311 | 0.360 | | 0.318 | 0.039 |
| d=18mm | 0.185 | 0.261 | 0.300 | 0.366 | 0.384 | 0.309 | 0.053 |
| d=24mm | 0.248 | 0.248 | 0.337 | 0.390 | 0.393 | 0.325 | 0.072 |

| H=1.2m,r=3cm,h=8cm | | | | | Average Value | standard deviation |
|---|---|---|---|---|---|---|
| d=2mm | | 0.229 | 0.235 | 0.277 | | 0.247 | 0.026 |

| d=6mm |  | 0.213 | 0.270 | 0.303 |  | 0.262 | 0.046 |
|---|---|---|---|---|---|---|---|
| d=10mm |  | 0.241 | 0.247 | 0.313 |  | 0.267 | 0.040 |
| d=14mm |  | 0.223 | 0.264 | 0.332 |  | 0.273 | 0.055 |
| d=18mm | 0.214 | 0.220 | 0.303 | 0.320 | 0.341 | 0.281 | 0.054 |
| d=24mm | 0.235 | 0.261 | 0.292 | 0.323 | 0.354 | 0.292 | 0.031 |

| H=1.2m,r=3cm,h=10cm |  |  |  |  |  | Average Value | standard deviation |
|---|---|---|---|---|---|---|---|
| d=2mm |  | 0.212 | 0.233 | 0.266 |  | 0.237 | 0.027 |
| d=6mm |  | 0.223 | 0.240 | 0.275 |  | 0.246 | 0.027 |
| d=10mm |  | 0.228 | 0.246 | 0.288 |  | 0.254 | 0.031 |
| d=14mm |  | 0.213 | 0.271 | 0.302 |  | 0.262 | 0.045 |
| d=18mm | 0.192 | 0.211 | 0.251 | 0.309 | 0.324 | 0.257 | 0.049 |
| d=24mm | 0.211 | 0.232 | 0.246 | 0.326 | 0.341 | 0.268 | 0.051 |

| H=1.2m,r=3cm,h=12cm |  |  |  |  |  | Average Value | standard deviation |
|---|---|---|---|---|---|---|---|
| d=2mm |  | 0.194 | 0.220 | 0.264 |  | 0.226 | 0.035 |
| d=6mm |  | 0.199 | 0.231 | 0.287 |  | 0.239 | 0.045 |
| d=10mm |  | 0.228 | 0.234 | 0.264 |  | 0.242 | 0.019 |
| d=14mm |  | 0.204 | 0.256 | 0.284 |  | 0.248 | 0.041 |
| d=18mm | 0.213 | 0.228 | 0.242 | 0.295 | 0.320 | 0.255 | 0.035 |
| d=24mm | 0.186 | 0.217 | 0.259 | 0.301 | 0.336 | 0.259 | 0.042 |

| H=1.2m,r=3cm,h=14cm |  |  |  |  |  | Average Value | standard deviation |
|---|---|---|---|---|---|---|---|
| d=2mm |  | 0.172 | 0.206 | 0.276 |  | 0.218 | 0.053 |
| d=6mm |  | 0.203 | 0.215 | 0.254 |  | 0.224 | 0.027 |
| d=10mm |  | 0.188 | 0.224 | 0.275 |  | 0.229 | 0.044 |
| d=14mm |  | 0.178 | 0.229 | 0.286 |  | 0.231 | 0.054 |
| d=18mm | 0.186 | 0.192 | 0.244 | 0.302 | 0.335 | 0.246 | 0.055 |
| d=24mm | 0.191 | 0.228 | 0.246 | 0.312 | 0.343 | 0.262 | 0.044 |

| H=1.2m,r=4cm,h=2cm |  |  |  |  |  | Average Value | standard deviation |
|---|---|---|---|---|---|---|---|
| d=2mm |  | 0.316 | 0.338 | 0.354 |  | 0.336 | 0.019 |
| d=6mm |  | 0.329 | 0.343 | 0.372 |  | 0.348 | 0.022 |
| d=10mm |  | 0.333 | 0.351 | 0.384 |  | 0.356 | 0.026 |
| d=14mm |  | 0.321 | 0.358 | 0.416 |  | 0.365 | 0.048 |
| d=18mm | 0.285 | 0.329 | 0.371 | 0.401 | 0.411 | 0.367 | 0.036 |
| d=24mm | 0.313 | 0.334 | 0.369 | 0.413 | 0.432 | 0.372 | 0.040 |

| H=1.2m,r=4cm,h=4cm | | | | | Average Value | standard deviation |
|---|---|---|---|---|---|---|
| d=2mm | | 0.284 | 0.307 | 0.336 | | 0.309 | 0.026 |
| d=6mm | | 0.298 | 0.319 | 0.346 | | 0.321 | 0.024 |
| d=10mm | | 0.288 | 0.310 | 0.347 | | 0.315 | 0.030 |
| d=14mm | | 0.308 | 0.316 | 0.351 | | 0.325 | 0.023 |
| d=18mm | 0.263 | 0.295 | 0.339 | 0.368 | 0.374 | 0.334 | 0.037 |
| d=24mm | 0.275 | 0.309 | 0.326 | 0.394 | 0.410 | 0.343 | 0.045 |

| H=1.2m,r=4cm,h=6cm | | | | | Average Value | standard deviation |
|---|---|---|---|---|---|---|
| d=2mm | | 0.267 | 0.279 | 0.294 | | 0.28 | 0.014 |
| d=6mm | | 0.294 | 0.304 | 0.329 | | 0.309 | 0.018 |
| d=10mm | | 0.273 | 0.286 | 0.317 | | 0.292 | 0.023 |
| d=14mm | | 0.280 | 0.292 | 0.304 | | 0.292 | 0.012 |
| d=18mm | 0.257 | 0.282 | 0.303 | 0.351 | 0.384 | 0.312 | 0.035 |
| d=24mm | 0.240 | 0.294 | 0.322 | 0.359 | 0.367 | 0.325 | 0.033 |

| H=1.2m,r=4cm,h=8cm | | | | | Average Value | standard deviation |
|---|---|---|---|---|---|---|
| d=2mm | | 0.249 | 0.250 | 0.269 | | 0.256 | 0.011 |
| d=6mm | | 0.250 | 0.267 | 0.296 | | 0.271 | 0.023 |
| d=10mm | | 0.248 | 0.274 | 0.306 | | 0.276 | 0.029 |
| d=14mm | | 0.252 | 0.271 | 0.299 | | 0.274 | 0.024 |
| d=18mm | 0.235 | 0.263 | 0.291 | 0.325 | 0.344 | 0.293 | 0.031 |
| d=24mm | 0.259 | 0.267 | 0.298 | 0.341 | 0.367 | 0.302 | 0.037 |

| H=1.2m,r=4cm,h=10cm | | | | | Average Value | standard deviation |
|---|---|---|---|---|---|---|
| d=2mm | | 0.239 | 0.249 | 0.268 | | 0.252 | 0.015 |
| d=6mm | | 0.235 | 0.261 | 0.278 | | 0.258 | 0.022 |
| d=10mm | | 0.247 | 0.264 | 0.296 | | 0.269 | 0.025 |
| d=14mm | | 0.254 | 0.249 | 0.292 | | 0.265 | 0.024 |
| d=18mm | 0.226 | 0.243 | 0.276 | 0.324 | 0.331 | 0.281 | 0.041 |
| d=24mm | 0.231 | 0.236 | 0.277 | 0.321 | 0.354 | 0.278 | 0.043 |

| H=1.2m,r=4cm,h=12cm | | | | | Average Value | standard deviation |
|---|---|---|---|---|---|---|
| d=2mm | | 0.229 | 0.231 | 0.248 | | 0.236 | 0.010 |
| d=6mm | | 0.221 | 0.244 | 0.270 | | 0.245 | 0.025 |
| d=10mm | | 0.216 | 0.227 | 0.268 | | 0.237 | 0.027 |
| d=14mm | | 0.207 | 0.239 | 0.283 | | 0.243 | 0.038 |
| d=18mm | 0.186 | 0.208 | 0.251 | 0.297 | 0.319 | 0.252 | 0.045 |

| d=24mm | 0.217 | 0.218 | 0.272 | 0.284 | 0.326 | 0.258 | 0.035 |
|---|---|---|---|---|---|---|---|

| H=1.2m,r=4cm,h=14cm | | | | | Average Value | standard deviation |
|---|---|---|---|---|---|---|
| d=2mm | | 0.217 | 0.218 | 0.237 | | 0.224 | 0.011 |
| d=6mm | | 0.215 | 0.231 | 0.259 | | 0.235 | 0.022 |
| d=10mm | | 0.196 | 0.229 | 0.271 | | 0.232 | 0.038 |
| d=14mm | | 0.219 | 0.224 | 0.268 | | 0.237 | 0.027 |
| d=18mm | 0.181 | 0.207 | 0.256 | 0.281 | 0.314 | 0.248 | 0.038 |
| d=24mm | 0.196 | 0.219 | 0.247 | 0.293 | 0.335 | 0.253 | 0.037 |

| H=1.2m,r=5cm,h=2cm | | | | | Average Value | standard deviation |
|---|---|---|---|---|---|---|
| d=2mm | | 0.328 | 0.336 | 0.356 | | 0.34 | 0.014 |
| d=6mm | | 0.320 | 0.342 | 0.364 | | 0.342 | 0.022 |
| d=10mm | | 0.324 | 0.351 | 0.393 | | 0.356 | 0.035 |
| d=14mm | | 0.342 | 0.364 | 0.398 | | 0.368 | 0.028 |
| d=18mm | 0.317 | 0.338 | 0.374 | 0.401 | 0.426 | 0.371 | 0.032 |
| d=24mm | 0.302 | 0.354 | 0.365 | 0.421 | 0.437 | 0.38 | 0.036 |

| H=1.2m,r=5cm,h=4cm | | | | | Average Value | standard deviation |
|---|---|---|---|---|---|---|
| d=2mm | | 0.316 | 0.317 | 0.339 | | 0.324 | 0.013 |
| d=6mm | | 0.297 | 0.306 | 0.330 | | 0.311 | 0.017 |
| d=10mm | | 0.294 | 0.321 | 0.354 | | 0.323 | 0.030 |
| d=14mm | | 0.318 | 0.340 | 0.374 | | 0.344 | 0.028 |
| d=18mm | 0.318 | 0.321 | 0.322 | 0.386 | 0.397 | 0.343 | 0.037 |
| d=24mm | 0.284 | 0.328 | 0.355 | 0.373 | 0.403 | 0.352 | 0.023 |

| H=1.2m,r=5cm,h=6cm | | | | | Average Value | standard deviation |
|---|---|---|---|---|---|---|
| d=2mm | | 0.286 | 0.286 | 0.301 | | 0.291 | 0.009 |
| d=6mm | | 0.274 | 0.287 | 0.315 | | 0.292 | 0.021 |
| d=10mm | | 0.301 | 0.324 | 0.329 | | 0.318 | 0.015 |
| d=14mm | | 0.291 | 0.298 | 0.338 | | 0.309 | 0.025 |
| d=18mm | 0.244 | 0.286 | 0.314 | 0.378 | 0.398 | 0.326 | 0.047 |
| d=24mm | 0.270 | 0.293 | 0.316 | 0.381 | 0.401 | 0.33 | 0.046 |

| H=1.2m,r=5cm,h=8cm | | | | | Average Value | standard deviation |
|---|---|---|---|---|---|---|
| d=2mm | | 0.254 | 0.262 | 0.279 | | 0.265 | 0.013 |
| d=6mm | | 0.270 | 0.276 | 0.294 | | 0.28 | 0.012 |
| d=10mm | | 0.261 | 0.294 | 0.309 | | 0.288 | 0.025 |

| | | | | | | Average Value | standard deviation |
|---|---|---|---|---|---|---|---|
| d=14mm | | 0.272 | 0.283 | 0.324 | | 0.293 | 0.027 |
| d=18mm | 0.226 | 0.256 | 0.294 | 0.356 | 0.362 | 0.302 | 0.050 |
| d=24mm | 0.234 | 0.286 | 0.291 | 0.362 | 0.374 | 0.313 | 0.043 |

| H=1.2m,r=5cm,h=10cm | | | | | | Average Value | standard deviation |
|---|---|---|---|---|---|---|---|
| d=2mm | | 0.247 | 0.261 | 0.281 | | 0.263 | 0.017 |
| d=6mm | | 0.233 | 0.273 | 0.289 | | 0.265 | 0.029 |
| d=10mm | | 0.249 | 0.257 | 0.301 | | 0.269 | 0.028 |
| d=14mm | | 0.250 | 0.268 | 0.298 | | 0.272 | 0.024 |
| d=18mm | 0.208 | 0.237 | 0.261 | 0.315 | 0.320 | 0.271 | 0.040 |
| d=24mm | 0.221 | 0.255 | 0.273 | 0.336 | 0.373 | 0.288 | 0.043 |

| H=1.2m,r=5cm,h=12cm | | | | | | Average Value | standard deviation |
|---|---|---|---|---|---|---|---|
| d=2mm | | 0.230 | 0.236 | 0.254 | | 0.24 | 0.012 |
| d=6mm | | 0.213 | 0.251 | 0.265 | | 0.243 | 0.027 |
| d=10mm | | 0.220 | 0.245 | 0.291 | | 0.252 | 0.036 |
| d=14mm | | 0.242 | 0.240 | 0.289 | | 0.257 | 0.028 |
| d=18mm | 0.197 | 0.224 | 0.242 | 0.311 | 0.342 | 0.259 | 0.046 |
| d=24mm | 0.208 | 0.209 | 0.261 | 0.328 | 0.337 | 0.266 | 0.060 |

| H=1.2m,r=5cm,h=14cm | | | | | | Average Value | standard deviation |
|---|---|---|---|---|---|---|---|
| d=2mm | | 0.207 | 0.217 | 0.236 | | 0.22 | 0.015 |
| d=6mm | | 0.204 | 0.229 | 0.257 | | 0.23 | 0.027 |
| d=10mm | | 0.227 | 0.234 | 0.250 | | 0.237 | 0.012 |
| d=14mm | | 0.215 | 0.241 | 0.270 | | 0.242 | 0.028 |
| d=18mm | 0.184 | 0.189 | 0.235 | 0.278 | 0.304 | 0.234 | 0.045 |
| d=24mm | 0.201 | 0.218 | 0.258 | 0.286 | 0.316 | 0.254 | 0.034 |

Table 5 Comparison of the COR for different blocks colliding with an 8-cm thick cushion
(Standard deviation and average value of COR is calculated by the middle three
values for cushion particle sizes of 1.8 cm and 2.4 cm)

| h=8cm,r=2cm,H=0.4m | | | | | Average Value | standard deviation |
|---|---|---|---|---|---|---|
| d=2mm | | 0.197 | 0.214 | 0.237 | | 0.216 | 0.020 |
| d=6mm | | 0.221 | 0.222 | 0.241 | | 0.228 | 0.011 |
| d=10mm | | 0.216 | 0.228 | 0.264 | | 0.236 | 0.025 |
| d=14mm | | 0.223 | 0.256 | 0.283 | | 0.254 | 0.030 |
| d=18mm | 0.204 | 0.201 | 0.261 | 0.306 | 0.311 | 0.256 | 0.053 |
| d=24mm | 0.203 | 0.221 | 0.264 | 0.295 | 0.316 | 0.260 | 0.037 |

| h=8cm,r=2cm,H=0.8m | | | | | Average Value | standard deviation |
|---|---|---|---|---|---|---|
| d=2mm | | 0.219 | 0.231 | 0.237 | | 0.229 | 0.009 |
| d=6mm | | 0.200 | 0.248 | 0.254 | | 0.234 | 0.030 |
| d=10mm | | 0.220 | 0.242 | 0.273 | | 0.245 | 0.027 |
| d=14mm | | 0.213 | 0.246 | 0.270 | | 0.243 | 0.029 |
| d=18mm | 0.214 | 0.221 | 0.274 | 0.291 | 0.327 | 0.262 | 0.037 |
| d=24mm | 0.191 | 0.212 | 0.271 | 0.318 | 0.331 | 0.267 | 0.053 |

| h=8cm,r=2cm,H=1.2m | | | | | Average Value | standard deviation |
|---|---|---|---|---|---|---|
| d=2mm | | 0.228 | 0.236 | 0.265 | | 0.243 | 0.019 |
| d=6mm | | 0.217 | 0.267 | 0.278 | | 0.254 | 0.033 |
| d=10mm | | 0.231 | 0.262 | 0.296 | | 0.263 | 0.033 |
| d=14mm | | 0.222 | 0.283 | 0.308 | | 0.271 | 0.044 |
| d=18mm | 0.207 | 0.226 | 0.287 | 0.318 | 0.336 | 0.277 | 0.047 |
| d=24mm | 0.226 | 0.247 | 0.299 | 0.306 | 0.337 | 0.284 | 0.032 |

| h=8cm,r=2cm,H=1.6m | | | | | Average Value | standard deviation |
|---|---|---|---|---|---|---|
| d=2mm | | 0.232 | 0.239 | 0.258 | | 0.243 | 0.013 |
| d=6mm | | 0.240 | 0.243 | 0.273 | | 0.252 | 0.018 |
| d=10mm | | 0.223 | 0.294 | 0.296 | | 0.271 | 0.042 |
| d=14mm | | 0.245 | 0.287 | 0.338 | | 0.290 | 0.047 |
| d=18mm | 0.221 | 0.249 | 0.279 | 0.321 | 0.332 | 0.283 | 0.036 |
| d=24mm | 0.218 | 0.234 | 0.276 | 0.336 | 0.359 | 0.282 | 0.051 |

| h=8cm,r=3cm,H=0.4m | | | | | Average Value | standard deviation |
|---|---|---|---|---|---|---|
| d=2mm | | 0.208 | 0.227 | 0.237 | | 0.224 | 0.015 |
| d=6mm | | 0.212 | 0.226 | 0.255 | | 0.231 | 0.022 |
| d=10mm | | 0.228 | 0.231 | 0.270 | | 0.243 | 0.023 |
| d=14mm | | 0.216 | 0.251 | 0.289 | | 0.252 | 0.037 |
| d=18mm | 0.186 | 0.222 | 0.267 | 0.306 | 0.316 | 0.265 | 0.042 |

| d=24mm | 0.179 | 0.211 | 0.272 | 0.321 | 0.331 | 0.268 | 0.055 |

| h=8cm,r=3cm,H=0.8m | | | | | Average Value | standard deviation |
|---|---|---|---|---|---|---|
| d=2mm | | 0.222 | 0.235 | 0.251 | | 0.236 | 0.015 |
| d=6mm | | 0.218 | 0.247 | 0.264 | | 0.243 | 0.023 |
| d=10mm | | 0.235 | 0.251 | 0.306 | | 0.264 | 0.037 |
| d=14mm | | 0.220 | 0.274 | 0.292 | | 0.262 | 0.037 |
| d=18mm | 0.231 | 0.242 | 0.254 | 0.305 | 0.329 | 0.267 | 0.033 |
| d=24mm | 0.224 | 0.234 | 0.270 | 0.324 | 0.354 | 0.276 | 0.045 |

| h=8cm,r=3cm,H=1.2m | | | | | Average Value | standard deviation |
|---|---|---|---|---|---|---|
| d=2mm | | 0.229 | 0.243 | 0.269 | | 0.247 | 0.020 |
| d=6mm | | 0.241 | 0.265 | 0.28 | | 0.262 | 0.020 |
| d=10mm | | 0.234 | 0.27 | 0.297 | | 0.267 | 0.032 |
| d=14mm | | 0.221 | 0.288 | 0.310 | | 0.273 | 0.046 |
| d=18mm | 0.198 | 0.239 | 0.283 | 0.321 | 0.334 | 0.281 | 0.041 |
| d=24mm | 0.227 | 0.251 | 0.287 | 0.338 | 0.356 | 0.292 | 0.044 |

| h=8cm,r=3cm,H=1.6m | | | | | Average Value | standard deviation |
|---|---|---|---|---|---|---|
| d=2mm | | 0.242 | 0.251 | 0.269 | | 0.254 | 0.014 |
| d=6mm | | 0.228 | 0.279 | 0.288 | | 0.265 | 0.032 |
| d=10mm | | 0.261 | 0.284 | 0.313 | | 0.286 | 0.026 |
| d=14mm | | 0.256 | 0.286 | 0.325 | | 0.289 | 0.035 |
| d=18mm | 0.262 | 0.276 | 0.291 | 0.312 | 0.323 | 0.293 | 0.018 |
| d=24mm | 0.214 | 0.272 | 0.295 | 0.336 | 0.351 | 0.301 | 0.032 |

| h=8cm,r=4cm,H=0.4m | | | | | Average Value | standard deviation |
|---|---|---|---|---|---|---|
| d=2mm | | 0.220 | 0.227 | 0.246 | | 0.231 | 0.013 |
| d=6mm | | 0.233 | 0.234 | 0.259 | | 0.242 | 0.015 |
| d=10mm | | 0.212 | 0.241 | 0.264 | | 0.239 | 0.026 |
| d=14mm | | 0.241 | 0.252 | 0.299 | | 0.264 | 0.031 |
| d=18mm | 0.209 | 0.239 | 0.253 | 0.294 | 0.321 | 0.262 | 0.029 |
| d=24mm | 0.213 | 0.236 | 0.278 | 0.314 | 0.316 | 0.276 | 0.039 |

| h=8cm,r=4cm,H=0.8m | | | | | Average Value | standard deviation |
|---|---|---|---|---|---|---|
| d=2mm | | 0.223 | 0.247 | 0.265 | | 0.245 | 0.021 |
| d=6mm | | 0.248 | 0.252 | 0.271 | | 0.257 | 0.012 |
| d=10mm | | 0.236 | 0.257 | 0.293 | | 0.262 | 0.029 |

| d=14mm |       | 0.260 | 0.285 | 0.316 |       | 0.287 | 0.028 |
|--------|-------|-------|-------|-------|-------|-------|-------|
| d=18mm | 0.216 | 0.247 | 0.287 | 0.324 | 0.338 | 0.286 | 0.039 |
| d=24mm | 0.224 | 0.230 | 0.302 | 0.338 | 0.351 | 0.290 | 0.055 |

| h=8cm,r=4cm,H=1.2m | | | | | Average Value | standard deviation |
|--------|-------|-------|-------|-------|-------|-------|-------|
| d=2mm  |       | 0.245 | 0.254 | 0.269 |       | 0.256 | 0.012 |
| d=6mm  |       | 0.230 | 0.287 | 0.296 |       | 0.271 | 0.036 |
| d=10mm |       | 0.249 | 0.281 | 0.298 |       | 0.276 | 0.025 |
| d=14mm |       | 0.268 | 0.278 | 0.306 |       | 0.284 | 0.020 |
| d=18mm | 0.225 | 0.256 | 0.291 | 0.332 | 0.348 | 0.293 | 0.038 |
| d=24mm | 0.248 | 0.282 | 0.303 | 0.321 | 0.367 | 0.302 | 0.020 |

| h=8cm,r=4cm,H=1.6m | | | | | Average Value | standard deviation |
|--------|-------|-------|-------|-------|-------|-------|-------|
| d=2mm  |       | 0.243 | 0.257 | 0.283 |       | 0.261 | 0.020 |
| d=6mm  |       | 0.269 | 0.282 | 0.304 |       | 0.285 | 0.018 |
| d=10mm |       | 0.254 | 0.283 | 0.321 |       | 0.286 | 0.034 |
| d=14mm |       | 0.237 | 0.327 | 0.333 |       | 0.299 | 0.054 |
| d=18mm | 0.234 | 0.273 | 0.306 | 0.354 | 0.375 | 0.311 | 0.041 |
| d=24mm | 0.211 | 0.262 | 0.307 | 0.361 | 0.394 | 0.310 | 0.050 |

| h=8cm,r=5cm,H=0.4m | | | | | Average Value | standard deviation |
|--------|-------|-------|-------|-------|-------|-------|-------|
| d=2mm  |       | 0.229 | 0.231 | 0.248 |       | 0.236 | 0.010 |
| d=6mm  |       | 0.243 | 0.247 | 0.269 |       | 0.253 | 0.014 |
| d=10mm |       | 0.211 | 0.256 | 0.283 |       | 0.25  | 0.036 |
| d=14mm |       | 0.232 | 0.259 | 0.298 |       | 0.263 | 0.033 |
| d=18mm | 0.213 | 0.228 | 0.284 | 0.316 | 0.354 | 0.276 | 0.045 |
| d=24mm | 0.207 | 0.250 | 0.281 | 0.321 | 0.348 | 0.284 | 0.036 |

| h=8cm,r=5cm,H=0.8m | | | | | Average Value | standard deviation |
|--------|-------|-------|-------|-------|-------|-------|-------|
| d=2mm  |       | 0.239 | 0.246 | 0.271 |       | 0.252 | 0.017 |
| d=6mm  |       | 0.252 | 0.268 | 0.281 |       | 0.267 | 0.015 |
| d=10mm |       | 0.261 | 0.284 | 0.304 |       | 0.283 | 0.022 |
| d=14mm |       | 0.230 | 0.288 | 0.298 |       | 0.272 | 0.037 |
| d=18mm | 0.218 | 0.253 | 0.291 | 0.338 | 0.351 | 0.294 | 0.043 |
| d=24mm | 0.206 | 0.247 | 0.316 | 0.331 | 0.383 | 0.298 | 0.045 |

| h=8cm,r=5cm,H=1.2m | | | | | Average Value | standard deviation |
|--------|-------|-------|-------|-------|-------|-------|-------|
| d=2mm  |       | 0.258 | 0.259 | 0.278 |       | 0.265 | 0.011 |

| d=6mm | | 0.237 | 0.300 | 0.303 | | 0.28 | 0.037 |
|---|---|---|---|---|---|---|---|
| d=10mm | | 0.259 | 0.286 | 0.319 | | 0.288 | 0.030 |
| d=14mm | | 0.247 | 0.287 | 0.345 | | 0.293 | 0.049 |
| d=18mm | 0.251 | 0.266 | 0.298 | 0.342 | 0.354 | 0.302 | 0.038 |
| d=24mm | 0.236 | 0.272 | 0.306 | 0.361 | 0.381 | 0.313 | 0.045 |

| h=8cm,r=5cm,H=1.6m | | | | | Average Value | standard deviation |
|---|---|---|---|---|---|---|
| d=2mm | | 0.244 | 0.279 | 0.296 | | 0.273 | 0.027 |
| d=6mm | | 0.268 | 0.284 | 0.309 | | 0.287 | 0.021 |
| d=10mm | | 0.254 | 0.307 | 0.336 | | 0.299 | 0.042 |
| d=14mm | | 0.271 | 0.311 | 0.348 | | 0.31 | 0.039 |
| d=18mm | 0.246 | 0.258 | 0.307 | 0.359 | 0.387 | 0.308 | 0.051 |
| d=24mm | 0.225 | 0.279 | 0.338 | 0.349 | 0.368 | 0.322 | 0.038 |

Table 6 Orthogonal test results (COR/Damage depth L)

| r=2cm,H=0.4m,h=2cm,d=2mm | | | Average Value | standard deviation |
|---|---|---|---|---|
| 0.269/0.67 | 0.274/0.72 | 0.291/0.56 | 0.278/0.65 | 0.012/0.082 |
| r=2cm,H=0.8m,h=4cm,d=6mm | | | Average Value | |
| 0.253/0.79 | 0.268/0.75 | 0.298/0.68 | 0.273/0.74 | 0.023/0.056 |
| r=2cm,H=1.2m,h=6cm,d=10mm | | | Average Value | |
| 0.255/1.02 | 0.278/0.91 | 0.313/0.86 | 0.282/0.93 | 0.029/0.082 |
| r=2cm,H=1.6m,h=8cm,d=14mm | | | Average Value | |
| 0.266/1.10 | 0.298/1.01 | 0.321/1.04 | 0.295/1.05 | 0.028/0.046 |

| r=3cm,H=0.4m,h=2cm,d=6mm | | | Average Value | standard deviation |
|---|---|---|---|---|
| 0.287/0.60 | 0.287/0.62 | 0.308/0.52 | 0.294/0.58 | 0.012/0.053 |
| r=3cm,H=0.8m,h=4cm,d=2mm | | | Average Value | |
| 0.252/1.34 | 0.261/1.64 | 0.282/1.37 | 0.265/1.45 | 0.015/0.165 |
| r=3cm,H=1.2m,h=6cm,d=14mm | | | Average Value | |
| 0.275/1.17 | 0.319/1.08 | 0.357/0.84 | 0.317/1.03 | 0.041/0.171 |
| r=3cm,H=1.6m,h=8cm,d=10mm | | | Average Value | |
| 0.264/1.74 | 0.273/1.68 | 0.303/1.38 | 0.280/1.60 | 0.020/0.193 |

| r=4cm,H=0.4m,h=4cm,d=10mm | | | Average Value | standard deviation |
|---|---|---|---|---|
| 0.265/0.65 | 0.304/0.58 | 0.319/0.63 | 0.296/0.62 | 0.028/0.036 |
| r=4cm,H=0.8m,h=2cm,d=14mm | | | Average Value | |
| 0.304/0.51 | 0.354/0.68 | 0.356/0.49 | 0.338/0.56 | 0.029/0.104 |
| r=4cm,H=1.2m,h=8cm,d=2mm | | | Average Value | |

| | | | Average Value | |
|---|---|---|---|---|
| 0.232/2.79 | 0.261/2.76 | 0.275/2.25 | 0.256/2.60 | 0.022/0.303 |
| r=4cm,H=1.6m,h=6cm,d=6mm | | | Average Value | |
| 0.247/2.58 | 0.286/2.19 | 0.319/1.83 | 0.284/2.20 | 0.036/0.375 |

| r=5cm,H=0.4m,h=4cm,d=14mm | | | Average Value | standard deviation |
|---|---|---|---|---|
| 0.283/0.62 | 0.300/0.68 | 0.344/0.53 | 0.309/0.61 | 0.031/0.076 |
| r=5cm,H=0.8m,h=2cm,d=10mm | | | Average Value | |
| 0.288/0.60 | 0.336/0.59 | 0.360/0.55 | 0.328/0.58 | 0.037/0.026 |
| r=5cm,H=1.2m,h=8cm,d=6mm | | | Average Value | |
| 0.251/2.37 | 0.291/2.01 | 0.298/1.98 | 0.280/2.12 | 0.025/0.217 |
| r=5cm,H=1.6m,h=6cm,d=2mm | | | Average Value | |
| 0.249/3.05 | 0.277/2.48 | 0.293/3.02 | 0.273/2.85 | 0.022/0.321 |

| r=2cm,H=0.4m,h=8cm,d=2mm | | | Average Value | standard deviation |
|---|---|---|---|---|
| 0.199/1.33 | 0.218/1.38 | 0.231/1.37 | 0.216/1.36 | 0.016/0.026 |
| r=2cm,H=0.8m,h=6cm,d=6mm | | | Average Value | |
| 0.239/1.28 | 0.267/1.32 | 0.289/1.12 | 0.265/1.24 | 0.025/0.106 |
| r=2cm,H=1.2m,h=4cm,d=10mm | | | Average Value | |
| 0.273/1.24 | 0.298/1.19 | 0.335/0.96 | 0.302/1.13 | 0.031/0.149 |
| r=2cm,H=1.6m,h=2cm,d=14mm | | | Average Value | |
| 0.319/0.75 | 0.361/0.59 | 0.394/0.70 | 0.358/0.68 | 0.038/0.082 |

| r=3cm,H=0.4m,h=8cm,d=6mm | | | Average Value | standard deviation |
|---|---|---|---|---|
| 0.211/0. 94 | 0.239/1.03 | 0.243/0.79 | 0.231/0.92 | 0.017/0.121 |
| r=3cm,H=0.8m,h=6cm,d=2mm | | | Average Value | |
| 0.243/1.64 | 0.258/1.55 | 0.267/1.28 | 0.256/1.49 | 0.012/0.187 |
| r=3cm,H=1.2m,h=4cm,d=14mm | | | Average Value | |
| 0.291/1.12 | 0.344/1.03 | 0.346/1.09 | 0.327/1.08 | 0.031/0.046 |
| r=3cm,H=1.6m,h=2cm,d=10mm | | | Average Value | |
| 0.324/0.77 | 0.347/0.83 | 0.382/0.92 | 0.351/0.84 | 0.029/0.076 |

| r=4cm,H=0.4m,h=6cm,d=10mm | | | Average Value | standard deviation |
|---|---|---|---|---|
| 0.254/0.88 | 0.284/0.81 | 0.323/0.62 | 0.287/0.77 | 0.035/0.135 |
| r=4cm,H=0.8m,h=8cm,d=14mm | | | Average Value | |
| 0.259/0.93 | 0.273/0.66 | 0.311/0. 84 | 0.281/0.81 | 0.027/0.137 |
| r=4cm,H=1.2m,h=2cm,d=2mm | | | Average Value | |
| 0.315/1.09 | 0.337/1.15 | 0.356/0.85 | 0.336/1.03 | 0.021/0.159 |
| r=4cm,H=1.6m,h=4cm,d=6mm | | | Average Value | |
| 0.291/1.83 | 0.312/2.05 | 0.351/2.00 | 0.318/1.96 | 0.030/0.115 |

| r=5cm,H=0.4m,h=6cm,d=14mm | | | Average Value | standard deviation |
|---|---|---|---|---|
| 0.272/0.72 | 0.295/0.64 | 0.309/0.65 | 0.292/0.67 | 0.019/0.044 |
| r=5cm,H=0.8m,h=8cm,d=10mm | | | Average Value | |
| 0.193/1.03 | 0.284/1.15 | 0.348/0.97 | 0.275/1.05 | 0.078/0.092 |
| r=5cm,H=1.2m,h=2cm,d=6mm | | | Average Value | |
| 0.323/1.03 | 0.346/1.22 | 0.372/1.17 | 0.347/1.14 | 0.025/0.098 |
| r=5cm,H=1.6m,h=4cm,d=2mm | | | Average Value | |
| 0.270/2.75 | 0.289/2.46 | 0.323/2.41 | 0.294/2.54 | 0.027/0.184 |

---

## Author Comment (AC4) · 29 Mar 2018

RC: Although the effort of attaining the comprising set of 628 tests is valued, the specified test procedure and its error handling is not evaluated profoundly enough to strengthen the authors claims. The authors themselves state that "if an obviously outlying result was obtained, the test was repeated to reduce the error." In experimental series, a unified test procedure yields the given results and subsequent data analysis then shows the range of extremal values. The judgment of an "obviously outlying result" does not correspond to a scientifically detached experimental setup and mind setting. Outliers – or at least unexpected measurement results - might hint to unexpected ef-

fects and processes, to experimental limitations, to faulty experimental procedures, etc. and are not to be discarded a priori. The authors are urged to rethink their approach to experimental findings and focus on a nonbiased data collection.

The full table of measurement results presented in the appendix do not show the uncertainties, but has to be evaluated by the reader itself. The requirement on a scientifically valuable publication and its figures is to concisely transport the gathered knowledge to the readership. It should serve to showcase the received results together with their experimental limitations – in case they are of significant size. A summary of barely treated raw results is favorable with respect to the data origin point of view, but does not serve the purpose of transporting knowledge to the reader. As an example for a minimal data treatment, a revised version of Figure 8 is attached with the uncertainties drawn from the presented measurement data. If taken the standard deviation from the attached measurements in Table 1 of the Appendix, then the graph looks the following. It is clear to the reader that the change in COR from 4 cm to 5 cm block size is not apparent within the error bars. Only after such considerations, any conclusion on data quality and/or experimental sufficiency in terms of number of drops per series are possible. As example for a thorough and concise presentation of similar results, consult for example the article "Geotechnical and kinematic parameters affecting the coefficients of restitution for rock fall analysis" P.Asteriou et. al. (https://doi.org/10.1016/j.ijrmms.2012.05.029). Such data presentation is expected and needed for the COR data to merit possible publication.

AC: Thanks for the reviewer's instructive suggestion. I had stated that "if an obviously outlying result was obtained, the test was repeated to reduce the error." The obviously outlying results were the two rare conditions that VCOR=0 or VCOR > 1. When the blocks of a small radius collided with the cushion of large particle size, such as blocks of 2 cm radius collided with the cushion of 24 mm particle size, blocks can be stuck in the seam between the particles due to the occasionality. When the blocks with relative high kinetic energy collided with the cushion of large particle size, many particles had

collided out from the platform, which may be captured by the cameras, posing some error, and these conditions are rare. Therefore if an obviously outlying result (a clear error) was obtained, the test was repeated to reduce the error. I am sorry to make you misunderstand due to my expression problem.

I have calculated the standard deviation of test data in the Tables according to the reference you recommended "Geotechnical and kinematic parameters affecting the coefficients of restitution for rock fall analysis", and I have redrawn the Figure 8 with the error bar (Mean $\pm$ SD) as an example (See attachment). However, the Figure 9 and Figure 10 include too many curves, if I redraw each curve with the error bar, the Figure 9 and Figure 10 will be confusing and Intricate, thus I have added the standard deviation for three test results of the same experiment as the supplemental material for the paper (See attachment). Thank you for your understanding and suggestion.

RC: Optimization analysis and discussion of test result: Due to the lack of data quality verification in the first step, the optimization analysis is based on bad ground. However, if amendments are done and the presented search for the leading parameter for COR and damage depth should remain unchanged, the following improvements need to be done: The presented formula (6) lacks Ry, so the derivation of Ry is not clear. Additionally, it is suggested that instead of 4 it is suggested to state "Number of levels" such that a reader who jumps to the equations is not baffled by this specific number. As the formula is not complete and in Table 3 levels are labelled k1 to k4 but in the upper discussion it is only a kxy, it is not clear how to obtain the factors presented in Table 3 for the individual levels. A clarification in notation and procedure is needed.

AC: Thanks for the reviewer's suggestion. I have revised the Eqn. (6) and the related text contents. The location of Ry is at Figure 11, so the representation of Ry was moved to the proper position. I have replaced the '4' with 'Number of levels', and revised the Table 3, adopting the expression of 'kx1 to kx4' to substitute for 'k1 to k4 to facilitate the readers' understanding (See attachment).

(In the manuscript) The analysis method used to optimize the calculation results and the optimization process is shown in Figure 11, and Ry is the range of factory.

Fig.11 Flow chart for the optimization analysis of the test (See attachment)

The four parameters, rockfall block radius, r, movement height, H, cushion thickness, h, and particle size, d, belong to the factor set $x \in (A, B, C, D)$, and the number of levels for all factors is four. The statistical test parameter under level y of factor set x can be calculated by determining Kxy (x=A, B, C, D; y=1, 2, 3, 4), i.e., the sum of all the test result indices Pxy containing level y of factor x, and dividing it by the total number of levels to obtain the average value kxy in which Pxy is the random variable of the normal distribution:

Formula (6) (See attachment)

where Kxy is the statistical parameter of factor x at level y, kxy is the average value of Kxy, and Ny is the number of levels.

RC: Furthermore, Figures 12 and 13 show many trend lines. Again, no uncertainties are given with respect to the trend lines. This is mandatory for the reader to judge the significance of the trend. An uncertainty boundary for the trend lines, estimating the error propagation from experiment to statistical evaluation should be included.

AC: Thanks for the reviewer's suggestion. According to the method of Influencing factor range analysis of all evaluation indices, the trend lines in Figures 12 is the line for average value of the COR statistical value of factor x at level y (y=1, 2, 3. 4). Due to the definition of error bar, it is meaningful to obtain the error bar of the test data if all the tests are conducted with the same conditions, and the range analysis method don't require all the conditions are same, it just needs to calculate the average value of the COR statistical value of a factor xA (either of the four factors) at level y (y=1, 2, 3. 4) in all the orthogonal test results, no matter what other factors xB, xC, xD (the other three of the four factors) are different, so it is inappropriate to calculated the error

bar of every data in the Table 3, because every data are calculated based on different test parameters, thus I think the Figures 12 and 13 with no uncertainties is reasonable. However, I fully agree with the review's point that it should give uncertainties for readers, so I supplemented the standard deviation of COR and damage depth L for three tests results in same test as the supplemental material (See attachment). According to the tables of standard deviation of COR and damage depth L, it can be seen the standard deviation are relative small, and the average vale of three test results of each test can be used for the subsequent range analysis.

RC: A few comments to further authors responses: Orthogonal test theory: The procedure is introduced, but merely as a disclaimer. Orthogonality is a basic concept in Linear Algebra. The labelling, though, has been misused in software testing and in test procedures as describe in this work, where it only should label the treated input factors as "independent". Although an "orthogonal test design" sounds elaborate, it is not to be advertised as "uniformly dispersed, neat and comparable, making it highly representative". A deletion of this text section is requested. Only data quality can judge whether a test procedure lives up to those high expectations. It is strongly suggested not to oversell used techniques.

Although a deletion of "orthogonal" is favored, a clear statement of "orthogonal test design meaning changing four input parameters independently" or similar is favored. Although some improvements in language and readability have been carried out, the text still is full of typos, especially tangled words are ubiquitous. A more careful proof reading of any final submission is mandatory. Special attention should be given to consistent variable labelling, figure layouts, figure captions, page breaks, typos. Overall the presented work still needs major refurbishments in order to be eligible for publication.

AC: Thanks for the reviewer's suggestion. I agree with your point, and I have revised the introduction of orthogonal test theory to avoid overselling used techniques, the style of the variable names are revised to keep the consistency, the typos in my manuscript are rectified carefully to facilitate readers to understand, and the style of the references

are also revised fully according to the requirements of Journal. I am really sorry about this mistake due to the version difference. Thank you sincerely for your careful and patient revision.

(In the manuscript) To explore the degree of influence of cushion particle size and thickness on COR when a rockfall moves through the cushion, orthogonal test theory was adopted to design a test program (Tao et al., 2017). Orthogonal testing is a design method that allows testing of multiple factors and multiple levels. It is based on orthogonality and selects representative points from a comprehensive experiment for testing, so that fewer trials can fully reflect the impact of the variation of each factor on the index. When these factors cannot be considered in full, the leading factor is considered to achieve the expected effects to a great extent. Four independent parameters, the rockfall block radius, r, movement height, H, cushion thickness, h, and particle size, d, were selected as the basic factors of orthogonal design to test. The purpose of doing an orthogonal test is to explore the degree of influence of the four different factors on the COR and damage depth, L, and find the best combination to reach the optimal protective effect when a rockfall collides with a cushion. The damage depth (L) is the depth to which the cushion is influenced after a rockfall has collided with it and can be used to represent the degree of damage to the cushion.

Please also note the supplement to this comment:
https://www.nat-hazards-earth-syst-sci-discuss.net/nhess-2018-16/nhess-2018-16-AC4-supplement.pdf

**Supplement:**

$$k_{xy} = \frac{K_{xy}}{N_y} = \sum P_{xy} \Big/ N_y \ , \ (6)$$

where $K_{xy}$ is the statistical parameter of factor *x* at level *y*, $k_{xy}$ is the average value of $K_{xy}$, and $N_y$ is the number of levels.

[Figure]

Fig. 8 The *COR* (Mean ± SD) of block collisions with the plate. (Error bars: one standard deviation)

[Figure]

Fig.11 Flow chart for the optimization analysis of the test

Table 1 Orthogonal test results (In the manuscript)

| Test number | Rockfall radius r/cm | Movement height H/m | Cushion thickness h/cm | Particle size d/cm | Damage depth of cushion L/cm (Mean/Std dev) | COR of collision between rockfall and cushion (Mean/Std dev) |
|---|---|---|---|---|---|---|
| 1 | 2 | 0.4 | 2 | 0.2 | 0.65/0.082 | 0.278/0.012 |
| 2 | 2 | 0.8 | 4 | 0.6 | 0.74/0.056 | 0.273/0.023 |
| 3 | 2 | 1.2 | 6 | 1.0 | 0.93/0.082 | 0.282/0.029 |
| 4 | 2 | 1.6 | 8 | 1.4 | 1.05/0.046 | 0.295/0.028 |
| 5 | 3 | 0.4 | 2 | 0.6 | 0.58/0.053 | 0.294/0.012 |
| 6 | 3 | 0.8 | 4 | 0.2 | 1.45/0.165 | 0.265/0.015 |
| 7 | 3 | 1.2 | 6 | 1.4 | 1.03/0.171 | 0.317/0.041 |
| 8 | 3 | 1.6 | 8 | 1.0 | 1.60/0.193 | 0.280/0.020 |
| 9 | 4 | 0.4 | 4 | 1.0 | 0.62/0.036 | 0.296/0.028 |
| 10 | 4 | 0.8 | 2 | 1.4 | 0.56/0.104 | 0.338/0.029 |
| 11 | 4 | 1.2 | 8 | 0.2 | 2.60/0.303 | 0.256/0.022 |
| 12 | 4 | 1.6 | 6 | 0.6 | 2.20/0.375 | 0.284/0.036 |
| 13 | 5 | 0.4 | 4 | 1.4 | 0.61/0.076 | 0.309/0.031 |
| 14 | 5 | 0.8 | 2 | 1.0 | 0.58/0.026 | 0.328/0.037 |
| 15 | 5 | 1.2 | 8 | 0.6 | 2.12/0.217 | 0.280/0.025 |

| 16 | 5 | 1.6 | 6 | 0.2 | 2.85/0.321 | 0.273/0.022 |
|----|---|-----|---|-----|------------|-------------|
| 17 | 2 | 0.4 | 8 | 0.2 | 1.36/0.026 | 0.216/0.016 |
| 18 | 2 | 0.8 | 6 | 0.6 | 1.24/0.106 | 0.265/0.025 |
| 19 | 2 | 1.2 | 4 | 1.0 | 1.13/0.149 | 0.302/0.031 |
| 20 | 2 | 1.6 | 2 | 1.4 | 0.68/0.082 | 0.358/0.038 |
| 21 | 3 | 0.4 | 8 | 0.6 | 0.92/0.121 | 0.231/0.017 |
| 22 | 3 | 0.8 | 6 | 0.2 | 1.49/0.187 | 0.256/0.012 |
| 23 | 3 | 1.2 | 4 | 1.4 | 1.08/0.046 | 0.327/0.031 |
| 24 | 3 | 1.6 | 2 | 1.0 | 0.84/0.076 | 0.351/0.029 |
| 25 | 4 | 0.4 | 6 | 1.0 | 0.77/0.135 | 0.287/0.035 |
| 26 | 4 | 0.8 | 8 | 1.4 | 0.81/0.137 | 0.281/0.027 |
| 27 | 4 | 1.2 | 2 | 0.2 | 1.03/0.159 | 0.336/0.021 |
| 28 | 4 | 1.6 | 4 | 0.6 | 1.96/0.115 | 0.318/0.030 |
| 29 | 5 | 0.4 | 6 | 1.4 | 0.67/0.044 | 0.292/0.019 |
| 30 | 5 | 0.8 | 8 | 1.0 | 1.05/0.092 | 0.275/0.078 |
| 31 | 5 | 1.2 | 2 | 0.6 | 1.14/0.098 | 0.347/0.025 |
| 32 | 5 | 1.6 | 4 | 0.2 | 2.54/0.184 | 0.294/0.027 |

Table 2 Influencing factor range analysis of all evaluation indices (In the manuscript)

| Evaluation index | Levels | Rockfall radius r/cm | Movement height H/m | Cushions thickness h/cm | Particle size d/cm |
|------------------|--------|----------------------|---------------------|-------------------------|--------------------|
| COR of collision between rockfall and cushion | $k_{x1}$ | 0.285 | 0.271 | 0.325 | 0.270 |
| | $k_{x2}$ | 0.288 | 0.287 | 0.296 | 0.285 |
| | $k_{x3}$ | 0.298 | 0.305 | 0.281 | 0.301 |
| | $k_{x4}$ | 0.299 | 0.306 | 0.267 | 0.313 |
| | $R_y$ | 0.014 | 0.035 | 0.058 | 0.043 |
| Damage depth of cushion L | $k_{x1}$ | 0.97 | 0.78 | 0.76 | 1.75 |
| | $k_{x2}$ | 1.12 | 0.99 | 1.26 | 1.35 |
| | $k_{x3}$ | 1.32 | 1.38 | 1.40 | 0.94 |
| | $k_{x4}$ | 1.45 | 1.72 | 1.44 | 0.81 |
| | $R_y$ | 0.48 | 0.94 | 0.68 | 0.94 |

Table 3 The COR of block collisions with the plate (supplemental material)

| H=1.2m ,h=0cm,d=0mm | | | | | Average Value | standard deviation |
|---------------------|---|-------|-------|-------|---------------|--------------------|
| r=2cm | | 0.363 | 0.368 | 0.421 | | 0.384 | 0.032 |
| r=3cm | | 0.398 | 0.428 | 0.437 | | 0.421 | 0.020 |
| r=4cm | | 0.386 | 0.482 | 0.443 | | 0.437 | 0.048 |
| r=5cm | | 0.403 | 0.458 | 0.471 | | 0.444 | 0.036 |

Table 4 Comparison of the COR of different blocks released from a height of 1.2m

(Standard deviation and average value of COR is calculated by the middle three values for cushion particle sizes of 1.8 cm and 2.4 cm)
(supplemental material)

| H=1.2m,r=2cm,h=2cm | | | | | Average Value | standard deviation |
|---|---|---|---|---|---|---|
| d=2mm | | 0.310 | 0.329 | 0.339 | | 0.326 | 0.326 |
| d=6mm | | 0.298 | 0.348 | 0.35 | | 0.332 | 0.332 |
| d=10mm | | 0.319 | 0.343 | 0.376 | | 0.346 | 0.346 |
| d=14mm | | 0.314 | 0.344 | 0.371 | | 0.343 | 0.343 |
| d=18mm | 0.262 | 0.291 | 0.337 | 0.416 | 0.443 | 0.348 | 0.348 |
| d=24mm | 0.234 | 0.288 | 0.371 | 0.403 | 0.421 | 0.354 | 0.354 |

| H=1.2m,r=2cm,h=4cm | | | | | Average Value | standard deviation |
|---|---|---|---|---|---|---|
| d=2mm | | 0.281 | 0.285 | 0.316 | | 0.294 | 0.019 |
| d=6mm | | 0.292 | 0.338 | 0.345 | | 0.325 | 0.029 |
| d=10mm | | 0.261 | 0.314 | 0.331 | | 0.302 | 0.037 |
| d=14mm | | 0.285 | 0.324 | 0.360 | | 0.323 | 0.038 |
| d=18mm | 0.215 | 0.252 | 0.323 | 0.376 | 0.386 | 0.317 | 0.062 |
| d=24mm | 0.267 | 0.270 | 0.304 | 0.362 | 0.403 | 0.312 | 0.047 |

| H=1.2m,r=2cm,h=6cm | | | | | Average Value | standard deviation |
|---|---|---|---|---|---|---|
| d=2mm | | 0.240 | 0.267 | 0.270 | | 0.259 | 0.017 |
| d=6mm | | 0.239 | 0.277 | 0.306 | | 0.274 | 0.034 |
| d=10mm | | 0.246 | 0.283 | 0.317 | | 0.282 | 0.036 |
| d=14mm | | 0.251 | 0.267 | 0.331 | | 0.283 | 0.042 |
| d=18mm | 0.231 | 0.259 | 0.299 | 0.345 | 0.382 | 0.301 | 0.043 |
| d=24mm | 0.226 | 0.261 | 0.291 | 0.336 | 0.370 | 0.296 | 0.038 |

| H=1.2m,r=2cm,h=8cm | | | | | Average Value | standard deviation |
|---|---|---|---|---|---|---|
| d=2mm | | 0.215 | 0.243 | 0.271 | | 0.243 | 0.028 |
| d=6mm | | 0.211 | 0.261 | 0.290 | | 0.254 | 0.040 |
| d=10mm | | 0.216 | 0.261 | 0.312 | | 0.263 | 0.048 |
| d=14mm | | 0.223 | 0.286 | 0.304 | | 0.271 | 0.043 |
| d=18mm | 0.205 | 0.225 | 0.285 | 0.321 | 0.356 | 0.277 | 0.048 |
| d=24mm | 0.193 | 0.206 | 0.292 | 0.354 | 0.371 | 0.284 | 0.074 |

| H=1.2m,r=2cm,h=10cm | | | | | Average Value | standard deviation |
|---|---|---|---|---|---|---|
| d=2mm | | 0.201 | 0.245 | 0.277 | | 0.241 | 0.038 |
| d=6mm | | 0.198 | 0.250 | 0.293 | | 0.247 | 0.048 |

| | | | | | | Average Value | standard deviation |
|---|---|---|---|---|---|---|---|
| d=10mm | | 0.226 | 0.251 | 0.288 | | 0.255 | 0.031 |
| d=14mm | | 0.210 | 0.253 | 0.311 | | 0.258 | 0.051 |
| d=18mm | 0.190 | 0.192 | 0.273 | 0.327 | 0.363 | 0.264 | 0.068 |
| d=24mm | 0.204 | 0.214 | 0.292 | 0.325 | 0.350 | 0.277 | 0.057 |

| H=1.2m,r=2cm,h=12cm | | | | | | Average Value | standard deviation |
|---|---|---|---|---|---|---|---|
| d=2mm | | 0.207 | 0.218 | 0.259 | | 0.228 | 0.027 |
| d=6mm | | 0.190 | 0.236 | 0.273 | | 0.233 | 0.042 |
| d=10mm | | 0.214 | 0.225 | 0.302 | | 0.247 | 0.048 |
| d=14mm | | 0.200 | 0.243 | 0.313 | | 0.252 | 0.057 |
| d=18mm | 0.189 | 0.198 | 0.236 | 0.319 | 0.328 | 0.251 | 0.062 |
| d=24mm | 0.198 | 0.221 | 0.251 | 0.326 | 0.343 | 0.266 | 0.054 |

| H=1.2m,r=2cm,h=14cm | | | | | | Average Value | standard deviation |
|---|---|---|---|---|---|---|---|
| d=2mm | | 0.184 | 0.230 | 0.246 | | 0.22 | 0.032 |
| d=6mm | | 0.199 | 0.214 | 0.283 | | 0.232 | 0.045 |
| d=10mm | | 0.204 | 0.251 | 0.265 | | 0.24 | 0.032 |
| d=14mm | | 0.183 | 0.224 | 0.301 | | 0.236 | 0.060 |
| d=18mm | 0.162 | 0.196 | 0.260 | 0.291 | 0.314 | 0.249 | 0.048 |
| d=24mm | 0.194 | 0.208 | 0.250 | 0.316 | 0.353 | 0.258 | 0.054 |

| H=1.2m,r=3cm,h=2cm | | | | | | Average Value | standard deviation |
|---|---|---|---|---|---|---|---|
| d=2mm | | 0.313 | 0.338 | 0.351 | | 0.334 | 0.019 |
| d=6mm | | 0.326 | 0.348 | 0.349 | | 0.341 | 0.013 |
| d=10mm | | 0.308 | 0.354 | 0.379 | | 0.347 | 0.036 |
| d=14mm | | 0.311 | 0.342 | 0.409 | | 0.354 | 0.050 |
| d=18mm | 0.261 | 0.333 | 0.336 | 0.387 | 0.396 | 0.352 | 0.030 |
| d=24mm | 0.256 | 0.322 | 0.369 | 0.413 | 0.420 | 0.368 | 0.046 |

| H=1.2m,r=3cm,h=4cm | | | | | | Average Value | standard deviation |
|---|---|---|---|---|---|---|---|
| d=2mm | | 0.261 | 0.316 | 0.329 | | 0.302 | 0.036 |
| d=6mm | | 0.276 | 0.309 | 0.360 | | 0.315 | 0.042 |
| d=10mm | | 0.267 | 0.329 | 0.352 | | 0.316 | 0.044 |
| d=14mm | | 0.282 | 0.320 | 0.379 | | 0.327 | 0.049 |
| d=18mm | 0.245 | 0.298 | 0.314 | 0.366 | 0.381 | 0.326 | 0.036 |
| d=24mm | 0.253 | 0.276 | 0.321 | 0.405 | 0.415 | 0.334 | 0.065 |

| H=1.2m,r=3cm,h=6cm | | | | | | Average Value | standard deviation |
|---|---|---|---|---|---|---|---|

| | | | | | | Average Value | standard deviation |
|---|---|---|---|---|---|---|---|
| d=2mm | | 0.254 | 0.273 | 0.304 | | 0.277 | 0.025 |
| d=6mm | | 0.262 | 0.281 | 0.309 | | 0.284 | 0.024 |
| d=10mm | | 0.255 | 0.288 | 0.321 | | 0.288 | 0.033 |
| d=14mm | | 0.283 | 0.311 | 0.360 | | 0.318 | 0.039 |
| d=18mm | 0.185 | 0.261 | 0.300 | 0.366 | 0.384 | 0.309 | 0.053 |
| d=24mm | 0.248 | 0.248 | 0.337 | 0.390 | 0.393 | 0.325 | 0.072 |

| H=1.2m,r=3cm,h=8cm | | | | | | Average Value | standard deviation |
|---|---|---|---|---|---|---|---|
| d=2mm | | 0.229 | 0.235 | 0.277 | | 0.247 | 0.026 |
| d=6mm | | 0.213 | 0.270 | 0.303 | | 0.262 | 0.046 |
| d=10mm | | 0.241 | 0.247 | 0.313 | | 0.267 | 0.040 |
| d=14mm | | 0.223 | 0.264 | 0.332 | | 0.273 | 0.055 |
| d=18mm | 0.214 | 0.220 | 0.303 | 0.320 | 0.341 | 0.281 | 0.054 |
| d=24mm | 0.235 | 0.261 | 0.292 | 0.323 | 0.354 | 0.292 | 0.031 |

| H=1.2m,r=3cm,h=10cm | | | | | | Average Value | standard deviation |
|---|---|---|---|---|---|---|---|
| d=2mm | | 0.212 | 0.233 | 0.266 | | 0.237 | 0.027 |
| d=6mm | | 0.223 | 0.240 | 0.275 | | 0.246 | 0.027 |
| d=10mm | | 0.228 | 0.246 | 0.288 | | 0.254 | 0.031 |
| d=14mm | | 0.213 | 0.271 | 0.302 | | 0.262 | 0.045 |
| d=18mm | 0.192 | 0.211 | 0.251 | 0.309 | 0.324 | 0.257 | 0.049 |
| d=24mm | 0.211 | 0.232 | 0.246 | 0.326 | 0.341 | 0.268 | 0.051 |

| H=1.2m,r=3cm,h=12cm | | | | | | Average Value | standard deviation |
|---|---|---|---|---|---|---|---|
| d=2mm | | 0.194 | 0.220 | 0.264 | | 0.226 | 0.035 |
| d=6mm | | 0.199 | 0.231 | 0.287 | | 0.239 | 0.045 |
| d=10mm | | 0.228 | 0.234 | 0.264 | | 0.242 | 0.019 |
| d=14mm | | 0.204 | 0.256 | 0.284 | | 0.248 | 0.041 |
| d=18mm | 0.213 | 0.228 | 0.242 | 0.295 | 0.320 | 0.255 | 0.035 |
| d=24mm | 0.186 | 0.217 | 0.259 | 0.301 | 0.336 | 0.259 | 0.042 |

| H=1.2m,r=3cm,h=14cm | | | | | | Average Value | standard deviation |
|---|---|---|---|---|---|---|---|
| d=2mm | | 0.172 | 0.206 | 0.276 | | 0.218 | 0.053 |
| d=6mm | | 0.203 | 0.215 | 0.254 | | 0.224 | 0.027 |
| d=10mm | | 0.188 | 0.224 | 0.275 | | 0.229 | 0.044 |
| d=14mm | | 0.178 | 0.229 | 0.286 | | 0.231 | 0.054 |
| d=18mm | 0.186 | 0.192 | 0.244 | 0.302 | 0.335 | 0.246 | 0.055 |
| d=24mm | 0.191 | 0.228 | 0.246 | 0.312 | 0.343 | 0.262 | 0.044 |

| H=1.2m,r=4cm,h=2cm | | | | | Average Value | standard deviation |
|---|---|---|---|---|---|---|
| d=2mm | | 0.316 | 0.338 | 0.354 | | 0.336 | 0.019 |
| d=6mm | | 0.329 | 0.343 | 0.372 | | 0.348 | 0.022 |
| d=10mm | | 0.333 | 0.351 | 0.384 | | 0.356 | 0.026 |
| d=14mm | | 0.321 | 0.358 | 0.416 | | 0.365 | 0.048 |
| d=18mm | 0.285 | 0.329 | 0.371 | 0.401 | 0.411 | 0.367 | 0.036 |
| d=24mm | 0.313 | 0.334 | 0.369 | 0.413 | 0.432 | 0.372 | 0.040 |

| H=1.2m,r=4cm,h=4cm | | | | | Average Value | standard deviation |
|---|---|---|---|---|---|---|
| d=2mm | | 0.284 | 0.307 | 0.336 | | 0.309 | 0.026 |
| d=6mm | | 0.298 | 0.319 | 0.346 | | 0.321 | 0.024 |
| d=10mm | | 0.288 | 0.310 | 0.347 | | 0.315 | 0.030 |
| d=14mm | | 0.308 | 0.316 | 0.351 | | 0.325 | 0.023 |
| d=18mm | 0.263 | 0.295 | 0.339 | 0.368 | 0.374 | 0.334 | 0.037 |
| d=24mm | 0.275 | 0.309 | 0.326 | 0.394 | 0.410 | 0.343 | 0.045 |

| H=1.2m,r=4cm,h=6cm | | | | | Average Value | standard deviation |
|---|---|---|---|---|---|---|
| d=2mm | | 0.267 | 0.279 | 0.294 | | 0.28 | 0.014 |
| d=6mm | | 0.294 | 0.304 | 0.329 | | 0.309 | 0.018 |
| d=10mm | | 0.273 | 0.286 | 0.317 | | 0.292 | 0.023 |
| d=14mm | | 0.280 | 0.292 | 0.304 | | 0.292 | 0.012 |
| d=18mm | 0.257 | 0.282 | 0.303 | 0.351 | 0.384 | 0.312 | 0.035 |
| d=24mm | 0.240 | 0.294 | 0.322 | 0.359 | 0.367 | 0.325 | 0.033 |

| H=1.2m,r=4cm,h=8cm | | | | | Average Value | standard deviation |
|---|---|---|---|---|---|---|
| d=2mm | | 0.249 | 0.250 | 0.269 | | 0.256 | 0.011 |
| d=6mm | | 0.250 | 0.267 | 0.296 | | 0.271 | 0.023 |
| d=10mm | | 0.248 | 0.274 | 0.306 | | 0.276 | 0.029 |
| d=14mm | | 0.252 | 0.271 | 0.299 | | 0.274 | 0.024 |
| d=18mm | 0.235 | 0.263 | 0.291 | 0.325 | 0.344 | 0.293 | 0.031 |
| d=24mm | 0.259 | 0.267 | 0.298 | 0.341 | 0.367 | 0.302 | 0.037 |

| H=1.2m,r=4cm,h=10cm | | | | | Average Value | standard deviation |
|---|---|---|---|---|---|---|
| d=2mm | | 0.239 | 0.249 | 0.268 | | 0.252 | 0.015 |
| d=6mm | | 0.235 | 0.261 | 0.278 | | 0.258 | 0.022 |
| d=10mm | | 0.247 | 0.264 | 0.296 | | 0.269 | 0.025 |
| d=14mm | | 0.254 | 0.249 | 0.292 | | 0.265 | 0.024 |
| d=18mm | 0.226 | 0.243 | 0.276 | 0.324 | 0.331 | 0.281 | 0.041 |

| | | | | | Average Value | standard deviation |
|---|---|---|---|---|---|---|
| d=24mm | 0.231 | 0.236 | 0.277 | 0.321 | 0.354 | 0.278 | 0.043 |

| H=1.2m,r=4cm,h=12cm | | | | | Average Value | standard deviation |
|---|---|---|---|---|---|---|
| d=2mm | | 0.229 | 0.231 | 0.248 | | 0.236 | 0.010 |
| d=6mm | | 0.221 | 0.244 | 0.270 | | 0.245 | 0.025 |
| d=10mm | | 0.216 | 0.227 | 0.268 | | 0.237 | 0.027 |
| d=14mm | | 0.207 | 0.239 | 0.283 | | 0.243 | 0.038 |
| d=18mm | 0.186 | 0.208 | 0.251 | 0.297 | 0.319 | 0.252 | 0.045 |
| d=24mm | 0.217 | 0.218 | 0.272 | 0.284 | 0.326 | 0.258 | 0.035 |

| H=1.2m,r=4cm,h=14cm | | | | | Average Value | standard deviation |
|---|---|---|---|---|---|---|
| d=2mm | | 0.217 | 0.218 | 0.237 | | 0.224 | 0.011 |
| d=6mm | | 0.215 | 0.231 | 0.259 | | 0.235 | 0.022 |
| d=10mm | | 0.196 | 0.229 | 0.271 | | 0.232 | 0.038 |
| d=14mm | | 0.219 | 0.224 | 0.268 | | 0.237 | 0.027 |
| d=18mm | 0.181 | 0.207 | 0.256 | 0.281 | 0.314 | 0.248 | 0.038 |
| d=24mm | 0.196 | 0.219 | 0.247 | 0.293 | 0.335 | 0.253 | 0.037 |

| H=1.2m,r=5cm,h=2cm | | | | | Average Value | standard deviation |
|---|---|---|---|---|---|---|
| d=2mm | | 0.328 | 0.336 | 0.356 | | 0.34 | 0.014 |
| d=6mm | | 0.320 | 0.342 | 0.364 | | 0.342 | 0.022 |
| d=10mm | | 0.324 | 0.351 | 0.393 | | 0.356 | 0.035 |
| d=14mm | | 0.342 | 0.364 | 0.398 | | 0.368 | 0.028 |
| d=18mm | 0.317 | 0.338 | 0.374 | 0.401 | 0.426 | 0.371 | 0.032 |
| d=24mm | 0.302 | 0.354 | 0.365 | 0.421 | 0.437 | 0.38 | 0.036 |

| H=1.2m,r=5cm,h=4cm | | | | | Average Value | standard deviation |
|---|---|---|---|---|---|---|
| d=2mm | | 0.316 | 0.317 | 0.339 | | 0.324 | 0.013 |
| d=6mm | | 0.297 | 0.306 | 0.330 | | 0.311 | 0.017 |
| d=10mm | | 0.294 | 0.321 | 0.354 | | 0.323 | 0.030 |
| d=14mm | | 0.318 | 0.340 | 0.374 | | 0.344 | 0.028 |
| d=18mm | 0.318 | 0.321 | 0.322 | 0.386 | 0.397 | 0.343 | 0.037 |
| d=24mm | 0.284 | 0.328 | 0.355 | 0.373 | 0.403 | 0.352 | 0.023 |

| H=1.2m,r=5cm,h=6cm | | | | | Average Value | standard deviation |
|---|---|---|---|---|---|---|
| d=2mm | | 0.286 | 0.286 | 0.301 | | 0.291 | 0.009 |
| d=6mm | | 0.274 | 0.287 | 0.315 | | 0.292 | 0.021 |
| d=10mm | | 0.301 | 0.324 | 0.329 | | 0.318 | 0.015 |

| d=14mm | | 0.291 | 0.298 | 0.338 | | 0.309 | 0.025 |
| d=18mm | 0.244 | 0.286 | 0.314 | 0.378 | 0.398 | 0.326 | 0.047 |
| d=24mm | 0.270 | 0.293 | 0.316 | 0.381 | 0.401 | 0.33 | 0.046 |

| H=1.2m,r=5cm,h=8cm | | | | | | Average Value | standard deviation |
|---|---|---|---|---|---|---|---|
| d=2mm | | 0.254 | 0.262 | 0.279 | | 0.265 | 0.013 |
| d=6mm | | 0.270 | 0.276 | 0.294 | | 0.28 | 0.012 |
| d=10mm | | 0.261 | 0.294 | 0.309 | | 0.288 | 0.025 |
| d=14mm | | 0.272 | 0.283 | 0.324 | | 0.293 | 0.027 |
| d=18mm | 0.226 | 0.256 | 0.294 | 0.356 | 0.362 | 0.302 | 0.050 |
| d=24mm | 0.234 | 0.286 | 0.291 | 0.362 | 0.374 | 0.313 | 0.043 |

| H=1.2m,r=5cm,h=10cm | | | | | | Average Value | standard deviation |
|---|---|---|---|---|---|---|---|
| d=2mm | | 0.247 | 0.261 | 0.281 | | 0.263 | 0.017 |
| d=6mm | | 0.233 | 0.273 | 0.289 | | 0.265 | 0.029 |
| d=10mm | | 0.249 | 0.257 | 0.301 | | 0.269 | 0.028 |
| d=14mm | | 0.250 | 0.268 | 0.298 | | 0.272 | 0.024 |
| d=18mm | 0.208 | 0.237 | 0.261 | 0.315 | 0.320 | 0.271 | 0.040 |
| d=24mm | 0.221 | 0.255 | 0.273 | 0.336 | 0.373 | 0.288 | 0.043 |

| H=1.2m,r=5cm,h=12cm | | | | | | Average Value | standard deviation |
|---|---|---|---|---|---|---|---|
| d=2mm | | 0.230 | 0.236 | 0.254 | | 0.24 | 0.012 |
| d=6mm | | 0.213 | 0.251 | 0.265 | | 0.243 | 0.027 |
| d=10mm | | 0.220 | 0.245 | 0.291 | | 0.252 | 0.036 |
| d=14mm | | 0.242 | 0.240 | 0.289 | | 0.257 | 0.028 |
| d=18mm | 0.197 | 0.224 | 0.242 | 0.311 | 0.342 | 0.259 | 0.046 |
| d=24mm | 0.208 | 0.209 | 0.261 | 0.328 | 0.337 | 0.266 | 0.060 |

| H=1.2m,r=5cm,h=14cm | | | | | | Average Value | standard deviation |
|---|---|---|---|---|---|---|---|
| d=2mm | | 0.207 | 0.217 | 0.236 | | 0.22 | 0.015 |
| d=6mm | | 0.204 | 0.229 | 0.257 | | 0.23 | 0.027 |
| d=10mm | | 0.227 | 0.234 | 0.250 | | 0.237 | 0.012 |
| d=14mm | | 0.215 | 0.241 | 0.270 | | 0.242 | 0.028 |
| d=18mm | 0.184 | 0.189 | 0.235 | 0.278 | 0.304 | 0.234 | 0.045 |
| d=24mm | 0.201 | 0.218 | 0.258 | 0.286 | 0.316 | 0.254 | 0.034 |

Table 5 Comparison of the COR for different blocks colliding with an 8-cm thick cushion
(Standard deviation and average value of COR is calculated by the middle three
values for cushion particle sizes of 1.8 cm and 2.4 cm)

(supplemental material)

| h=8cm,r=2cm,H=0.4m | | | | | | Average Value | standard deviation |
|---|---|---|---|---|---|---|---|
| d=2mm | | 0.197 | 0.214 | 0.237 | | 0.216 | 0.020 |
| d=6mm | | 0.221 | 0.222 | 0.241 | | 0.228 | 0.011 |
| d=10mm | | 0.216 | 0.228 | 0.264 | | 0.236 | 0.025 |
| d=14mm | | 0.223 | 0.256 | 0.283 | | 0.254 | 0.030 |
| d=18mm | 0.204 | 0.201 | 0.261 | 0.306 | 0.311 | 0.256 | 0.053 |
| d=24mm | 0.203 | 0.221 | 0.264 | 0.295 | 0.316 | 0.260 | 0.037 |

| h=8cm,r=2cm,H=0.8m | | | | | | Average Value | standard deviation |
|---|---|---|---|---|---|---|---|
| d=2mm | | 0.219 | 0.231 | 0.237 | | 0.229 | 0.009 |
| d=6mm | | 0.200 | 0.248 | 0.254 | | 0.234 | 0.030 |
| d=10mm | | 0.220 | 0.242 | 0.273 | | 0.245 | 0.027 |
| d=14mm | | 0.213 | 0.246 | 0.270 | | 0.243 | 0.029 |
| d=18mm | 0.214 | 0.221 | 0.274 | 0.291 | 0.327 | 0.262 | 0.037 |
| d=24mm | 0.191 | 0.212 | 0.271 | 0.318 | 0.331 | 0.267 | 0.053 |

| h=8cm,r=2cm,H=1.2m | | | | | | Average Value | standard deviation |
|---|---|---|---|---|---|---|---|
| d=2mm | | 0.228 | 0.236 | 0.265 | | 0.243 | 0.019 |
| d=6mm | | 0.217 | 0.267 | 0.278 | | 0.254 | 0.033 |
| d=10mm | | 0.231 | 0.262 | 0.296 | | 0.263 | 0.033 |
| d=14mm | | 0.222 | 0.283 | 0.308 | | 0.271 | 0.044 |
| d=18mm | 0.207 | 0.226 | 0.287 | 0.318 | 0.336 | 0.277 | 0.047 |
| d=24mm | 0.226 | 0.247 | 0.299 | 0.306 | 0.337 | 0.284 | 0.032 |

| h=8cm,r=2cm,H=1.6m | | | | | | Average Value | standard deviation |
|---|---|---|---|---|---|---|---|
| d=2mm | | 0.232 | 0.239 | 0.258 | | 0.243 | 0.013 |
| d=6mm | | 0.240 | 0.243 | 0.273 | | 0.252 | 0.018 |
| d=10mm | | 0.223 | 0.294 | 0.296 | | 0.271 | 0.042 |
| d=14mm | | 0.245 | 0.287 | 0.338 | | 0.290 | 0.047 |
| d=18mm | 0.221 | 0.249 | 0.279 | 0.321 | 0.332 | 0.283 | 0.036 |

| d=24mm | 0.218 | 0.234 | 0.276 | 0.336 | 0.359 | 0.282 | 0.051 |
|---|---|---|---|---|---|---|---|

| h=8cm,r=3cm,H=0.4m | | | | | Average Value | standard deviation |
|---|---|---|---|---|---|---|
| d=2mm | | 0.208 | 0.227 | 0.237 | | 0.224 | 0.015 |
| d=6mm | | 0.212 | 0.226 | 0.255 | | 0.231 | 0.022 |
| d=10mm | | 0.228 | 0.231 | 0.270 | | 0.243 | 0.023 |
| d=14mm | | 0.216 | 0.251 | 0.289 | | 0.252 | 0.037 |
| d=18mm | 0.186 | 0.222 | 0.267 | 0.306 | 0.316 | 0.265 | 0.042 |
| d=24mm | 0.179 | 0.211 | 0.272 | 0.321 | 0.331 | 0.268 | 0.055 |

| h=8cm,r=3cm,H=0.8m | | | | | Average Value | standard deviation |
|---|---|---|---|---|---|---|
| d=2mm | | 0.222 | 0.235 | 0.251 | | 0.236 | 0.015 |
| d=6mm | | 0.218 | 0.247 | 0.264 | | 0.243 | 0.023 |
| d=10mm | | 0.235 | 0.251 | 0.306 | | 0.264 | 0.037 |
| d=14mm | | 0.220 | 0.274 | 0.292 | | 0.262 | 0.037 |
| d=18mm | 0.231 | 0.242 | 0.254 | 0.305 | 0.329 | 0.267 | 0.033 |
| d=24mm | 0.224 | 0.234 | 0.270 | 0.324 | 0.354 | 0.276 | 0.045 |

| h=8cm,r=3cm,H=1.2m | | | | | Average Value | standard deviation |
|---|---|---|---|---|---|---|
| d=2mm | | 0.229 | 0.243 | 0.269 | | 0.247 | 0.020 |
| d=6mm | | 0.241 | 0.265 | 0.28 | | 0.262 | 0.020 |
| d=10mm | | 0.234 | 0.27 | 0.297 | | 0.267 | 0.032 |
| d=14mm | | 0.221 | 0.288 | 0.310 | | 0.273 | 0.046 |
| d=18mm | 0.198 | 0.239 | 0.283 | 0.321 | 0.334 | 0.281 | 0.041 |
| d=24mm | 0.227 | 0.251 | 0.287 | 0.338 | 0.356 | 0.292 | 0.044 |

| h=8cm,r=3cm,H=1.6m | | | | | Average Value | standard deviation |
|---|---|---|---|---|---|---|
| d=2mm | | 0.242 | 0.251 | 0.269 | | 0.254 | 0.014 |
| d=6mm | | 0.228 | 0.279 | 0.288 | | 0.265 | 0.032 |
| d=10mm | | 0.261 | 0.284 | 0.313 | | 0.286 | 0.026 |
| d=14mm | | 0.256 | 0.286 | 0.325 | | 0.289 | 0.035 |
| d=18mm | 0.262 | 0.276 | 0.291 | 0.312 | 0.323 | 0.293 | 0.018 |
| d=24mm | 0.214 | 0.272 | 0.295 | 0.336 | 0.351 | 0.301 | 0.032 |

| h=8cm,r=4cm,H=0.4m | | | | | Average Value | standard deviation |
|---|---|---|---|---|---|---|
| d=2mm | | 0.220 | 0.227 | 0.246 | | 0.231 | 0.013 |
| d=6mm | | 0.233 | 0.234 | 0.259 | | 0.242 | 0.015 |
| d=10mm | | 0.212 | 0.241 | 0.264 | | 0.239 | 0.026 |

| d=14mm | | 0.241 | 0.252 | 0.299 | | 0.264 | 0.031 |
|---|---|---|---|---|---|---|---|
| d=18mm | 0.209 | 0.239 | 0.253 | 0.294 | 0.321 | 0.262 | 0.029 |
| d=24mm | 0.213 | 0.236 | 0.278 | 0.314 | 0.316 | 0.276 | 0.039 |

| h=8cm,r=4cm,H=0.8m | | | | | Average Value | standard deviation |
|---|---|---|---|---|---|---|---|
| d=2mm | | 0.223 | 0.247 | 0.265 | | 0.245 | 0.021 |
| d=6mm | | 0.248 | 0.252 | 0.271 | | 0.257 | 0.012 |
| d=10mm | | 0.236 | 0.257 | 0.293 | | 0.262 | 0.029 |
| d=14mm | | 0.260 | 0.285 | 0.316 | | 0.287 | 0.028 |
| d=18mm | 0.216 | 0.247 | 0.287 | 0.324 | 0.338 | 0.286 | 0.039 |
| d=24mm | 0.224 | 0.230 | 0.302 | 0.338 | 0.351 | 0.290 | 0.055 |

| h=8cm,r=4cm,H=1.2m | | | | | Average Value | standard deviation |
|---|---|---|---|---|---|---|---|
| d=2mm | | 0.245 | 0.254 | 0.269 | | 0.256 | 0.012 |
| d=6mm | | 0.230 | 0.287 | 0.296 | | 0.271 | 0.036 |
| d=10mm | | 0.249 | 0.281 | 0.298 | | 0.276 | 0.025 |
| d=14mm | | 0.268 | 0.278 | 0.306 | | 0.284 | 0.020 |
| d=18mm | 0.225 | 0.256 | 0.291 | 0.332 | 0.348 | 0.293 | 0.038 |
| d=24mm | 0.248 | 0.282 | 0.303 | 0.321 | 0.367 | 0.302 | 0.020 |

| h=8cm,r=4cm,H=1.6m | | | | | Average Value | standard deviation |
|---|---|---|---|---|---|---|---|
| d=2mm | | 0.243 | 0.257 | 0.283 | | 0.261 | 0.020 |
| d=6mm | | 0.269 | 0.282 | 0.304 | | 0.285 | 0.018 |
| d=10mm | | 0.254 | 0.283 | 0.321 | | 0.286 | 0.034 |
| d=14mm | | 0.237 | 0.327 | 0.333 | | 0.299 | 0.054 |
| d=18mm | 0.234 | 0.273 | 0.306 | 0.354 | 0.375 | 0.311 | 0.041 |
| d=24mm | 0.211 | 0.262 | 0.307 | 0.361 | 0.394 | 0.310 | 0.050 |

| h=8cm,r=5cm,H=0.4m | | | | | Average Value | standard deviation |
|---|---|---|---|---|---|---|---|
| d=2mm | | 0.229 | 0.231 | 0.248 | | 0.236 | 0.010 |
| d=6mm | | 0.243 | 0.247 | 0.269 | | 0.253 | 0.014 |
| d=10mm | | 0.211 | 0.256 | 0.283 | | 0.25 | 0.036 |
| d=14mm | | 0.232 | 0.259 | 0.298 | | 0.263 | 0.033 |
| d=18mm | 0.213 | 0.228 | 0.284 | 0.316 | 0.354 | 0.276 | 0.045 |
| d=24mm | 0.207 | 0.250 | 0.281 | 0.321 | 0.348 | 0.284 | 0.036 |

| h=8cm,r=5cm,H=0.8m | | | | | Average Value | standard deviation |
|---|---|---|---|---|---|---|---|
| d=2mm | | 0.239 | 0.246 | 0.271 | | 0.252 | 0.017 |

| | | | | | | Average Value | standard deviation |
|---|---|---|---|---|---|---|---|
| d=6mm | | 0.252 | 0.268 | 0.281 | | 0.267 | 0.015 |
| d=10mm | | 0.261 | 0.284 | 0.304 | | 0.283 | 0.022 |
| d=14mm | | 0.230 | 0.288 | 0.298 | | 0.272 | 0.037 |
| d=18mm | 0.218 | 0.253 | 0.291 | 0.338 | 0.351 | 0.294 | 0.043 |
| d=24mm | 0.206 | 0.247 | 0.316 | 0.331 | 0.383 | 0.298 | 0.045 |

| h=8cm,r=5cm,H=1.2m | | | | | | Average Value | standard deviation |
|---|---|---|---|---|---|---|---|
| d=2mm | | 0.258 | 0.259 | 0.278 | | 0.265 | 0.011 |
| d=6mm | | 0.237 | 0.300 | 0.303 | | 0.28 | 0.037 |
| d=10mm | | 0.259 | 0.286 | 0.319 | | 0.288 | 0.030 |
| d=14mm | | 0.247 | 0.287 | 0.345 | | 0.293 | 0.049 |
| d=18mm | 0.251 | 0.266 | 0.298 | 0.342 | 0.354 | 0.302 | 0.038 |
| d=24mm | 0.236 | 0.272 | 0.306 | 0.361 | 0.381 | 0.313 | 0.045 |

| h=8cm,r=5cm,H=1.6m | | | | | | Average Value | standard deviation |
|---|---|---|---|---|---|---|---|
| d=2mm | | 0.244 | 0.279 | 0.296 | | 0.273 | 0.027 |
| d=6mm | | 0.268 | 0.284 | 0.309 | | 0.287 | 0.021 |
| d=10mm | | 0.254 | 0.307 | 0.336 | | 0.299 | 0.042 |
| d=14mm | | 0.271 | 0.311 | 0.348 | | 0.31 | 0.039 |
| d=18mm | 0.246 | 0.258 | 0.307 | 0.359 | 0.387 | 0.308 | 0.051 |
| d=24mm | 0.225 | 0.279 | 0.338 | 0.349 | 0.368 | 0.322 | 0.038 |

Table 6 Orthogonal test results (COR/Damage depth L) (supplemental material)

| r=2cm,H=0.4m,h=2cm,d=2mm | | | Average Value | standard deviation |
|---|---|---|---|---|
| 0.269/0.67 | 0.274/0.72 | 0.291/0.56 | 0.278/0.65 | 0.012/0.082 |
| r=2cm,H=0.8m,h=4cm,d=6mm | | | Average Value | |
| 0.253/0.79 | 0.268/0.75 | 0.298/0.68 | 0.273/0.74 | 0.023/0.056 |
| r=2cm,H=1.2m,h=6cm,d=10mm | | | Average Value | |
| 0.255/1.02 | 0.278/0.91 | 0.313/0.86 | 0.282/0.93 | 0.029/0.082 |
| r=2cm,H=1.6m,h=8cm,d=14mm | | | Average Value | |
| 0.266/1.10 | 0.298/1.01 | 0.321/1.04 | 0.295/1.05 | 0.028/0.046 |

| r=3cm,H=0.4m,h=2cm,d=6mm | | | Average Value | standard deviation |
|---|---|---|---|---|
| 0.287/0.60 | 0.287/0.62 | 0.308/0.52 | 0.294/0.58 | 0.012/0.053 |
| r=3cm,H=0.8m,h=4cm,d=2mm | | | Average Value | |
| 0.252/1.34 | 0.261/1.64 | 0.282/1.37 | 0.265/1.45 | 0.015/0.165 |
| r=3cm,H=1.2m,h=6cm,d=14mm | | | Average Value | |
| 0.275/1.17 | 0.319/1.08 | 0.357/0.84 | 0.317/1.03 | 0.041/0.171 |

| r=3cm,H=1.6m,h=8cm,d=10mm | | | Average Value | |
|---|---|---|---|---|
| 0.264/1.74 | 0.273/1.68 | 0.303/1.38 | 0.280/1.60 | 0.020/0.193 |

| r=4cm,H=0.4m,h=4cm,d=10mm | | | Average Value | standard deviation |
|---|---|---|---|---|
| 0.265/0.65 | 0.304/0.58 | 0.319/0.63 | 0.296/0.62 | 0.028/0.036 |
| r=4cm,H=0.8m,h=2cm,d=14mm | | | Average Value | |
| 0.304/0.51 | 0.354/0.68 | 0.356/0.49 | 0.338/0.56 | 0.029/0.104 |
| r=4cm,H=1.2m,h=8cm,d=2mm | | | Average Value | |
| 0.232/2.79 | 0.261/2.76 | 0.275/2.25 | 0.256/2.60 | 0.022/0.303 |
| r=4cm,H=1.6m,h=6cm,d=6mm | | | Average Value | |
| 0.247/2.58 | 0.286/2.19 | 0.319/1.83 | 0.284/2.20 | 0.036/0.375 |

| r=5cm,H=0.4m,h=4cm,d=14mm | | | Average Value | standard deviation |
|---|---|---|---|---|
| 0.283/0.62 | 0.300/0.68 | 0.344/0.53 | 0.309/0.61 | 0.031/0.076 |
| r=5cm,H=0.8m,h=2cm,d=10mm | | | Average Value | |
| 0.288/0.60 | 0.336/0.59 | 0.360/0.55 | 0.328/0.58 | 0.037/0.026 |
| r=5cm,H=1.2m,h=8cm,d=6mm | | | Average Value | |
| 0.251/2.37 | 0.291/2.01 | 0.298/1.98 | 0.280/2.12 | 0.025/0.217 |
| r=5cm,H=1.6m,h=6cm,d=2mm | | | Average Value | |
| 0.249/3.05 | 0.277/2.48 | 0.293/3.02 | 0.273/2.85 | 0.022/0.321 |

| r=2cm,H=0.4m,h=8cm,d=2mm | | | Average Value | standard deviation |
|---|---|---|---|---|
| 0.199/1.33 | 0.218/1.38 | 0.231/1.37 | 0.216/1.36 | 0.016/0.026 |
| r=2cm,H=0.8m,h=6cm,d=6mm | | | Average Value | |
| 0.239/1.28 | 0.267/1.32 | 0.289/1.12 | 0.265/1.24 | 0.025/0.106 |
| r=2cm,H=1.2m,h=4cm,d=10mm | | | Average Value | |
| 0.273/1.24 | 0.298/1.19 | 0.335/0.96 | 0.302/1.13 | 0.031/0.149 |
| r=2cm,H=1.6m,h=2cm,d=14mm | | | Average Value | |
| 0.319/0.75 | 0.361/0.59 | 0.394/0.70 | 0.358/0.68 | 0.038/0.082 |

| r=3cm,H=0.4m,h=8cm,d=6mm | | | Average Value | standard deviation |
|---|---|---|---|---|
| 0.211/0. 94 | 0.239/1.03 | 0.243/0.79 | 0.231/0.92 | 0.017/0.121 |
| r=3cm,H=0.8m,h=6cm,d=2mm | | | Average Value | |
| 0.243/1.64 | 0.258/1.55 | 0.267/1.28 | 0.256/1.49 | 0.012/0.187 |
| r=3cm,H=1.2m,h=4cm,d=14mm | | | Average Value | |
| 0.291/1.12 | 0.344/1.03 | 0.346/1.09 | 0.327/1.08 | 0.031/0.046 |
| r=3cm,H=1.6m,h=2cm,d=10mm | | | Average Value | |
| 0.324/0.77 | 0.347/0.83 | 0.382/0.92 | 0.351/0.84 | 0.029/0.076 |

| r=4cm,H=0.4m,h=6cm,d=10mm | | | Average Value | standard deviation |
|---|---|---|---|---|
| 0.254/0.88 | 0.284/0.81 | 0.323/0.62 | 0.287/0.77 | 0.035/0.135 |
| r=4cm,H=0.8m,h=8cm,d=14mm | | | Average Value | |
| 0.259/0.93 | 0.273/0.66 | 0.311/0. 84 | 0.281/0.81 | 0.027/0.137 |
| r=4cm,H=1.2m,h=2cm,d=2mm | | | Average Value | |
| 0.315/1.09 | 0.337/1.15 | 0.356/0.85 | 0.336/1.03 | 0.021/0.159 |
| r=4cm,H=1.6m,h=4cm,d=6mm | | | Average Value | |
| 0.291/1.83 | 0.312/2.05 | 0.351/2.00 | 0.318/1.96 | 0.030/0.115 |

| r=5cm,H=0.4m,h=6cm,d=14mm | | | Average Value | standard deviation |
|---|---|---|---|---|
| 0.272/0.72 | 0.295/0.64 | 0.309/0.65 | 0.292/0.67 | 0.019/0.044 |
| r=5cm,H=0.8m,h=8cm,d=10mm | | | Average Value | |
| 0.193/1.03 | 0.284/1.15 | 0.348/0.97 | 0.275/1.05 | 0.078/0.092 |
| r=5cm,H=1.2m,h=2cm,d=6mm | | | Average Value | |
| 0.323/1.03 | 0.346/1.22 | 0.372/1.17 | 0.347/1.14 | 0.025/0.098 |
| r=5cm,H=1.6m,h=4cm,d=2mm | | | Average Value | |
| 0.270/2.75 | 0.289/2.46 | 0.323/2.41 | 0.294/2.54 | 0.027/0.184 |

---

## Author Response (AR1)

Manuscript number: nhess-2018-16
MS Type: Research Article
Title: "The effects of cushion's particle size and thickness on coefficient of restitution under the rockfall impacts"
Correspondence Author: Chun Zhu, Chen Cao
Authors: Chun Zhu, Dongsheng Wang, Xing Xia, ZhiGang Tao, ManChao He, and Chen Cao.
Dear Editor Gibson, Katie:

Thank you very much for your attention and the referee's evaluation and comments on our paper "The effects of cushion's particle size and thickness on coefficient of restitution under the rockfall impacts". We have revised the manuscript thoroughly according to your kind advices and referee's detailed suggestions. Enclosed please find the responses to the referee. We sincerely hope this manuscript will be finally acceptable to be published on "Natural Hazards and Earth System Sciences". Thank you very much for all your help and looking forward to hearing from you soon.

Best regards
Sincerely yours
Chun Zhu, Chen Cao.

Answer to referee comments
**Comment 1**: the error analysis must be integrated into the paper and not included in the supplementary material. This should be possible.

**Response**: Thanks for the reviewer's suggestion. I have added the error analysis in the Table 1-5 in the paper, which can fully help readers judge the results against their experimental uncertainties. Thus these data can be presented in a way that the results are assessable to other researchers working in this field.

According to the method of Influencing factor range analysis of all evaluation indices, the trend lines in Figures 12 is the line for average value of the COR statistical value of factor $x$ at level $y$ ($y$=1, 2, 3. 4). Due to the definition of error bar, it is meaningful to obtain the error bar of the test data if all the tests are conducted with the same conditions, and the range analysis method don't require all the conditions are same, it just needs to calculate the average value of the COR statistical value of a factor $x_A$ (either of the four factors) at level $y$ ($y$=1, 2, 3. 4) in all the orthogonal test results, no matter what other factors $x_B$, $x_C$, $x_D$ (the other three of the four factors) are different, so it is inappropriate to calculated the error bar of every data in the Table 6, because every data are calculated based on different test parameters, thus I think the Figures 12 and 13 with no uncertainties is reasonable. However, I fully agree with the review's point that it should give uncertainties for readers, so I supplemented the standard deviation of COR and damage depth L for three tests results in Table 5.

According to the tables of standard deviation of COR and damage depth L, it can be seen the standard deviation are relative small, and the average vale of three test results of each test can be used for the subsequent range analysis.

**Comment 2:** the English must be dramatically improved to consider final publication. Please take the comments of reviewer 1 seriously.

**Response**: Thanks for the reviewer's suggestion. My manuscript has been edited for English by using an English editing services, the embellishment proof is supplied as the attachment. The format of my manuscript have been readjusted to facilitate the readers to understand based on journal requirements.

**Comment 3**: I therefore believe publication is warranted after major revisions. I believe the paper should be submitted as a brief communication. Try to shorten the paper and concentrate on the tests and the results. There is no need to have a long introduction to motivate the problem and laboratory tests. I will not allow the paper to be published unless there are significant improvements to the written English.

**Response**: Thanks for the reviewer's suggestion. I have revised the manuscript according to the rule of briefness, the introduction and laboratory tests section are shortened. My manuscript has been edited for English by using an English editing services, the embellishment proof is supplied as the attachment. The structure and format of my manuscript have been readjusted to facilitate the readers to understand based on journal requirements.

[Figure]

**CERTIFICATE OF ENGLISH EDITING**

This is to certify that the manuscript entitled

**The effects of gravel cushion particle size and thickness on the coefficient of restitution in rockfall impacts**

commissioned to us has been carefully edited by a native English-speaking editor of MogoEdit, and the grammar, spelling, and punctuation have been verified and corrected where needed. Based on this review, we believe that the language in this paper meets academic journal requirements. Please contact us with any questions.

[Figure]

*Gang Zhang*

Dr. Gang Zhang
Founder & CEO of MogoEdit

Date of Issue
April 28, 2018

**Disclaimer:** The changes in the document may be accepted or rejected by the authors in their sole discretion after our editing. However, MogoEdit is not responsible for revisions made to the document after our edit on **April 28, 2018**.

MogoEdit is a professional English editing company who provides English language editing, translation, and publication support services to individuals and corporate customers worldwide. As a company invested by the affiliate fund of Chinese Academy of Science, MogoEdit is one of the leading language editing service providers in China, whose clients come from more than 1000 universities and research institutes.

MogoEdit Website:    http://en.mogoedit.com/
500+ native English editors:    http://en.mogoedit.com/editors

[Figure]

Mogo Internet Technology Co., LTD.
No. 25, 1st Gaoxin Road, Xi'an 710075, PR China +86 02988317483 support@mogoedit.com

**The effects of gravel cushion particle size and thickness on **the** coefficient of restitution **in** rockfall impacts**

*Zhu Chun[1,2,3], Wang Dongsheng[2,3], Xia Xing[2,3], Tao ZhiGang[2,3], He ManChao[1,2,3], Cao Chen*[1]*

Corresponding Email:zhuchuncumtb@163.com; ccao@jlu.edu.cn (Corresponding Author)

*(1.    College of Construction Engineering, Jilin University, Changchun 130026, China)*

*(2.    State Key Laboratory for Geomechanics & Deep Underground Engineering, Beijing 100083, China)*

*(3.    School of Mechanics and Civil Engineering, China University of Mining & Technology, Beijing 100083, China)*

**Abstracts:** Gravel cushions are widely used to absorb the impact energy of falling rocks in open-pit mines. A particularly important application is to enhance the energy- absorbing capacity of rockfall sheds. In this paper, we study how varying the thickness and particle size of a gravel cushion  influences its energy- consumption and buffering effects. We performed a series of laboratory drop tests by dropping blocks from varying heights onto  cushions of different thicknesses and particle sizes. The results indicate that, for a given impact energy, the  cushion thickness has a strong influence on the measured coefficient of restitution (COR) and therefore impact pressure. Additional tests were performed to study how the radius of the block and the height it is dropped from affect the measured COR. This showed that as the movement height of the block is increased the COR also increases, and blocks with larger radii exhibit a larger variability in measured COR. Finally, we investigated the influence of rockfall block radius, *r*, movement height, *H*, cushion thickness, *h*, and particle size, *d*, on the *COR* and the damage depth, *L*, of the cushion. The test results reveal that the cushion thickness, h, is the primary design parameter, controlling not only COR but also the stability of the cushion material. The results provide a theoretical and practical basis for the design of gravel cushions for rockfall protection.

**Keywords:** Rockfall; cushion thickness; laboratory test; particle size; coefficient of restitution (*COR*).

**1 Introduction**

Rockfall constitutes a serious hazard in the working areas and facilities of the world's open-pit mines. Where slope surfaces are seriously weathered and the disturbing forces from mining are strong, landslides and rock-body collapse are prone to occur during rainfall. In rockfall, rocks roll down slope due to instability caused by gravity or exogenic action and come to rest at an obstacle or in the gentler part of the slope (Huang et al., 2007). Rockfall is widely distributed and occurs suddenly, posing a serious threat to life and property (Pantelidis, 2009; Pantelidis, 2010). In response to frequent rockfall disasters in recent years, numerous scholars in China and abroad have conducted in-depth studies into the characteristics of rockfall movement through theoretical analysis, field investigation, and numerical simulation. For example, Mignelli et al. (2014),  applied a rockfall risk management approach to the road infrastructure network of the Regione Autonoma Valle D'Aosta in order to calculate the level of risk and the potential for its reduction by rockfall protection devices. A comparative analysis of road accidents in the Aosta Valley was then undertaken to verify the methodology.

calculation modes for normal collision coefficient of restitution and tangential collision coefficient of restitution of spheres are studied, respectively (Thornton et al., 1998).  Asteriou et al. (2016) examined the effects of rock shape by performing tests with spherical and cubic blocks, finding that spherical blocks show higher and more consistent coefficient of restitution (*COR*) values than cubic blocks. Howald et al. (2017) evaluated the protective capacity of existing and newly proposed protection measures and considered the possible reclassification of hazard as a function of the mitigation role played by the measure. Furthermore, numerical simulation software has been adopted to analyze the characteristics of rockfall movement. The software ROCFALL 3.0 has been adopted in dam construction, road construction and the protection of historical places to calculate the velocity and locus of rockfall and avoid damage to the project (Topal et al., 2006; Koleini and Van Rooy, 2011; Saroglou et al., 2012; Sadagah, 2015). State-of-the-art simulation techniques incorporating nonsmooth contact dynamics and multibody dynamics have been applied to and adapted for the efficient simulation of rockfall trajectories, and the influence of rock geometry on rockfall dynamics has been studied through numerical simulation (Leine et al., 2014).

The research outlined above indicates that several types of protection measure can be effective in controlling rockfall. Trees have a significant blocking effect on rolling rocks. Interception influence tests of the effect of trees on rockfall have been designed based on analysis of the velocity change, the distance traveled by the rockfall, and the probability of collision between trees and rockfall ( Notaro, 2012; Monnet et al., 2017). Semi-rigid rockfall protection barriers have been installed along areas threatened by rockfall events, and Miranda et al. (2015) have carried out a numerical investigation of such protection barriers to obtain essential structural information such as their energy-absorption capacity . has been conducted into the impact caused of rockfall on reinforced concrete beams has been conducted  and the process of dynamic response  studied and compared with the results of numerical simulation (Kishi et al., 2002; Bhatti et al., 2009; Kishi et al., 2010; Bhatti et al. 2010). Kawahara et al. (2006) conducted a large number of experiments for different soils under different combinations of falling mass and drop height and studied 
[revised manuscript text omitted]

---

## Referee Report (RR1)

The authors' resubmission with the full experimental data alongside with their uncertainties inside the main text are a clear improvement of the presented work. The scientific relevance of the contribution is generally improved.

Most of the required amendments in the previous evaluation have been considered. As final remark remains: The author's concerns that Figure 9 and 10 become confusing when all the uncertainties are plotted is a fair remark. However, inserting a representative error bar to on data point might strengthen the scientific information contained in the plot.

Overall the manuscript is presented in a much better way and merits publication after some minor revisions:

- As the presentation of the data is rather space consuming, a compression of Figures 2,3,4,5,7 to a multipanel figure might be helpful. Especially Figure 5 (a) might undergo a revisions as well as Figure 6 - increase readability of text. The authors might reevaluate whether all those figures are necessary.
- Reconsider inserting representative error estimations/boundaries for Figure 9 and 10 as well as for Fig. 12 and 13.

Future work of Chun and co-authors should strongly focus on their English grammar (proof reading before initial submission is highly recommended), thorough estimation of experimental errors and transparent communication thereof as well as appealing figure work.

---

## Author Response (AR2)

Manuscript number: nhess-2018-16

MS Type: Research Article

Title: "The effects of cushion's particle size and thickness on coefficient of restitution under the rockfall impacts"

Correspondence Author: Chun Zhu, Chen Cao

Authors: Chun Zhu, Dongsheng Wang, Xing Xia, Zhigang Tao, Manchao He, Chen Cao.

Dear Editor Perry Bartelt:

Thank you very much for your attention and the referee's evaluation and comments on our paper "The effects of cushion's particle size and thickness on coefficient of restitution under the rockfall impacts". We have revised the manuscript thoroughly according to your kind advices and referee's detailed suggestions. Enclosed please find the responses to the referee. We sincerely hope this manuscript will be finally acceptable to be published on "Natural Hazards and Earth System Sciences". Thank you very much for all your help and looking forward to hearing from you soon.

Best regards
Sincerely yours
Chun Zhu, Dr.

Answer to referee 1 comments
    Reviewer #1

**R1(1)**: The paper has been further improved considering the suggestions from the reviewer. However, some of the questions need to be further clarified. The major problem is that the format of the paper is still less ideal. For example, some variables are italic but some are not. This occurs (but may not limited to) in the abstract for COR, in Equations 3-6, in Figure 5a, in Tables 1-6, and in Figures 9-10.

**AC**: Thanks for the reviewer's suggestion. My manuscript has revised carefully the format of this paper to achieve the requirements of Journal. The variables in the manuscript are all italic according to the format of Journal.

**R1(2):** The section 3 should be 'Experimental studies' instead of 'Experimental Studies'. In Figures 5b and 6 it is suggested to use different colors for the texts so that one can see more clearly. The axis labels in Figures 8-10 are different, i.e. some are with the variable name but some are not. The size of the figure title is not consistent such as in Figures 7 and 8. In Figures 8 and 10(r=3cm) the axis should not be bold. The quality of Figures 12-13 is worse compared to the other figures and need to be improved. Some variables are bold such as in line 298 and in Figures 12-13.

**AC**: Thanks for the reviewer's suggestion. I have replaced the 'Experimental Studies'

with 'Experimental studies'. All the figures in my manuscript have been reorganized carefully according to the reviewer's suggestions. The quality of Figures 12-13 have been improved to facilitate the readers to understand.

**R1(3):** The references are not following the same style, such as [2], [3], [16], [22], and [27]. In fact the numbering in the reference should be removed and the references should be formatted according to the requirement of the journal (ordered according to the author names). It is thus suggested that the authors VERY CAREFULLY check the paper again and ensure that the format issue is solved. On the other hand, a technique question is: In line 181 the authors mentioned 'If an obviously outlying result was obtained, the test was repeated to reduce the error'. Could the authors clearly define what is 'an obviously outlying result'?

**AC**: Thanks for the reviewer's suggestion. The references in my manuscript have been formatted according to the requirement of the journal (ordered according to the author names). I had stated that "if an obviously outlying result was obtained, the test was repeated to reduce the error." The obviously outlying results were the two rare conditions that $V_{COR}=0$ or $V_{COR} > 1$. When the blocks of a small radius collided with the cushion of large particle size, such as blocks of 2 cm radius collided with the cushion of 24 mm particle size, blocks can be stuck in the seam between the particles due to the occasionality. When the blocks with relative high kinetic energy collided with the cushion of large particle size, many particles had collided out from the platform, which may be captured by the cameras, posing some error, and these conditions are rare. Therefore if an obviously outlying result (a clear error) was obtained, the test was repeated to reduce the error.

Answer to referee 2 comments
    Reviewer #2

**R2(1)**: The authors' resubmission with the full experimental data alongside with their uncertainties inside the main text are a clear improvement of the presented work. The scientific relevance of the contribution is generally improved.

Most of the required amendments in the previous evaluation have been considered. As final remark remains: The author's concerns that Figure 9 and 10 become confusing when all the uncertainties are plotted is a fair remark. However, inserting a representative error bar to on data point might strengthen the scientific information contained in the plot.

Overall the manuscript is presented in a much better way and merits publication after some minor revisions:

□ As the presentation of the data is rather space consuming, a compression of Figures 2,3,4,5,7 to a multipanel figure might be helpful. Especially Figure 5 (a) might undergo a revisions as well as Figure 6 - increase readability of text. The authors might reevaluate whether all those figures are necessary. Reconsider inserting representative error estimations/boundaries for Figure 9 and 10 as well as for Fig. 12

and 13.

AC: Thanks for the reviewer's suggestion. My manuscript has reorganized carefully the figures according to the reviewer's suggestions. The Figures 2, 3, 4, 5, 7 have been compressed to avoid consuming space. I also have inserted representative error estimations/boundaries for Figure 9 and 10. However, the Figure 12 and 13 are not appropriate to be inserted representative error estimations/boundaries. According to the method of Influencing factor range analysis of all evaluation indices, the trend lines in Figures 12 is the line for average value of the COR statistical value of factor $x$ at level $y$ ($y$=1, 2, 3. 4). Due to the definition of error bar, it is meaningful to obtain the error bar of the test data if all the tests are conducted with the same conditions, and the range analysis method don't require all the conditions are same, it just needs to calculate the average value of the COR statistical value of a factor $x_A$ (either of the four factors) at level $y$ ($y$=1, 2, 3. 4) in all the orthogonal test results, no matter what other factors $x_B$, $x_C$, $x_D$ (the other three of the four factors) are different, so it is inappropriate to calculated the error bar of every data in the Table 3, because every data are calculated based on different test parameters, thus I think the Figures 12 and 13 with no uncertainties is reasonable. However, I fully agree with the review's point that it should give uncertainties for readers, so I supplemented the standard deviation of COR and damage depth L for three tests results in same test in Table 5. According to the tables of standard deviation of COR and damage depth L, it can be seen the standard deviation are relative small, and the average vale of three test results of each test can be used for the subsequent range analysis.

**R2(2)**: Future work of Chun and co-authors should strongly focus on their English grammar (proof reading before initial submission is highly recommended), thorough estimation of experimental errors and transparent communication thereof as well as appealing figure work.

AC: Thanks for the reviewer's suggestion. I and other co-authors have carefully checked the English grammar, and reappraised the experimental errors and transparent communication. Finally, my manuscript has been checked for English by using an English editing service.

[revised manuscript text omitted]